# Developing and exploring a theory for the lateral erosion of bedrock channels for use in landscape evolution models

Abigail L. Langston[1] and Gregory E. Tucker[2,3]

[1]Department of Geography, Kansas State University, Manhattan, USA
[2]Cooperative Institute for Research in Environmental Sciences (CIRES), University of Colorado, Boulder, USA
[3]Department of Geological Sciences, University of Colorado, Boulder, USA

*Correspondence to:* A.L. Langston (alangston@ksu.edu)

**Abstract.** Understanding how a bedrock river erodes its banks laterally is a frontier in geomorphology. Theory for the vertical incision of bedrock channels is widely implemented in the current generation of landscape evolution models. However, in general existing models do not seek to implement the lateral migration of bedrock channel walls. This is problematic, as modeling geomorphic processes such as terrace formation and hillslope-channel coupling depends on accurate simulation of valley widening. We have developed and implemented a theory for the lateral migration of bedrock channel walls in a catchment-scale landscape evolution model. Two model formulations are presented, one representing the slow process of widening a bedrock canyon, the other representing undercutting, slumping, and rapid downstream sediment transport that occurs in softer bedrock. Model experiments were run with a range of values for bedrock erodibility and tendency towards transport- or detachment-limited behavior and varying magnitudes of sediment flux and water discharge in order to determine the role each plays in the development of wide bedrock valleys. Results show that this simple, physics-based theory for the lateral erosion of bedrock channels produces bedrock valleys that are many times wider than the grid discretization scale. This theory for the lateral erosion of bedrock channel walls and the numerical implementation of the theory in a catchment-scale landscape evolution model is a significant first step towards understanding the factors that control the rates and spatial extent of wide bedrock valleys.

## 1 Introduction

Understanding the processes that control the lateral migration of bedrock rivers is fundamental for understanding the genesis of landscapes in which valley width is many times the channel width. Strath terraces are a clear indication of a landscape that has experienced an interval where lateral erosion has outpaced vertical incision (Hancock and Anderson, 2002). Broad strath terraces and wide bedrock valleys that are many times wider than the channels that carved them are found in mountainous and hilly landscapes throughout the world (e.g., Chadwick et al., 1997; Lavé and Avouac, 2001; Dühnforth et al., 2012) and provide clues about the nature of their evolution. Wide bedrock valleys and their evolutionary descendants, strath terraces, are erosional features in bedrock that are several times wider than the channels that carved them and range in spatial scale tens to thousands of meters (Figure 1). Wide bedrock valleys created by incising rivers provide the opportunity for sediment storage

in the valley bottom, influence hydraulic dynamics by allowing peak flows to spread out across the valley, and decrease the average transport velocity of sediment grains (Pizzuto et al., 2017).

Changes in climate that drive changes in sediment flux, changes in discharge magnitude, and/or changes in discharge frequency have been cited as causes of periods of lateral erosion in bedrock rivers. The frequency of intense rain is correlated with higher channel sinuosity and lateral erosion rates on regional scales (Stark et al., 2010). Several studies demonstrate that significant lateral erosion in rapidly incising rivers is accomplished by large flood events (Hartshorn et al., 2002; Barbour et al., 2009), resulting from cover on the bed during extreme flood events (Turowski et al., 2008) and exposure of the bedrock walls to sediment and flow (Beer et al., 2017). Sediment cover on the bed that suppresses vertical incision and allows lateral erosion to continue unimpeded is a critical element for the development of wide bedrock valleys, as determined from modeling, field, and experimental studies (Hancock and Anderson, 2002; Brocard and Van der Beek, 2006; Johnson and Whipple, 2010). Lateral erosion that outpaces vertical incision and creates wide bedrock valleys and strath terraces has been linked to weak underlying lithology, such as shale (Montgomery, 2004; Snyder and Kammer, 2008; Schanz and Montgomery, 2016), although strath terraces certainly exist in stronger lithologies, such as quartzite (Pratt-Sitaula et al., 2004). The relationships among river sediment flux, discharge, lithology, and rates of lateral bedrock erosion are not well defined. Because we do not sufficiently understand the processes of lateral erosion, landscape evolution models lack a physical mechanism for allowing channels to migrate laterally and widen bedrock valleys, in addition to incising bedrock valleys.

Existing landscape evolution models do not address the lateral erosion of bedrock channel walls and the consequential migration of the channel, in no small part because of the lack of a rigorous understanding of the processes that control lateral erosion of bedrock channel walls. If this theoretical hurdle can be cleared, an algorithm for lateral erosion must be applied within a framework of models that currently only erode and deposit vertically. To our knowledge, this study is the first attempt at incorporating a generalized physics-based algorithm for lateral bedrock erosion and channel migration on a drainage basin scale to a two-dimensional landscape evolution model.

## 2   Background

Theory for the vertical incision of bedrock channels has advanced considerably since the first physics-based bedrock incision models were presented in the early 1990's. For example, bedrock incision models now include theories for adjustment of channel width (e.g. Stark and Stark, 2001; Wobus et al., 2006; Turowski et al., 2009; Yanites and Tucker, 2010), the role of sediment size and bed cover (e.g. Whipple and Tucker, 2002; Sklar and Dietrich, 2004; Yanites et al., 2011), and thresholds for incision (e.g. Tucker and Bras, 2000; Snyder et al., 2003b). Rivers may respond to changing boundary conditions by adjusting both slope and channel width (Lavé and Avouac, 2001; Duvall et al., 2004; Snyder and Kammer, 2008, e.g.) and landscape evolution models must be able capture both of these responses if we are to fully describe the behavior and function of landscapes. Research on bedrock channel width gives important insights into the larger scale problem of bedrock valley widening. In particular, sediment cover on the bed plays an important role in the evolution of channel cross-sectional shape because sediment cover on the bed can slow or halt vertical incision (Sklar and Dietrich, 2004; Turowski et al., 2007), while

allowing lateral erosion to continue. Models of channel cross-sectional evolution predict that increasing sediment supply to a steady-state stream results in a wider, steeper channel for a given rate of base level fall (Yanites and Tucker, 2010). While theories that account for dynamic adjustment to bedrock channel width continue to be refined (for a review, see (Lague, 2014)), landscape evolution models that include a relationship between sediment size and cover (Gasparini et al., 2004), and incision thresholds in bedrock channels (Tucker et al., 2001; Crave and Davy, 2001; Tucker et al., 2013) are available and widely used

(Tucker and Hancock, 2010).

Numerical models for alluvial rivers have made considerable advances in capturing the planform dynamics both meandering and braided rivers, which necessarily include lateral bank erosion. Howard and Knutson (1984) developed the first numerical model that simulates lateral bank movement in alluvial rivers and produces realistic patterns of river meandering. In this study, bank erosion scales inversely with the radius of curvature, such that more rapid erosion occurs in tighter bends with a smaller

radius of curvature. A more recent treatment of radius of curvature as a control on lateral erosion rates has been implemented in CAESAR, a cellular landscape evolution model that calculates a 2-D flow field (Coulthard et al., 2002; Coulthard and van de Wiel, 2006; Coulthard et al., 2013). This model is appropriate for studying alluvial river dynamics in meandering or braided streams at reach and small catchment scales and time scales of up to thousands of years (Van De Wiel et al., 2007), but is not designed to model the evolution of bedrock rivers. The EROS model is a morphdynamic/hydrodynamic model that also

allows for lateral erosion of bank material (Crave and Davy, 2001; Davy and Lague, 2009; Carretier et al., 2016). In Eros, lateral erosion of bank material is equal to vertical erosion rate multiplied by the lateral topographic slope and a coefficient of unknown value (Davy and Lague, 2009). This treatment of lateral erosion allows for lateral channel mobility and the development of realistic braided rivers, but it lacks a mechanistic process, specifically for the lateral erosion of bedrock channels.

As noted above, considerable advances have been made in developing theory and models for the planform dynamics of

single-thread meandering channels. As a result, the scientific community has a good understanding of how meander patterns form and evolve, and how meander wavelength and migration rate scale with properties such as water discharge, valley gradient, and sediment grain size (e.g., Hooke, 1975; Schumm, 1967; Nanson and Hickin, 1986; Sun et al., 2001; Lancaster and Bras, 2002; Parker et al., 2011). This body of work addresses the planform pattern of river channels, but does not deal with the broader drainage-basin topography in which those channels are embedded. The principal state variable in channel-meander models is

the trace of the channel, $\mathbf{x}(\lambda)$, where $\lambda$ represents streamwise distance $\mathbf{x} = (x, y, t)$ is the channel centerline position. Some more recent models also incorporate a vertical channel coordinate, so that $\mathbf{x} = (x, y, z, t)$ (e.g., Limaye and Lamb, 2013), but the emphasis remains on the channel trace rather than on the topography. For example, the slope of the channel and/or valley is normally treated as a boundary condition rather as an element of topography that evolves dynamically as it steers the flow of water, sediment, and energy.

There is also a well-developed literature on process models of landscape evolution, and in particular the evolution of ridge-valley topography sculpted around drainage networks. We refer to these models as Landscape Evolution Models, or LEMs (e.g., Coulthard, 2001; Willgoose, 2005; Tucker and Hancock, 2010; Valters, 2016; Temme et al., 2017). With LEMs, the emphasis lies in computing the topographic elevation field, $\eta(x, y, t)$. Water and sediment cascade passively downhill across this surface. In some of these models, channel segments are assumed to exist as sub-grid-scale features that are free to switch

direction arbitrarily as the topography around them changes. Other LEMs represent water movement as a two-dimensional flow field, whether through multiple-direction routing algorithms (e.g., Coulthard et al., 2002; Pelletier, 2004; Perron et al., 2008) or with a simplified form of the shallow-water equations (Adams et al., 2017; Simpson and Castelltort, 2006). Regardless of the approach to flow routing, LEMs differ from meander models in treating a self-forming, two-dimensional flow network rather than a single channel reach, and in explicitly modeling the evolution of topography.

Lateral migration of bedrock channel walls has only been implemented into landscape evolution models in a limited number of studies (Lancaster, 1998; Hancock and Anderson, 2002; Clevis et al., 2006a; Finnegan and Dietrich, 2011; Limaye and Lamb, 2013). Hancock and Anderson (2002) model bedrock valley widening using a 1-D stream power model for vertical incision and assume that valley widening rates depend on stream power. They note that the width of the valley floor is related to the duration of steady state in the river, as theorized by Suzuki (1982). This model is based on the key observation that lateral erosion exceeds vertical incision when the channel is carrying the maximum sediment load dictated by the transport capacity. By varying sediment supply to the channel, their model predicts the development of a series of strath terraces. Strath terrace sequences have also been produced by coupling a meandering model with a river incision model (Finnegan and Dietrich, 2011). Clevis et al. (2006a) modeled meandering channels in a valley section using a 2-D landscape evolution model and an adaptive grid approach. A vector-based approach to modeling lateral migration of meandering streams in heterogeneous bed material has been used to reproduce a range of bedrock valley forms (Limaye and Lamb, 2014), but this model is primarily a channel-scale model. While each of these studies model lateral migration of bedrock channel banks, they all operate with a meandering model that is not applicable to lateral migration in low-sinuosity channels or in a generalized landscape evolution model.

## 3 Approach and Scope

Until now, landscape evolution models have lacked a generic mechanism for allowing channels to migrate laterally and widen bedrock valleys, as well as incise bedrock valleys. While advances in controls on bedrock valley width have been made using meandering models, the representation of a sinuous channel doesn't describe all rivers, and often such models are constructed on a channel scale rather than on a drainage basin scale. In this study, we develop a theory for the lateral migration of bedrock channel walls and implement this theory in a 2-D landscape evolution model for the first time. We seek to explore the parameters that exert primary control on the morphology of bedrock valleys and the rate of bedrock valley widening using a series of numerical experiments.

With a few exceptions noted below, most LEMs treat erosion and sedimentation as purely vertical processes. When the flow of water and sediment collects in a "digital valley", the elevation of that location may rise or fall, but lateral erosion by channel impingement against a valley wall is usually neglected. Yet nature seems to be perfectly capable of forming erosional river valleys much wider than the channels they contain (Figure 1). The question arises of how one might honor the process of valley widening by lateral erosion (and narrowing by incision) within the topographically oriented framework of a LEM. In other words, how might the key features of LEMs and channel-planform models be usefully combined?

In addressing this issue, it is useful to consider that the typical LEM treatment of topography as a two-dimensional field $\eta(x,y,t)$ is itself a simplification, albeit a practical one. Consider an alternative framework in which the boundary between solid material (rock, sediment, soil) and fluid (air, water) is treated as a surface in three-dimensional space, $\sigma(x,y,z,t)$ (Braun et al., 2008). The surface possesses, at each point, a surface-normal velocity, $\dot{\sigma}$, which represents the combined surface-normal rates of erosion, sedimentation, and tectonic motion. Such a framework would lend itself to representing lateral erosion, because any movement of this surface where it is not flat implies a horizontal component of motion. The cost of such an approach lies in computational complexity. For practical reasons, it is desirable to find methods by which a lateral component of erosion by stream channels could be represented within the much simpler framework of a two-dimensional elevation field $\eta(x,y,t)$.

In this paper, our objective is to define and explore a theory for lateral erosion that has the following characteristics: simple and sufficiently general in nature to be applicable in landscape evolution models; containing as few parameters as possible; requiring relatively few input variables, such as channel gradient and water discharge plus gross channel planform configuration. The aim of this theory is to model valley widening or narrowing over time scales relevant to drainage basin evolution, and across multiple branches within a drainage network. The theory is not designed to predict the movement of a particular channel segment over a period of a few years, but rather is intended to provide a general basis for understanding when and why valleys tend to narrow or widen during the course of their long-term geomorphic evolution. Theoretical predictions about these trends then serve as quantitative, mechanistically based hypotheses that can be tested by experiment and observations. Through a set of numerical experiments, we seek to answer the following set of questions:

– How does this lateral erosion model compare with purely vertical erosion models?

– How do two alternative formulations, which treat bank material differently, compare to each other?

– What combinations of bedrock erodibility, sediment mobility, water flux, sediment flux, and model type result in wide bedrock valleys?

– What are predictions of the model that could be readily tested through experiment and/or observation?

In the following sections we outline our theory for lateral channel wall migration and explain the two algorithms we have developed to apply this theory to an existing model. We then present the results from our set of numerical experiments and discuss how well the model describes the formation of wide bedrock valleys. The approach presented here is intended to be a starting point, but not an ending point. Our main goal is to draw attention to the importance of lateral stream erosion within the context of drainage-basin evolution, and to offer some ideas for how this might be addressed in the framework of a conventional grid-based LEM.

## 4 Theory

We have deliberately chosen the most simple formulation possible for deposition and erosion, while still capturing the role of sediment. We do this in order to focus on developing the lateral erosion component of our model. Evolution of the height of

the landscape, $\eta$, through time is described by deposition rate, $d$, minus erosion rate, $e$, plus a constant rate of uplift relative to baselevel, $U$.

$$\frac{\partial \eta}{\partial t} = -e + d + U \tag{1}$$

Deposition rate is assumed to depend on the concentration of sediment ($C_s$) in active transport and its effective settling velocity, $\nu_s$. Sediment concentration is expressed as the ratio of volumetric sediment flux, $Q_s$, to water discharge, $Q$:

$$C_s = \frac{Q_s}{Q} \tag{2}$$

We treat water discharge as the product of runoff rate and drainage area, such that $Q = RA$. Deposition rate is therefore given by:

$$d = \frac{\nu_s d_* Q_s}{RA} \tag{3}$$

where $d_*$ is a dimensionless number describing the vertical distribution of sediment in the water column, which is equal to 1 if sediment is equally distributed through the flow (Davy and Lague, 2009). $\nu_s$, $d_*$, and $R$ are lumped into a single dimensionless parameter, $\alpha$, that represents the potential for deposition.

$$\alpha = \frac{\nu_s d_*}{R} \tag{4}$$

A large value for $\alpha$ implies more rapid deposition (all else being equal), either because settling velocity, $\nu_s$, is high and sediment is quickly lost from the flow, or because runoff rate, $R$ is low and there is little water in the channels to dilute the sediment. A small value for $\alpha$ represents slower settling velocity, or more intuitively, greater runoff. $\alpha$ can be thought of as a sediment mobility number: when $\alpha < 1$, sediment is easily transported and the model tends towards detachment-limited behavior; when $\alpha > 1$, sediment is less mobile and the model tends towards transport-limited behavior.

## 4.1 Vertical erosion theory

In the model presented here, we use the stream power incision model (e.g., Howard, 1994) to calculate vertical incision rate; the stream power model is the simplest bedrock incision model that represents fluvial erosion for steady state topography. Vertical erosion rate is derived from the rate of energy dissipation on the channel bed, which is given by:

$$\omega_v = \rho g \frac{Q}{W} S \tag{5}$$

where $\rho$ is the density of water, $g$ is gravitational acceleration, $Q$ is water discharge, $W$ is channel width, and $S$ is channel slope. We assume that the rate of vertical erosion scales as:

$$E_v = K_v' \frac{\omega_v}{C_e} \tag{6}$$

where $K_v'$ is a dimensionless vertical erosion coefficient and $C_e$ is cohesion of bed and bank material. We use bulk cohesion simply as a convenient reference scale for rock resistance to erosion. This choice allows us to express erosion rate as a function of the hydraulic power applied ($\omega_v$), a commonly used measure of material strength ($C_e$), and a dimensionless efficiency factor ($K_v'$).

We assume that channel width is a function of discharge (Leopold and Maddock, 1953):

$$W = k_w Q^{0.5} \tag{7}$$

where $k_w$ is a width coefficient. It is important to recognize that channel width is not explicitly represented in the model we describe. Rather, it is one element of the lumped parameters $K_v$ and $K_l$ (erosion coefficients discussed below). The channel-width scaling parameter values we discuss ($k_w$) are used only in the estimation of reasonable ranges for these parameters. The bank width coefficient, $k_w$, is constant along the channel length based on data sets from both alluvial (Leopold and Maddock, 1953) and bedrock rivers (Montgomery and Gran, 2001) that show a relationship between channel width and discharge. Substituting $RA$ for $Q$ and equation 7 for $W$ in equation 5 and then combining equations 5 and 6 gives:

$$E_v = \frac{K_v' \rho g R^{1/2}}{k_w C_e} A^{1/2} S \tag{8a}$$

$$E_v = K_v A^{1/2} S \tag{8b}$$

Lumping several parameters gives $K_v$, a dimensional vertical erosion coefficient (with units of years$^{-1}$), which consists of known or measurable quantities, and one unknown dimensionless parameter, $K_v'$.

Although evidence indicates that sediment in the channel plays an important role in inciting lateral erosion in bedrock channels (Finnegan et al., 2007; Johnson and Whipple, 2010; Fuller et al., 2016), the model presented here uses the stream power incision model to represent vertical erosion, which does not account for sediment flux-dependent incision (e.g., Beaumont et al., 1992; Sklar and Dietrich, 2004; Turowski et al., 2007). The standard stream power model (Equation 8) has some limitations, especially in the lack of threshold effects and assumption of constant channel width (Lague, 2014). Despite these limitations, the stream power model is a good approximation for long term vertical bedrock incision on large spatial scales (e.g. Howard, 1994; Whipple and Tucker, 1999) and is appropriate here given the goal of this work is to explore dynamics of lateral bedrock erosion as a function of channel curvature.

## 4.2 Lateral erosion theory

Lateral erosion requires hydraulic energy expenditure to damage the bank material and/or dislodge previously weathered particles (Suzuki, 1982; Lancaster, 1998; Hancock and Anderson, 2002). Consistent with earlier meandering models (e.g.

Howard and Knutson, 1984), we hypothesize that the lateral erosion rate is proportional to the rate of energy dissipation per unit area of the channel wall created by centripetal acceleration around a bend. Erosion of the channel wall is the result of the force of water acting on the channel wall. We know from basic physics that the force of water acting on the wall is equal to the force of the wall acting on the water, which is equal to centripetal force. Centripetal force is $F_c = m\frac{v^2}{r_c}$, where $m$ is mass, $v$ is velocity, and $r_c$ is radius of curvature. The centripetal force of a unit of water can be found by replacing $m$ with $\rho LHW$, where $\rho$ is the density of water, and $L$, $H$, and $W$ are unit length, water depth, and channel width, respectively. Centripetal force of water flowing around a bend can be expressed in terms of centripetal shear stress, which is analogous to bed shear stress, by dividing both sides by $HL$ giving:

$$\sigma_c = \frac{\rho W v^2}{r_c} \tag{9}$$

Centripetal shear stress can be turned into a rate of energy expenditure by multiplying by fluid velocity, giving:

$$\omega_c = \frac{\rho W v^3}{r_c} \tag{10}$$

To express this in terms of discharge, $Q$, instead of velocity, we employ the Darcy-Weisbach equation, giving $v^3 = gqS/F$, where $q$ is discharge per unit width and $F$ is a friction factor, which yields

$$\omega_c = \frac{\rho g Q S}{r_c F} \tag{11}$$

Equation 11 describes a quantity that might be termed centripetal unit stream power, as it represents the rate of energy dissipation per unit bank area. The centripetal unit stream power is similar to the more familiar quantity unit stream power, except that channel width is replaced by the radius of curvature multiplied by a friction factor.

We hypothesize that lateral erosion rate scales with energy dissipation rate around a bend according to

$$E_l = K_l' \frac{\omega_c}{C_e} \tag{12}$$

where $K_l'$ is a dimensionless lateral erosion coefficient. Combining equations 11 and 12 gives

$$E_l = \frac{K_l' \rho g R}{C_e F} \frac{AS}{r_c} \tag{13a}$$

$$E_l = K_l \frac{AS}{r_c} \tag{13b}$$

where $K_l$ is a dimensional erosion coefficient for lateral erosion composed of known or measurable quantities, and one unknown dimensionless parameter, $K_l'$. If $K_l'$ is equal to $K_v'$, we find a ratio between $K_l$ and $K_v$, given by

$$\frac{K_l}{K_v} = \frac{R^{1/2} k_w}{F} \tag{14}$$

which consists of runoff rate, $R$, bank width coefficient, $k_w$, and friction factor, $F$. We can measure or make reasonable estimates of each of these parameters in order to determine what the ratio of lateral to vertical erodibility should be. Mean annual runoff rate can vary widely, but a higher peak runoff intensity will lead to a higher $K_l/K_v$ ratio and more lateral erosion.

A fixed $k_w$ is common in landscape evolution models that model long term landscape erosion (e.g., Tucker et al., 2001;
Gasparini et al., 2007), but channel width can vary with incision rate in models and natural systems (Yanites and Tucker, 2010; Duvall et al., 2004), suggesting there are cases when dynamic width scaling is important (Lague, 2014). In this model, $k_w$ is given a value of 10 m/(m$^3$/s)$^{1/2}$, which is reasonable for natural rivers (Leopold and Maddock, 1953), but the value can range between 1 and 10 due to differences in runoff variability, substrate properties, and sediment load (Whipple et al., 2013). The friction factor, $F$, is the Darcy-Weisbach friction factor, which can range from 0.01–1.0 for natural rivers (Gilley et al., 1992;
Hin et al., 2008). With a lower friction factor (representing smooth channel walls), the lateral erosion ratio would be higher due to less energy being dissipated on the channel walls, leaving more energy available for lateral erosion.

## 5   Numerical implementation

One challenge in modeling both vertical and lateral erosion in a drainage network lies in the representation of topography. Typically, landscape evolution models use a numerical scheme in which the terrain is represented by a grid of points whose
horizontal positions are fixed and whose elevation represents the primary state variable in the model. Such a framework does not lend itself to the motion of near-vertical to vertical interfaces (such as stream banks and cliffs), and for this reason, incorporating lateral stream erosion in a conventional landscape evolution model requires a modification to the basic numerical framework. A vertical rather than horizontal grid (Kirkby, 1999) can be used for near-vertical landforms in isolation, but is inappropriate when one wishes to represent vertical interfaces that are inset within a larger landscape. Grid-node movement combined with
adaptive re-gridding (Clevis et al., 2006a, b) provides a possible solution, but is computationally expensive, and particularly difficult to implement when multiple branches of a drainage network may undergo lateral motion. Here, we adopt a simpler approach in which valley walls are viewed as sub-grid-scale features that migrate through the fixed grid. Rather than tracking the position of these vertical interfaces, we instead track the cumulative sediment volume that has been removed from the cell surrounding a given grid node as a result of lateral erosion. When that cumulative loss exceeds a threshold volume, the
elevation of the grid node is lowered.

More specifically, at each node in the model, we calculate a vertical incision rate at the primary node and a lateral erosion rate at a neighboring node (Figure 2). The lateral neighbor node for the primary node is chosen on the outside bank of two stream segments that flow into and out of the primary node. The stream segments used to identify the neighboring node over which lateral erosion should occur are the incoming stream segment to the primary node with the greatest drainage area and the stream segment that connects the primary node to its downstream neighbor (Figure 2). If the two segments are straight, then a neighboring node of the primary node is chosen at random and lateral erosion occurs at this node until elevation changes at the node.

Calculation of radius of curvature along two stream segments in a raster grid with D8 flow routing presents a challenge, as the angle between segments is discretized; the two segments may form a straight line, in which case the angle is equal to $0°$, form a $45°$ angle, or form a $90°$ angle. In order to reduce the impact of this discretization, we assume that each of these three cases represents a continuum of possible radii of curvature. Cases of two straight segments are treated as if the actual angle between them ranges anywhere between $+22.5°$ to $-22.5°$. If one takes the average among these possible angles, the resulting inverse radius of curvature is $0.23/dx$, where $dx$ is the cell size in the flow direction. Similarly, we assume that a $45°$ bend represents a continuum of possible angles between the two segments, ranging from $22.5°–63.5°$, resulting in an inverse radius of curvature of $0.67/dx$. Following the same principle for a $90°$ bend gives a mean inverse radius of curvature of $1.37/dx$ (see Supplementary Materials).

The volumetric rate of material eroded laterally for each lateral node is calculated by $E_l \times dx \times H$, where $H$ is water depth, given in meters. Water depth at each node is calculated by $H = 0.4Q^{0.35}$ (Andrews, 1984), where $Q$ is given in $m^3/s$. The volume of sediment eroded laterally per time step is sent downstream along with any material eroded from the primary cell. Volumetric erosion rate is multiplied by the time step duration to get the volume eroded at the lateral nodes, and the cumulative volume eroded from each lateral node is tracked throughout the entire model run. When cumulative volume eroded from the lateral node equals or exceeds the volume needed to erode the node (see end member model descriptions below), the elevation of the lateral node is set to the elevation of the downstream node (Figure 2). Flow is then rerouted and water flows down the path of steepest descent. The model does not distinguish between sediment and bedrock in the model grid and all material that is eroded has the bedrock erodibility of the $K_v$ or $K_l$ terms. When material is eroded vertically or laterally from bedrock nodes, the volume of the eroded material is sent downstream as part of the $Q_s$ term. If deposition occurs in the model, deposited material is added to the topography of the node as bedrock. Thus, sediment is not "seen" in the model as material that can be easily re-eroded after deposition, rather sediment works to increase the deposition term (Equation 3).

Lateral erosion rate presented here (Equation 13) relates lateral erosion to radius of curvature, but the application of this model is not limited to meandering streams. Streams with fully developed meandering are part of a relatively small subset of streams that are able to widen valleys through lateral erosion; there are examples of streams that are classified as single-thread or braided, and yet which clearly show evidence of erosion and lateral migration at locations where an outer bend in the channel impinges on a valley wall or terrace (Cook et al., 2014; Finnegan and Balco, 2013). Conceptually, therefore, this approach is not meant to represent exclusively channels with fully developed meandering.

## 5.1 End member model formulations

We have implemented two ways of determining whether enough lateral erosion has occurred to lower the lateral node. The first method, the total block erosion model, dictates that the entire volume of the lateral node above the elevation of the downstream node must be eroded before its elevation is changed (Figure 2a,b). This formulation assumes that the height of the valley walls is a controlling factor in the ultimate width a valley can achieve, thus valley width scales with valley wall height. In this method, lateral migration depends on bank height so that taller banks experience slower lateral migration, as all of the volume of the lateral node must be eroded for the valley to widen. The second method, the undercutting-slump model, dictates that only the

volume of the water height on the bank times the cell area must be eroded for the elevation to change (Figure 2c,d), while the remaining material slumps into the channel and is transported away as wash load, i.e. not redeposited in the model or included in $Q_s$ calculations. This model formulation represents the migration of valley walls independent of valley wall height. With these two end member models, we address whether lateral erosion rate should scale with valley wall height. Valley wall or bank height is known to limit lateral channel migration and valley width in transport limited streams where additional sediment from valley walls cannot be transported out of the channel (Nicholas and Quine, 2007; Bufe et al., 2016; Malatesta et al., 2017). However whether valley wall height should limit valley widening in detachment-limited bedrock channels is less clear (Lancaster, 1998), and likely depends on the bedrock lithology (Finnegan and Dietrich, 2011; Johnson and Finnegan, 2015). The links between these end member model formulations and the natural processes they represent are explored in the discussion section.

## 6   Model experiments

In order to constrain the conditions that result in significant lateral bedrock erosion and valley widening, we ran sets of models using a range of values for bedrock erodibility, $\alpha$ (sediment mobility number), and $K_l/K_v$ ratio using both the total block erosion model and the undercutting-slump model (Table 1). The model domain was 600 m by 600 m with 10 m cell size, three closed boundary edges and uplift rate relative to baselevel of 0.0005 m/yr imposed on the entire model domain. Water flux was introduced at the top of the model by designating a node as an inlet with an area of 20,000 m$^2$ and sediment flux at carrying capacity. This setup allowed each run to have a primary channel on which to measure width and channel mobility. All models were spun up to an initial condition of approximately uniform erosion rate with vertical incision only. The models were then run for 100–200 ky with the lateral erosion component. In order to isolate the effect of bedrock erodibility, a set of model calculations were run where erodibility ranged from $5\times10^{-5}$ to $2.5\times10^{-4}$ (Stock and Montgomery, 1999) while $\alpha$ was held constant at 0.8. In order to isolate the effect of detachment-limited vs. transport-limited behavior, another set of models was run where erodibility was held constant at $1\times10^{-4}$ and $\alpha$ values ranged from 0.1 to 2, which represents a detachment-limited system when $\alpha < 1$ and a transport-limited system when $\alpha > 1$ (Davy and Lague, 2009) (Table 1). $K_l/K_v$ ratios for all model runs were set to 1.0 or 1.5, resulting in a runoff rate of 14 mm/hr or 36 mm/hr from Equation 14. These runoff rates do not represent a yearly mean annual runoff, rather peak event runoff rates that are likely to result in appreciable lateral erosion due to the scaling with $K_l/K_v$ ratio. Small et al. (2015) found that bedrock erosion rates in abrasion mill experiments are an order of magnitude higher in samples from channel margins compared to the channel thalweg. This suggests that $K_l$ in this model should be at least equal to $K_v$, and could be much higher (Finnegan and Dietrich, 2011).

Understanding the model behavior in response to detachment- vs. transport-limited behavior (represented by $\alpha$) and $K_l/K_v$ ratio is complex and requires understanding how runoff plays into both parameters. The value of $\alpha$ is calculated by $v_s$, a proxy for grain size, and runoff rate, $R$, although neither grain size nor runoff is explicitly set in the model runs. Values of $\alpha$ that capture a range of detachment- or transport-limited behavior is set instead ($\alpha$=0.2–2.0). When $K_l/K_v$ ratio is set for a given model (either 1.0 or 1.5 in all model runs), the runoff rate is calculated inside the model. Once a runoff rate for given $K_l/K_v$

ratio is calculated, by extension, a value of $v_s$ can be calculated from runoff rate and the set $\alpha$ value. Therefore, in model runs with the same $K_l/K_v$ ratio and therefore the same runoff rate, a transport-limited system ($\alpha$ greater than 1) has a larger grain size (approximated by $v_s$) compared to a detachment-limited system with a low $\alpha$.

**Table 1.** Model runs and parameters discussed in this paper.

| model version | $K_l/K_v$ | $K$ | $\alpha$ | number of runs |
|---|---|---|---|---|
| total block | 1.0–1.5 | $1 \times 10^{-4}$ | 0.2–2.0 | 10 |
| total block | 1.0–1.5 | $5 \times 10^{-5}$–$2.5 \times 10^{-4}$ | 0.8 | 10 |
| undercutting-slump | 1.0–1.5 | 0.0001 | 0.2–2.0 | 10 |
| undercutting-slump | 1.0–1.5 | 0.00005–0.00025 | 0.8 | 10 |
| | | | | |
| TB water flux | 1.0–1.5 | 0.00005–0.0025 | 0.8 | 6 |
| UC water flux | 1.0–1.5 | 0.00005–0.0025 | 0.8 | 6 |
| TB sed flux | 1.0–1.5 | 0.0001 | 0.2–2.0 | 10 |
| UC sed flux | 1.0–1.5 | 0.0001 | 0.2–2.0 | 10 |

## 6.1 Measures of lateral erosion in model landscapes

### 6.1.1 Channel mobility

Channel mobility distinguishes models with lateral erosion from models with only vertical incision. At steady state, channels in models with only vertical bedrock incision do not migrate across the model domain. However, a mobile channel is necessary to carve wide valleys and it is enticing to say that the more mobile the channels, the wider the bedrock valley. In our model, channel mobility is not controlled by sediment flux, as found in alluvial channels (Wickert et al., 2013; Bufe et al., 2016), but by the lateral erosion of bedrock. However the term "channel mobility" is used here in the same sense as in alluvial literature; channel mobility describes lateral channel planform changes along the length of the channel.

The effect of bedrock erodibility and $\alpha$ on channel migration through time for both model versions is shown in Figure 3. Channel migration over 200 ky is shown for six selected runs that span the range of bedrock erodibility and $\alpha$ values for the two different model formulations: the undercutting-slump model where $K_l/K_v$=1.5 and the total block erosion model where $K_l/K_v$=1.5. In all runs, the total block erosion model produced more confined channels compared to the undercutting-slump model. The undercutting-slump model produces more dynamic channel migration over the model domain, especially in the high $K$ model. In both model formulations, the high $K$ and high $\alpha$ runs have the widest extent of channel migration (recall that high $\alpha$ represents lower sediment mobility) and the low $K$ and low $\alpha$ runs have the most restricted channel migration.

In order to describe channel mobility in our model runs in a single term, we calculate a cumulative migration metric, $\lambda$. $\lambda$ is calculated by first determining the migration distance of the channel between time steps at all model cells the main channel occupies. Most often, the migration distance between time steps at a single cell will be 0 or 10 m, indicating no migration or migration to a neighboring cell. The mean of migration distances between time steps is taken and summed over the duration of the model run to give the cumulative migration metric. $\lambda$, indicates how often the channel has migrated during the model run; a model run can have the same $\lambda$ value if the channel marches across the entire model domain or if the channel repeatedly switches between two nearby channel courses. $\lambda$ can also be used as an indicator for the maximum lateral extent occupied by the channel during the model run. That is, the maximum possible extent of x positions occupied by the channel is equal to $\lambda$, but the actual x distance occupied by the main channel could be lower as the channel migrates over the same area repeatedly.

Bedrock erodibility and $K_l/K_v$ ratio have the strongest control on channel migration distance. Channel mobility increases as bedrock erodibility increases in both the total block erosion model and the undercutting-slump model (Figure 4a,b). When $K$ is low, representing strong bedrock lithology, there is limited channel movement in the total block erosion models with $\lambda$ values between 15–35 m. This means that on average during the model run the channel occupied 1–3 cells (Figure 3c). With low values of $K$, the undercutting-slump model had $\lambda$ values around 200 m, but a lateral extent of only 5 model cells (Figure 3c). This indicates that in the undercutting-slump model, the channel was actively migrating within a small area of the model domain. In model runs with high $K$ values representing weak bedrock, total channel migration, $\lambda$ increases, as well as the spatial extent of the channel migration (Figure 3a). With the total block model, $\lambda$ appears to be a good proxy for total spatial extent of channels, but for the undercutting-slump model, $\lambda$ tends to over estimate lateral extent of channel occupation (Figure 3).

Increasing the $K_l/K_v$ ratio from 1.0 to 1.5, results in 1.5–2 times more channel mobility, with the largest relative increases in total block erosion model runs with high erodibility and higher $\alpha$ values (Figure 4a,b). This is because the undercutting-slump models already have high channel mobility with $K_l/K_v$ equal to 1. Increasing $K_l/K_v$ ratio to 1.5 increases channel mobility in UC models, but the total block erosion models have a larger threshold for lateral erosion so the increased $K_l/K_v$ ratio results in relatively more channel mobility in the total block models.

For model runs with the same bedrock erodibility, but different $\alpha$ values (which represents sediment mobility), channel mobility is lower in models with lower values of $\alpha$ (representing high sediment mobility) and higher when $\alpha > 1$ (representing less mobile sediment) (Figure 4b). In the following presentation and discussion of models with varying values of $\alpha$, high $\alpha$ model runs will be termed "transport-limited" and low $\alpha$ model runs will be termed "detachment-limited". This effect is most pronounced in the total block erosion models, where channel mobility increases by a factor of four as $\alpha$ increases. In the undercutting-slump models, channel mobility also increases with $\alpha$, especially when $K_l/K_v = 1.5$. When $K_l/K_v = 1$ in the undercutting-slump models, the trend in channel mobility vs. $\alpha$ is less well defined.

### 6.1.2 Valley width

Valley width is the primary indicator of lateral erosion; a wide bedrock valley implies that significant lateral erosion has occurred relative to vertical incision. Valleys can be defined in a few different ways; valley width needs to be quantified in our

model. Many studies use low gradient areas of a DEM to determine valley width (e.g., Brocard and Van der Beek, 2006; May et al., 2013). This gives the width for the valley bottom that has been shaped by channel processes, but excludes areas that have been recently shaped by channel processes and then reworked by hillslope processes. Another way to measure valley width is

by determining the width of the valley at a certain height above the channel. This simple metric is often used for finding valley width in the field, for example using eye height above the channel (e.g., Snyder et al., 2003a; Whittaker et al., 2007). Using a certain height above the channel to determine valley width in the models cannot distinguish between a fluvially carved bedrock valley and low relief in a landscape with weak bedrock. Instead we define valley width as the width of the area perpendicular to the main channel where slope is characteristic of the fluvial channel rather than hillslopes for a given bedrock erodibility

and $\alpha$ value. The reference slope for a fluvial channel is given by the slope-area relationship, assuming that the height of the landscape and $Q_s$ are steady in time. When the height of the landscape is in equilibrium, Equations 1 and 3 are combined and rewritten as:

$$U = e - \frac{\nu_s d_* Q_s}{RA} \qquad (15)$$

At steady state, $Q_s$ is the total upstream eroded material, given by $Q_s = AU$. Substituting the steady state equation for $Q_s$ and

Equation 8 into Equation 15 gives

$$U = K_v A^{1/2} S - \alpha U \qquad (16)$$

Solving the above equation for $S$ gives the equation for reference slope that determines whether a model cell is shaped by fluvial or hillslope processes (Davy and Lague, 2009).

$$S = \frac{U}{K_v A^{1/2}} (\alpha + 1) \qquad (17)$$

25       Our models successfully produce bedrock valleys that are several model cells wider than the channels that created them (Figure 5). Models with only vertical incision have v-shaped valleys that are only 1 model cell wide (10 meters in our experiments) and the channels do not shift laterally (Figure 5a). Given the specifications of the total block and undercutting-slump models, it is not surprising that the total block models take longer to respond to the onset of lateral erosion and valleys are more narrow than in the undercutting-slump models. The total block erosion models take on the order of 10 ky to produce an observable

response to lateral erosion and ultimately produce bedrock valleys that are up to 25 meters wide, while the undercutting-slump models take about 5 ky show a response to lateral erosion and ultimately produce valleys that are up to 50 m wide.

Figure 6 shows slope maps of total block and undercutting-slump models that show the width of the valley shaped by fluvial processes. The blue areas have slopes that are characteristic of fluvial channels and red areas have slopes that are characteristic of hillslopes. The total block erosion model with a low $\alpha$ value shows very little bedrock valley widening as evidenced by the thin band of blue along the main channel 1–2 model cells wide (Figure 6a). Increasing $\alpha$ to obtain transport-limited behavior in the model results in wider valleys that have been shaped by the channel that are 2-3 model cells wide in the total block erosion

model (Figure 6b). The landscape in the undercutting-slump model has wider valleys that result from more extensive carving

by channels. The fluvially carved valleys in the detachment-limited model are about 2-3 model cells wide and the valleys in the transport-limited model are over 50 meters wide in some places (Figure 6c,d).

Figure 4c,d shows valley width for the lower two-thirds of the model channels averaged over the duration of the model runs in 54 model runs. To ensure that using characteristic fluvial slope as the criterion for a valley in all model runs gives valley width resulting from lateral erosion, and not valley width inherent in the model, we first use this criterion to measure valley width

for the spin up models that include no lateral erosion component. Valley width for the spin up models is consistently 10 m, the width of one model cell. Valley width does not change significantly for any of the total block model runs in which $K$ is varied and $\alpha$ is held constant (Figure 4c). When the $K_l/K_v$ ratio is increased from 1 to 1.5, valley width increase slightly for all model runs, but wide valleys are not possible in the total block erosion model with this value of $\alpha$. Valley width in the undercutting-slump model for changing bedrock erodibility shows a somewhat counter-intuitive signal (Figure 4c); the undercutting-slump

model results in wider valleys for lower values of bedrock erodibility. The reasons for this signal are discussed in the section below.

When $\alpha$ is varied and K is constant, valley width increases with the tendency towards transport-limited conditions ($\alpha > 1$) in all undercutting-slump models, but only in total block erosion models when the $K_l/K_v$ ratio is equal to 1.5 (Figure 4b). The widest valleys for a given bedrock erodibility occur with high $\alpha$ values as a result of higher slope. The models predict

more channel mobility and wider valleys under transport-limited streams (set by $\alpha$) compared to detachment-limited streams (Figure 4b,d). As $\alpha$ increases, the deposition term increases, and a steeper slope is needed to maintain the landscape in steady state relative to uplift. Higher channel slopes in transport-limited model runs also cause increased lateral erosion according to equation 13.

### 6.1.3    Linking channel mobility and valley width

We have shown that the greatest channel mobility occurs in the undercutting-slump models and increases significantly with increasingly soft bedrock (Figure 4a). However, maximum channel mobility does not translate into maximum valley width. In the undercutting-slump models, the widest valleys occur in the low erodibility model runs that have relatively low channel mobility. This reflects that the areas visited by the migrating channel in the low-relief, high K model runs are easily over-printed by small scale fluvial processes and lose the slope signature of the larger channel. This prevents our algorithm from

finding where an area of the model that has recently been shaped by the channel. The mismatch between channel mobility and valley width also reflects that hard bedrock valleys are allowed to erode very easily in the undercutting-slump model and the surface smoothed by the channel is persistent through time. The relationship between hard bedrock and wide valleys reflects the use of the undercutting-slump model, which is inappropriate for hard bedrock wall erosion in natural systems. With the undercutting-slump model, only a small volume threshold must be overcome for lateral erosion to occur, and the rest of the node material is transported downstream as wash load. However, models that have resistant bedrock (low $K$) are least suitable for the undercutting-slump model. In order for this model to be a good description of how nature works, the bed material would

need to be able to break up into small pieces that are easily transported away, which is conceivable for resistant clay banks.

However, the total block erosion model is generally more appropriate for representing the erosion of resistant bedrock channels that erode into material transported as bedload.

## 6.2 Adding complexity: water flux, sediment flux

### 6.2.1 Effects of increased discharge on lateral channel migration

In order to investigate how transience in landscapes affects lateral erosion, we introduce increased discharge at the inlet point in the upstream end of the model. Using drainage area as a proxy for discharge, increasing water flux in the model represents how a larger stream on the same landscape influences valley width. Increasing drainage area also allows us to observe the extent of landscape change and how rapidly the different model runs respond to an event such as stream capture. The drainage area at this input point is increased from 20,000 m$^2$ to 160,000 m$^2$ and sediment load is set to the carrying capacity of the new

drainage area. For a typical model run, the additional drainage area approximately doubles the magnitude of drainage area at the outlet of the main channel in the model domain, i.e., maximum drainage area in models runs increases from $\sim$1e5 m$^2$ to $\sim$3e5 m$^2$. Models with increased water flux were run using both model formulations, $K_l/K_v = 1.0$ and 1.5, and erodibility values that ranged from $5 \times 10^{-5}$–$2.5 \times 10^{-4}$, with alpha held constant at 0.8 (Table 1).

  Recalling that lateral erosion scales with drainage area (Equation 13), while vertical incision scales with the square root of

drainage area (Equation 8), we therefore expect that increasing drainage area will increase lateral erosion and valley width in every case for the undercutting-slumping model, where the numerically imposed condition for lateral erosion to occur is much smaller than in the total block erosion model. In the total block erosion model, lateral erosion temporarily stalls because of the volume threshold that must be exceeded before lateral erosion occurs. There is no threshold for vertical incision, which speeds up when additional water flux is added to the model.

### 6.2.2 Total block erosion models

  In all of the model runs, increased water flux resulted in increased lateral erosion and wider valleys. Figure 7 shows valley width averaged over the model domain vs. model time for all of the water flux models. The total block erosion model and undercutting-slump model respond differently to a step change in water flux. The total block erosion models first incise vertically to a new steady state stream profile, then erode laterally as a result of the increased water flux (Figure 8), while the

undercutting-slump model incises vertically and erodes laterally simultaneously (Figure 9).

  Total block erosion models where the $K_l/K_v$ ratio is equal to 1.5 (TB1.5) show an interesting pattern in valley widening after increased water flux (Figure 7c). All of the TB1.5 model runs show a significant increase in valley width during the 50 ky period of increased water flux. After 6 ky of increased water flux (model time = 106 ky), the high and medium erodibility model runs have greater valley widths, but the low erodibility model shows a gradual increase in valley width over 14 ky of increased water flux (model time 100–114 ky). For the first 14 ky of the increased water flux, the channel of the low K model run incises rapidly, increasing the gradient between the channel and the adjacent cells and preventing lateral erosion. After the

channel profile comes into new equilibrium, the increased water flux accelerates lateral erosion on the valley walls and valleys widen by 10 m compared to before increased water flux in the total block erosion models.

After the increased water flux stops at 150 ky, the wider valleys persist for ∼10–20 ky in the low and medium erodibility models (Figure 7c) for two reasons. First, after the cessation of increased water flux, the channel returns to equilibrium through aggradation and uplift. While aggradation occurs, lateral erosion can occur more easily in the total block erosion models. In this case, the total volume that must be eroded from any lateral node cell is reduced as the channel floor moves up in vertical space. The second reason for persistent wide valleys is that in the medium and low K model runs, the increase in water flux eroded

wide valleys into relatively resistant bedrock. These flat surfaces near the channel persist in harder bedrock, even after water flux has decreased to original levels. Following the end of the period of increased water flux, valley width in the the TB1.5 medium K model run remains elevated for 10 ky (model time 160 ky), before channel narrowing that propagates upstream (Figure 10). After cessation of the increased water flux at 150 ky, the channel profile returns to equilibrium through uplift and aggradation (Figure 10a). Channel aggradation begins at the bottom of the channel profile and results in a convexity that

propagates upstream (Figure 10a). At model position y=400, from 150–158 ky, the channel increases in elevation due to uplift (Figure 10b). Wide valleys created during increased water flux are maintained, and new lateral erosion of valley walls is seen (Figure 10b). At 159 ky, 9 ky after the cessation of increased water flux, the aggradational knickpoint reaches y=400 and incision and valley narrowing is observed (Figure 10d,e).

     Figure 8 shows surface topography and cross sections across the model domain for three times in the low erodibility model

run using the total block erosion model. This figure demonstrates the effect of valley deepening, then widening in response to increased water flux. Before water flux is increased, the channel is narrow and has steep valley walls (Figure 8a). After 15 ky of increased water flux and increased vertical incision, the topography reaches a new equilibrium and channel elevation is stationary. Only after this period of re-equilibration can lateral erosion begin to widen the valleys. After 30 ky of increased water flux, the entire channel has incised, especially in the upper valley. At y=420, the position of the cross section, the channel

has been incised by 3 m, and the valley has widened to about 20 m (Figure 8c). This response of primarily vertical incision is expected when using the total block erosion model, which sets a high threshold for lateral erosion.

### 6.2.3   Undercutting-slump models

In the undercutting-slump models, all of the model runs show a significant increase in channel mobility with additional water flux (Figure 7b,d). The largest valley widths occur in the models with low bedrock erodibility for reasons discussed above.

Unlike the total block erosion models, there is no discernible lag between onset of water flux and valley widening in the undercutting-slump models (Figure 9). This is because erosion of the valley wall is independent of the height of the valley wall for the undercutting-slump model formulation and the increase in drainage area results in larger increases in lateral erosion rates faster compared to vertical incision rates (Equation 8, 13). Figure 9 shows topography and cross sections for three times in the low erodibility model run using the undercutting-slump model. Before water flux is increased, the channel is significantly wider than in the total block erosion model. The cross section shows a 30 m wide valley, with low gradient areas next to the channel, indicating that these areas were shaped by the lateral erosion (Figure 9a). Following the increase in water flux,

the valley is much wider across the entire model domain, especially at the upstream segments of the channel. After 15 ky of increased water flux, the channel has both vertically incised and widened the valley to ∼40 m at y=420 (Figure 9b). After 30 ky of increased water flux, the valley has widened further to ∼60 m at y=420 (Figure 9c). The undercutting-slump model runs with medium and low erodibility maintain increased valley width after water flux has decreased, particularly in $K_l/K_v = 1.5$ models (UC1.5) (Figure 7d). This indicates that wide valley floors can persist for long periods of time after the conditions that created them have stopped.

### 6.2.4   Effects of increased sediment flux on lateral erosion

In order to explore how the addition of sediment to a stream affects lateral erosion and valley widening, we added sediment to the inlet point at the top of the model. The sediment flux models were run for 100 ky with 50 ky of standard lateral erosion followed by 50 ky of increased sediment flux. Before additional sediment was added, the sediment flux at the inlet was equal to the carrying capacity of the stream, which is equal to $UA$. Models with increased sediment flux were run using both model formulations, $K_l/K_v = 1.0$ and 1.5, and $\alpha$ values that ranged from 0.2–2.0, with bedrock erodibility held constant at $1 \times 10^{-4}$ (Table 1). During the 50 ky periods of increased sediment flux, five times more sediment flux was added, forcing all of the streams to aggrade initially. Adding sediment increases the deposition term (Equation 3), which results in aggradation if the model is initially in steady state, that is $e - d = U$. Aggradation in the channels continues until the channel slopes become steep enough to increase the vertical erosion term so that $e - d = U$ again, and the landscape is in a new equilibrium state. In this model, no distinction is made between the erodibility of deposited material and bedrock; any deposited material in the model has the properties of bedrock rather than sediment. The model responds to changes in sediment flux by adjusting channel slope, rather than both slope and channel width as observed in natural systems (Yanites et al., 2011) because of the fixed width scaling in this model.

Figure 11 shows valley width averaged over the upper half of the model domain (closest to the sediment source) plotted against model time. After sediment is added to the models, all of the model runs show a significant increase in valley width, except the low $\alpha$ model runs, which show little change in width. Valley width increases more and valleys stay wide for longer with higher values of $\alpha$. Valleys are narrowest and least persistent through time in the TB1 model group (Figure 11a), and valleys are widest and most persistent through time in the UC1.5 model group (Figure 11d). Valley widths and duration of wide valleys after the addition of sediment are similar between the TB1.5 group and the UC1 group (Figure 11b,c). The addition of sediment to these models results in channel aggradation and valley filling that accounts for a substantial fraction of measured increases in valley width for all of these model runs. It is not possible to distinguish between widening due to valley filling and widening due to bedrock wall retreat from this spatially averaged value of valley width. Lower values of $\alpha$ showed little or no increase in bedrock valley width after the addition of sediment flux. This is because channels in the low $\alpha$ runs (high sediment mobility) easily adapt to the increased sediment flux without significant or far-reaching changes to the channel slope. It is interesting to note that mean valley width increases at 50 ky for all model runs, then declines to close to pre-sediment values by about 80 ky. Mean valley width begins to decline as the models come into steady state with the increased sediment flux, indicating that lateral erosion can most readily occur when the channel is in a transient, aggradational state.

Figure 12 shows an example of simultaneous valley filling and significant bedrock erosion in the TB1.5 model group. Before the addition of sediment flux (t=50ky), the channel is 10 meters wide. Other channels shown in the cross section (at 80 m and 250 m) are immobile and show little evidence of lateral erosion. After the addition of sediment to the model, the main channel aggrades by 4 meters while also shifting 30 meters to the right, eroding a significant amount of bedrock valley wall over 20 ky.

Figure 13 shows the $\alpha = 1.5$ run from model group UC1.5, before and after added sediment flux that results in true bedrock valley widening. At 50 ky in the model run before the additional sediment is added, the valley in the upper half of the model domain (y=240) is about 30 m wide (Figure 13a). Over 50 ky, sediment is added to the model and the channel aggrades for ~20 ky before it comes into steady state, i.e., its slope is steep enough to carry the additional sediment load and aggradation stops. During the 20 ky of aggradation, this model run shows both retreat of the valley walls and channel aggradation. By 70 ky in the model run, the channel has aggraded by 5 meters and the valley is 50 m wide (Figure 13b). During this 20 ky period, the channel has migrated 50 m to the right, eroding the hillslope and forming steep valley walls.

Before the increase in sediment flux, all channels are in equilibrium by definition. Adding sediment to the inlet point in the models causes the channels to aggrade in all model runs, increasing the channel slope. This increase in channel slope increases the lateral erosion term and the vertical erosion term (Eqs. 8, 13); but while the channel is aggrading, vertical incision is effectively zero. Therefore, for the total block erosion models, most new lateral erosion should occur while the channel is aggrading, because the threshold volume that must be eroded becomes smaller when relief between the channel node and neighboring nodes decreases (Figure 2). Figure 11 shows that after sediment flux is added, there is a persistent increase in valley width for many model runs even after the channel profile has come into steady state with respect to the added sediment flux. The permanent increase in slope should result in higher lateral erosion rates, resulting in permanently wider valleys because the increased vertical incision rates that result from the higher slope is offset by increased deposition. This suggests the possibility that if a channel experiences increased slope through aggradation, then more lateral erosion occurs.

# 7 Discussion

## 7.1 Comparison among purely vertical incision models and end member lateral erosion models

The simple theory for lateral bedrock channel erosion presented here, combined with a landscape evolution model produces valleys that are several times wider than the channels they hold. The development of wide valleys is sensitive to the end member model formulation selected, which is discussed below. The widest valleys in this set of models occur in transport-limited model runs (high $\alpha$ values) when using the undercutting-slump model formulation, which represents lateral erosion that is independent of valley wall height. Wider bedrock valleys under conditions of relatively immobile sediment (high $\alpha$ value) (Figure 6) reflect conditions observed in natural systems, where wide bedrock valleys are considered a diagnostic feature of transport-limited streams (Brocard and Van der Beek, 2006). The results presented here show that the lateral erosion component allows for mobile channels in all model runs (Figure 4a,b), even when the model has reached steady state, unlike models with vertical incision only which have stationary channels at steady state. The modeling experiments show that landscapes with highly erodible bedrock have the most mobile channels. In the total block erosion model formulation, weak bedrock allows greater

channel mobility because the amount of lateral erosion that must occur to erode valley walls is lower in low-relief landscapes

with easily eroded bedrock (Whipple and Tucker, 1999). The model also predicts more channel mobility and wider valleys in models with high values of $\alpha$ (low sediment mobility), especially in the total block erosion models.

Channel mobility is a critical factor in the development of wide bedrock valleys, because all of the erosion of the valley must be accomplished through erosion by the channel (e.g., Tomkin et al., 2003). The width of surfaces beveled by lateral erosion has been framed as a competition between channel mobility and relative rock uplift rate (Bufe et al., 2016), with greater channel

mobility resulting in more area shaped by lateral erosion. The mobility of river channels increases with increasing sediment flux (Wickert et al., 2013), which emphasizes the potential importance of high sediment load as a requirement for the development of wide bedrock valleys. Landscapes in weaker bedrock are more likely to have more channel mobility and wider valleys (e.g., Montgomery, 2004; Snyder and Kammer, 2008; Schanz and Montgomery, 2016). Rivers flowing through soft bedrock are also more likely to behave as transport-limited rivers, as a result of the increased sediment flux in the stream from the surrounding

hillslopes and lower channel slopes in easily eroded bedrock. Channel mobility as a parameter extracted from the model is also important because measures of channel mobility during periods of lateral planation (e.g., Reimann et al., 2015) can be used to validate future lateral erosion models.

The two model formulations presented here describe end member behavior for how lateral erosion of valley walls scales with wall height, and can also be considered in terms of the physical processes of valley widening found in natural systems.

The total block erosion model, in which the entire volume of a neighboring node must be eroded before lateral erosion can occur, best describes lateral erosion in resistant and/or material that erodes into blocks that are not easily transported by the stream. This approach is used to represent, in a simple way, a system in which the undermining of a channel bank leads to gravitational collapse of resistant material that must itself then be eroded in place (Lancaster, 1998). The dependence of rates of valley widening on wall height has been demonstrated in alluvial systems where sediment transport rates in the channel are

low relative to the sediment eroded from valley walls (Bufe et al., 2016; Malatesta et al., 2017). One can imagine a similar limitation in bedrock gorges where lateral valley wall movement is accomplished through rockfall into the river (Shobe et al., 2016). Valley widening may also be limited when valley wall height exceeds the height of the flood stage; Collins et al. (2016) notes that vertical erosion of flat surfaces next to the channel can result in orders of magnitude greater valley erosion rates than lateral erosion rates alone suggest.

The undercutting-slump algorithm represents lateral erosion of valley walls that is independent of bank height. This model represents lateral erosion on a bank that has been laterally undercut and the remaining material slumps into the channel and is transported away as wash load, i.e. not added to the $Q_s$ term or redeposited in the model. The undercutting-slump model is applicable in locations with an under-capacity stream and lithology that slumps easily and rapidly breaks down into small grains that are easily transported (Finnegan and Dietrich, 2011; Johnson and Finnegan, 2015). Lateral erosion that is independent of valley wall height allows the development of wider bedrock valleys (Figure 6); the mechanism described by the undercutting-slump model more likely occurs in weak bedrock that breaks down into easily transportable grain sizes as observed in many natural systems (e.g., Montgomery, 2004; Snyder and Kammer, 2008; Schanz and Montgomery, 2016). The undercutting-slump model consistently produces wider bedrock valleys and more mobile channels than the total block erosion model because

less lateral erosion is required to erode valley walls in the undercutting-slump model algorithm. However, this undercutting-slump model is not appropriate for landscapes with very hard bedrock (low erodibility), as evidenced by overhanging cliffs along many rivers and persistent blocks of collapsed material following slumping or delivery from adjacent hillslopes (Shobe et al., 2016). The behavior of the models varies significantly based on which model is selected, although the same general trends are seen in both models. In nature, lateral erosion of valley walls likely follows neither one of these end members perfectly, but

will operate on a continuum between the two (Lancaster, 1998). Tomkin et al. (2003) presented two end member relationships between channel erosion and valley erosion that are similar to the models presented in this study, and found similar behavior between their two models.

## 7.2  Model limitations and future directions

While the model captures several important markers of lateral bedrock erosion, such as mobile channels and bedrock valleys

that are up to 5–6 times the channel width, the model did not develop broad, smooth, valleys that are up to 100 times the width of their channel (Figure 1) and that are sustained over many years, as observed in flights of strath terraces in the Front Range of Colorado, for example (Foster et al., 2017). Some important elements of reality have been simplified or omitted in this model, and future versions of the model should address: 1) resolving the effects of grid resolution on total lateral erosion and valley width, 2) setting runoff variability and magnitude separately from grain size, 3) including tools and cover effects and thresholds

in the vertical incision model, 4) treating sediment and bedrock erodibility separately.

In LEMs that use single-direction flow-routing schemes, such as the model presented here, it is possible in principle to have an "implied width" (implied by the width-discharge relation embedded in K (Equation 8)) that is larger than a grid-cell size. This issue is not unique to our particular model; any non-hydrodynamic LEM with sufficient resolution faces the same inconsistency. We explored the effects of a modification to the model where lateral erosion rate is calculated to account for both

the position of the channel within the model cell and cases where implied channel width is greater than the cell size (Figure S6, S7). Using a flow routing algorithm that allows flow to be distributed to two downstream pixels when the implied width is greater than the pixel size is a justifiable adaptation that would improve the hydrodynamic handing of water flow in this model, particularly with smaller pixel sizes. However, the intent of developing this new lateral erosion model within a LEM was to investigate how lateral erosion might be implemented within the context of an otherwise fairly generic and common model

formulation, without excessive complexity.

Sensitivity tests were conducted to explore the effect of grid size on total lateral erosion and valley width during model runs with dx=10 m, 15 m, and 20 m (Figure S8–S12). Grid size effects on cumulative lateral erosion are particularly pronounced in the total block erosion formulation (Figure S8), due to the increased volume that must be eroded for lateral erosion to occur when grid size is increased. Using the total block erosion model, where lateral erosion scales with valley height, larger grid

size can result in less lateral erosion, more narrow valleys, and longer response times for lateral erosion to occur. Using the undercutting-slump model resulted in valley widths estimated from cross sections and slope maps that are reasonably similar among models with dx=10 m, 15 m, and 20 m. The finding that grid size affects the magnitude of lateral erosion and valley widening to varying degrees is a limitation of the model that must be overcome before model parameters can be calibrated.

Grid-scale effects have been previously documented in LEMs, and achieving solutions that are grid-scale independent remains an open challenge (Passalacqua et al., 2006; Ganti et al., 2012). In the case of lateral erosion, we suggest that identifying and implementing a sub-grid scaling factor so that valley width becomes independent of cell size in all model realizations is needed in order to predict absolute timing and magnitude of lateral widening. There are several strategies that could usefully be explored, including use of multi-direction routing schemes to represent flow dispersion (Tarboton, 1997; Shelef and Hilley, 2013), and use of downscaling techniques to correct for resolution bias (Passalacqua et al., 2006).

In order to focus on implementing the equations for lateral erosion into the model, the simplest possible erosion-deposition model was used. This erosion-deposition model (Equation 1) has the advantage of not requiring the calculation of transport capacity and prevents potential problems with abrupt transitions from erosion to deposition, but does so at the expense of losing some details of runoff rate and grain size, which are lumped into the parameter $\alpha$. In this model, detachment- or transport-limited behavior is set through $\alpha$, which works well for general model exploration, but becomes problematic when exploring specific model responses to spatial and temporal changes in runoff rate and multiple grain sizes. Setting runoff and grain size explicitly is an important next step for determining how these factors independently impact bedrock valley width and channel mobility. Including a dynamic $K_l/K_v$ that is calculated with runoff from discrete events and channel widths is a target for future models. Runoff rate can vary widely, but a higher runoff intensity will lead to a higher $K_l/K_v$ ratio and more lateral erosion, as suggested by field observations of lateral erosion in bedrock channels during large flood events (Hartshorn et al., 2002) and correlation of increased sinuosity and storminess of climate (Stark et al., 2010). The model presented here does not have the capability to represent changes in $K_l/K_v$ based on processes that cause increased lateral erodibility, such as changes in the distribution of sediment during high flow (Hartshorn et al., 2002) or increased mass wasting of hillslopes (Stark et al., 2010). More process specific representation of $K_l/K_v$ ratio is a target for future model development.

The model presented in this paper uses the stream power incision model, the simplest reasonable vertical incision model, in order to focus on our goal of exploring the novel application of lateral bedrock erosion in a landscape evolution model. Using a tools and cover incision model (e.g., Beaumont et al., 1992; Sklar and Dietrich, 2004; Gasparini et al., 2004; Turowski et al., 2007) in a future lateral erosion bedrock model would be closer to the way we conceptualize lateral erosion in natural systems. The main impact of using a tools and cover incision model in a lateral erosion model would be less efficient vertical incision as relative sediment flux increases (Hobley et al., 2011). Slowing vertical erosion so that lateral erosion can catch up is an important part of the mechanism cited by many studies for allowing lateral erosion in incising streams (Hancock and Anderson, 2002; Turowski et al., 2008; Johnson and Whipple, 2010). Slowing vertical incision may be a necessary condition for significant lateral erosion and bedrock valley widening, but it is not by itself a sufficient condition. A model that describes how sediment tools carry out lateral erosion needs to be constructed (Fuller et al., 2016), but tools and cover incision models do not offer any mechanism for changing the rate of lateral erosion, just decreasing the efficiency of vertical incision.

Another limitation of the current model is that sediment is not treated explicitly, but rather is tracked in the model through the $Q_s$ term. No distinction in erodibility is made between sediment and bedrock. In the current model, when the landscape is in steady state, vertical erosion minus deposition is equal to the uplift rate. Increasing sediment flux, $Q_s$, in the deposition term immediately results in channel aggradation and increasing channel slope. In natural systems, channels respond to increased

sediment flux by increasing both slope and width. Changes in channel width are not captured in this model due to the fixed value of $k_w$, which is appropriate for landscapes in quasi-equilibrium (Whipple et al., 2013). How bedrock channel width responds to changes in boundary conditions, such as uplift rate and sediment, is the subject of ongoing research (e.g., Lague, 2014; Whittaker et al., 2007; Turowski et al., 2009), with important implications for driving channel incision of slump deposits and terrace generation (Croissant et al., 2017).

In not differentiating between sediment and bedrock explicitly in this model, the different erodibilities of sediment and bedrock are not accounted for. In most cases, sediment in a channel should be much easier to erode than the bedrock in a channel, allowing more rapid lateral migration through cells that have previously been occupied and are contain some amount of sediment (Limaye and Lamb, 2013). But in some cases, sediment in a soft bedrock channel can be composed of coarse grained, resistant lithology sourced from upstream. For example, the streams that drain the Colorado Front Range flow from hard, crystalline bedrock onto soft, friable shale bedrock (Langston et al., 2015). The granitic cobbles that cover the channel bed in stream segments underlain by shale bedrock, take much more energy to move than it does to transport the friable flakes of shale that line the walls of the channel. Different erodibilities should also result in more active channel migration once a wide valley is established because the channel erodes laterally through sediment that is more easily eroded than bedrock (Limaye and Lamb, 2014).

### 7.3   Comparison between models and field studies

Lateral erosion rates depend on the magnitude of shear stress and tools applied to channel walls, and the resistance of the bedrock to erosion. Our model of lateral bedrock erosion proposes that channel curvature controls lateral erosion rate. Cook et al. (2014) showed that extremely efficient bedrock wall erosion of up to ∼80 m over 5 years occurred where the river encountered sharp bends. They attribute this rapid lateral bedrock erosion in river bends to abrasion from sediment particles that detach from flow lines in the curve and impact the wall. Fuller et al. (2016) also suggest that lateral erosion rate by bedrock abrasion depends on how often sediment particles are deflected towards the channel walls, specifically by channel roughness elements. There is an important distinction between this study and the work of Fuller et al. (2016) in that their conclusions are based on observations of lateral erosion in a straight flume. Lateral erosion that occurs in the absence of channel curvature highlights the point that channel curvature is not the only control on lateral erosion, but it appears to be an important one.

The total block erosion model demonstrates how landscapes with hard bedrock and detachment-limited conditions respond to increased discharge by first incising the channel bed, and then widening after the channel has come into equilibrium (Figure 8). This behavior is similar to narrowing and incision of bedrock channels in response to increased uplift (Duvall et al., 2004) or vertical incision followed by channel widening in response to increased discharge (Anton et al., 2015). The model predicts that not only will channels in easily eroded bedrock reach equilibrium more quickly than channels in resistant bedrock, but valleys will begin to widen faster in easily eroded bedrock than in more resistant bedrock (Lavé and Avouac, 2001).

One of the few studies that has been able to report bedrock valley widening through time is from a unique case in Death Valley (Snyder and Kammer, 2008). Stream capture increased the drainage area of a small basin by 75 fold in the 1940's and channel response over the following 60 years was mapped by aerial photos. Snyder and Kammer (2008) found that mean

valley width in a channel segment with weak bedrock increased by 9 meters in 60 years. In contrast, in channel segments in hard bedrock, they found vertical channel incision and the development of knickpoints. They attribute the difference in response to lithological differences and suggest that the presence of sediment on the bed in the weak bedrock channel segments protects the bed from incision, allowing the valley walls to migrate laterally. This difference in response is similar to the behavior of the

end-member models presented here: the total block erosion model shows rapid incision and narrowing in response to increased water flux, whereas the undercutting-slump models show incision and valley widening.

Lateral erosion in nature is often attributed to increased sediment delivery to channels, which suppresses vertical bedrock incision and gives lateral erosion a chance to become the dominant mode of bedrock erosion (e.g. Bull, 1990; Hancock and Anderson, 2002). If this is the case, then we expect increased sediment flux to have the largest effect on the low $\alpha$/detachment-

limited model runs. The same amount of new sediment was added to each model run, but the sediment resulted in more aggradation in the high $\alpha$ runs. In the high $\alpha$/transport-limited runs, the channels already behave as if they are loaded with sediment. In low $\alpha$ runs, the model tends towards detachment-limited behavior, so additional sediment is rapidly and easily transported out of the system. The slope needed to transport the additional sediment is lower in the detachment-limited runs, resulting in less aggradation in response to the increased sediment flux. The addition of sediment in this model does not lead

to increased sediment cover on the bed, as bedrock and sediment are not differentiated in the model; rather increased sediment flux results in "deposition" of bedrock material that aggrades the channel. This channel aggradation in the model certainly indicates that vertical incision has stopped, allowing lateral erosion to become the primary erosive agent, even in models where $K_l/K_v$ ratio is low or in the total block erosion models. This predicted increase in lateral erosion (relative to vertical incision) during periods of aggradation occurs in some of the model runs, especially those with high $\alpha$ values. When the channel has

reached a new equilibrium following increased sediment flux, many model runs maintain wider valleys due to the higher slope and increased lateral erosion rates.

## 7.4   A potential test of the lateral erosion model

One of the goals of developing landscape evolution models is to develop and test hypotheses about how dynamics in natural systems work over spatial and temporal scales that are not readily observable. A challenge remains in how to test a newly de-

veloped numerical model with field data. In order to test simply how well this model captures the development of wide bedrock valleys, we would need a field location where channel curvature is identified as the primary mechanism for lateral erosion, for example, rivers in mudstone bedrock where particle detachment from the bank is from fluid stresses alone (Finnegan and Dietrich, 2011; Johnson and Finnegan, 2015). A field data set to test this lateral erosion model could conceivably be derived from experimental data, a well constrained "natural experiment" of wide bedrock valleys that developed over geologic time scales (Tucker, 2009), or from rapid valley widening associated with an extraordinary event. To our knowledge, experimental data sets that describe the effect of channel curvature on lateral bedrock erosion do not exist, nor have we identified an appropriate natural experiment to evaluate bedrock valley widening over geologic timescales.

Researchers have only recently started to study the mechanistic processes of lateral bedrock erosion (e.g. Fuller et al., 2016; Beer et al., 2017). The model presented here does not include all of the processes the community has identified as relevant to

lateral erosion; rather, we formulated the simplest reasonable model to test the hypothesis that stream power exerted on channel walls is a primary control on lateral bedrock erosion. We do not consider small scale processes, such as abrasion of channel walls by sediment, rather focus on reach-scale drivers of valley wall erosion. Because of the simplicity of our model and the grid size effects on valley widths produced by the model (Figure S8), this model is not currently suitable for predicting absolute timing or magnitude of lateral widening in natural systems.

In the broadest terms, the key prediction of this model that can be compared to field sites is the relationship of increasing valley width with drainage area. So far, no other landscape evolution models consider lateral bedrock erosion in a catchment scale model; therefore, most LEMs predict no relationship between valley width and drainage area. Our model does predict increasing valley width with drainage area (Figure 6, Figure S7) and a scaling relationship between width and drainage area that can be compared with data from natural systems. Figure 14 shows valley width vs. drainage area data from one undercutting-slump model run, with increased water flux for a period of 50 ky. The data shown in the figure is from six time slices when the model is in steady state and are time averaged over 2,500 model years. The time-averaged valley width data has some scatter and varies by $\sim$30 m for a given value of drainage area and covers a limited range of drainage areas. Log-binned averages of valley width show a scaling prediction that can be tested against field measurements of valley width and drainage area. The scaling relationship predicted by this model has a $K_v$ coefficient of 0.16$\pm$ 0.052 and a $c$ exponent of 0.46$\pm$0.027.

Several studies have shown a power law relationship between valley width and drainage area in natural systems (Brocard and Van der Beek, 2006; Snyder et al., 2003a; Tomkin et al., 2003). The power law equation describing the relationship between valley width and drainage area takes the generic form of $W = K_v A^c$, where $K_v$ is a widening factor and c is exponent that ranges in value from $\sim$0.3–0.75. Comparing model data with a field data set of valley width vs. drainage area could be used to determine how well this model of lateral erosion driven by channel curvature captures valley widening in natural systems. A key next step in this line of research is to analyze in detail the predicted scaling relationship between width vs. drainage area through a sensitivity analysis on grids much larger than those used in this paper so as to cover several orders of magnitude in drainage area and use this as a basic test of our model formulation.

## 8  Conclusions

The most important finding of this work is that a simple, physics-based theory for lateral bedrock channel migration, when combined with a landscape evolution model, produces wide bedrock valleys that scale with drainage area, as predicted in natural systems. So far, other landscape evolution models do not address lateral bedrock erosion and therefore predict no relationship between valley width and drainage area. Two end-member algorithms were presented that describe how lateral erosion occurs on the model grid: the total block erosion model requires that the entire volume of a node is laterally eroded before elevation is changed, while the undercutting-slump model requires that the node is laterally undercut and the overlying material is transported away as wash load. These two algorithms represent end-members of how lateral bedrock erosion can occur in natural systems and show significant differences in the patterns and timing of lateral erosion and the development of wide bedrock valleys. Significant bedrock valley widening, where valleys are several model cells wide only occur when

using the undercutting-slump model. Differences in the transient model response to changes in boundary conditions (e.g., first vertical incision followed by lateral erosion vs. simultaneous vertical and lateral erosion) can be used to determine the appropriate application of the end member models.

The model presented here also produces mobile channels in an eroding, rather than aggrading landscape. Channel mobility is a fundamental factor for developing and maintaining a bedrock valley that is several times wider than the channel it holds

(Tomkin et al., 2003). Increased channel mobility and wider flat-bottomed valleys under transport-limited conditions in the model, suggests that slowing vertical incision amplifies the effect of lateral erosion (Hancock and Anderson, 2002). However, this model lacks some important elements that prevent it from predicting absolute timing and magnitude of lateral erosion, specifically lateral erosion that is independent of grid size and separate treatment of bedrock and sediment. Our theory for the lateral erosion of bedrock channel walls and the numerical implementation of the theory in a catchment-scale landscape

evolution model is a significant first step towards understanding the factors that control the rates and spatial extent of wide bedrock valleys.

*Code availability.*  The lateral erosion models described in this text will be made available as a Landlab component in the fall of 2017.

*Competing interests.*  The authors declare that there are no competing interests present.

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

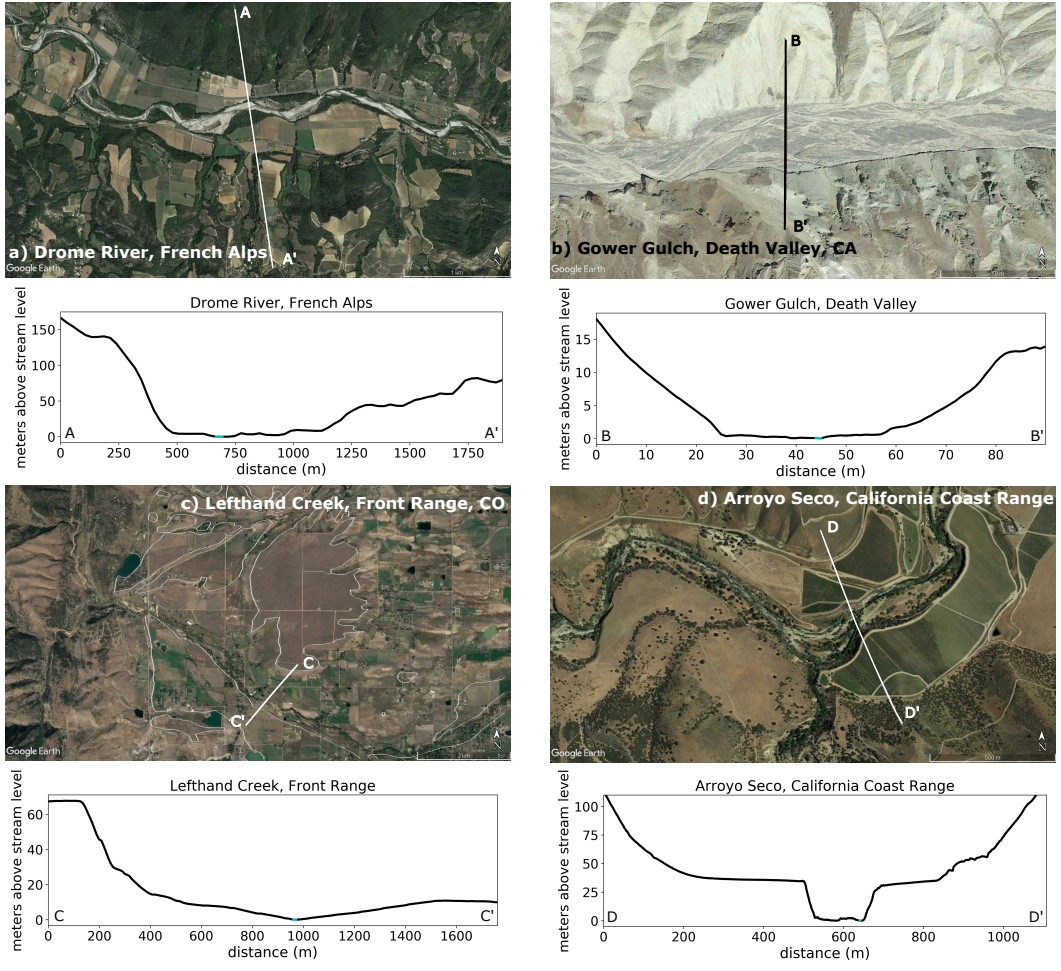

**Figure 1.** Field examples of wide bedrock valleys cut by lateral erosion. All cross sections are from north to south. a) The Drôme River in the French Alps is transport-limited and meandering in reaches that carve wide bedrock valleys. The bedrock valley at this location (44.69°N, 5.14°E) is 500 m wide and the channel is ∼45 m wide (indicated by light blue shade of cross section line). b) Gower Gulch (36.41°N, 116.83°W) in Death Valley, USA widened significantly in response to increased discharge from a stream diversion in the 1940's (Snyder and Kammer, 2008). The bedrock valley is 30 m wide and the channel braids are ∼2 m wide (indicated by light blue shade of cross section line). c) Lefthand Creek drains the Colorado Front Range (40.11°N, 105.25°W) and has undergone multiple cycles of lateral erosion that produced flights of strath terraces, outlined in white on the image. The cross section shows Table Mountain at ∼70 m above the current stream height on the north side of cross section and a lower terrace level at 10 m above current stream level on the south side of the cross section. d) Arroyo Seco in the California Coast Range (36.27°N, 121.33°W) carved a 600 m wide strath terrace during a period of lateral erosion that is 30 m above the current stream level. The current bedrock valley is 125 m wide and the channel is ∼15 m wide (indicated by light blue shade of cross section line). Images: Google Earth. Cross sections: NCALM/30m SRTM

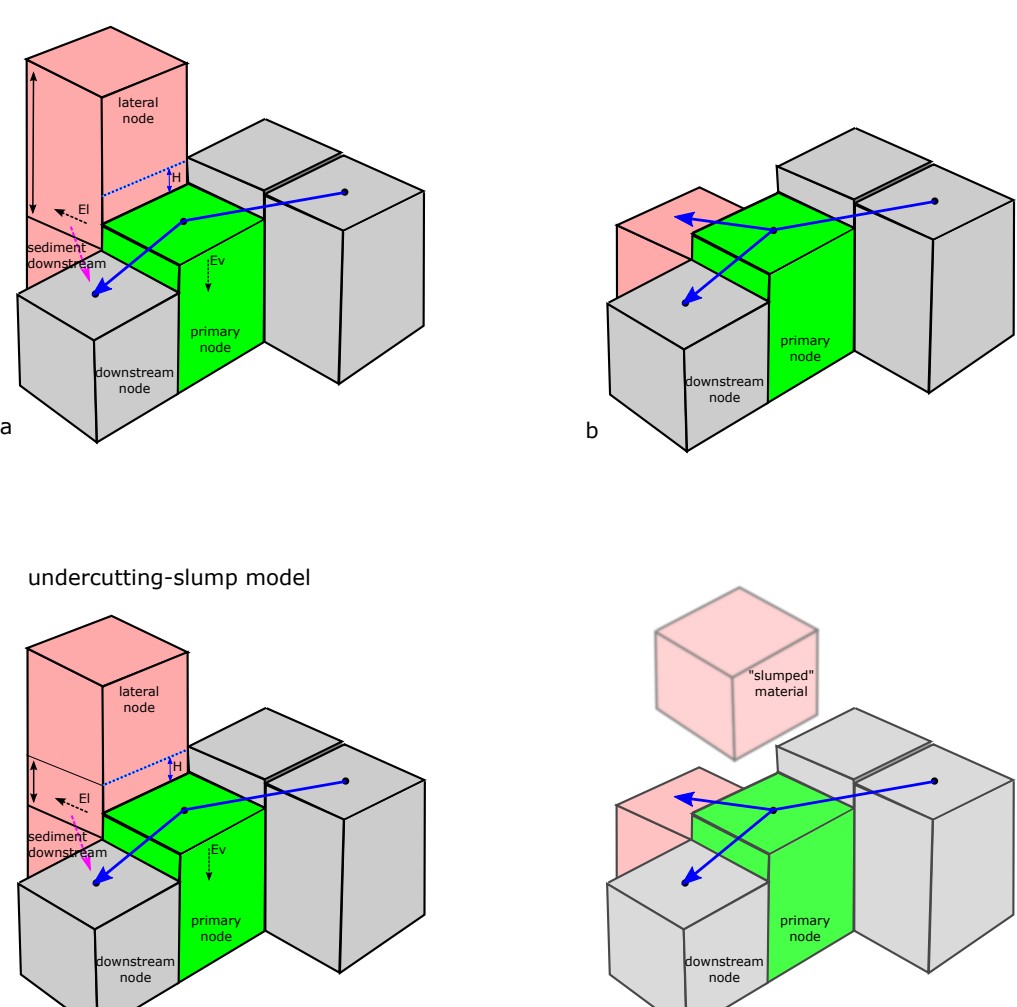

**Figure 2.** Conceptual illustration of model nodes showing the stream segments (in light blue) from the upstream node to the primary node (in green), to the downstream node. Vertical erosion ($E_v$) occurs at the primary node. The neighbor node (in pink) where lateral erosion ($E_l$) occurs is located on the outside bend of the stream segments. The height over which lateral erosion occurs, H, is shown by the dashed blue line. a) For the total block erosion model, the volume that must be laterally eroded before elevation is changed is $(Z_n - Z_d)dx^2$, the difference in elevation between the neighbor node and the downstream node (indicated with double-sided black arrow) times the surface area of the neighbor node. b) Elevation of the lateral node is changed after the entire block is eroded and flow can potentially be rerouted. c) In the undercutting-slump model, the volume that must be laterally eroded (representing bank undercutting) before elevation is changed is $(H - Z_d)dx^2$. $H - Z_d$ is the difference in elevation between the water surface height and the elevation of the downstream node, indicated with the double-sided black arrow. d) When the neighbor node has been undercut, elevation is changed, allowing water to be re-routed, while the slumped material is transported downstream as washload.

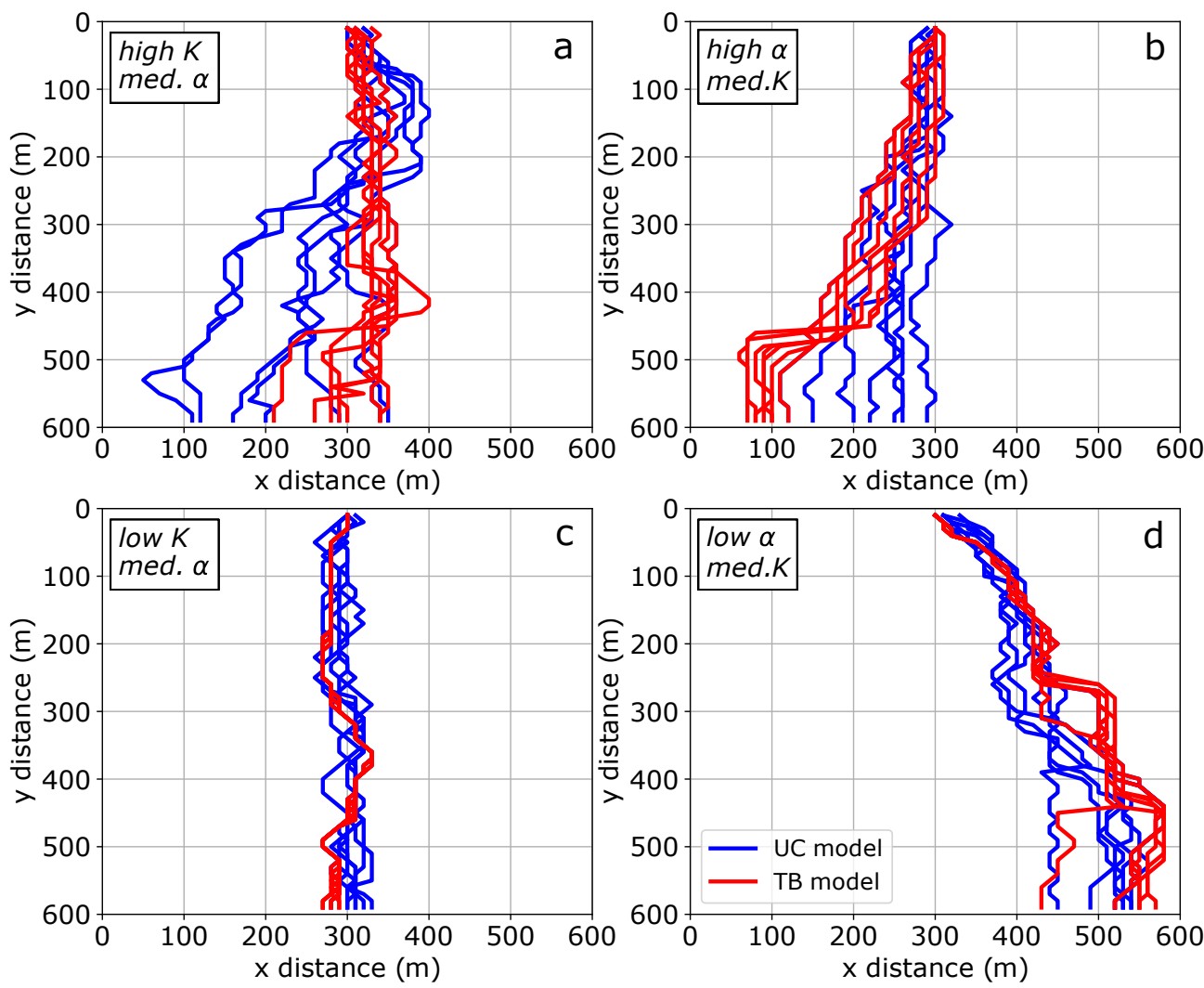

**Figure 3.** Channel positions over 200 ky with different values for bedrock erodibility and $\alpha$ in the undercutting-slump model (UC model, blue lines) and total block erosion model (TB model, red lines). a) high bedrock erodibility ($K = 2.5 \times 10^{-4}$), medium $\alpha$ value ($\alpha$=0.8). b) high $\alpha$ value, indicating low sediment transport ($\alpha$=2.0), medium bedrock erodibility ($K = 10^{-4}$). c) low bedrock erodibility ($K = 5 \times 10^{-5}$), medium $\alpha$ value ($\alpha$=0.8). d) low $\alpha$, indicating high sediment transport ($\alpha$=0.2), medium bedrock erodibility ($K = 10^{-4}$).

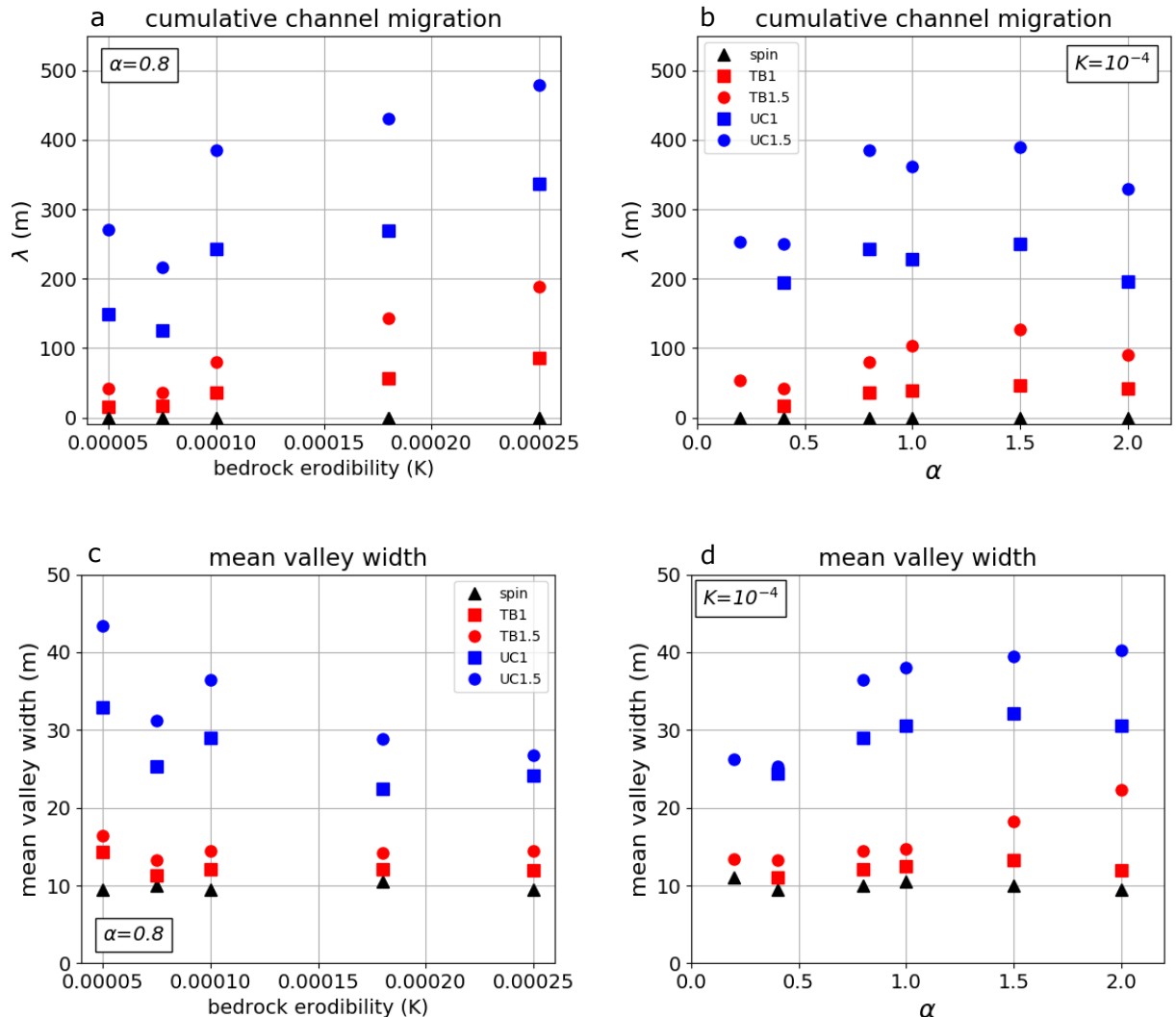

**Figure 4.** Cumulative channel-averaged migration (a,b) and mean valley width (c,d) averaged over 100 ky for spin up models with no lateral erosion (spin, black triangles), total block erosion models (TB, red markers) and undercutting-slump models (UC, blue markers) with $K_l/K_v = 1$ (square markers) and 1.5 (circle markers). a) Cumulative channel-averaged migration ($\lambda$) for model runs with $\alpha = 0.8$ plotted against bedrock erodibility, $K$. b) $\lambda$ for model runs with $K = 10^{-4}$ plotted against $\alpha$. Mean valley width averaged over 100 ky of the model runs. c) Mean valley width for model runs with $\alpha = 0.8$ plotted against bedrock erodibility, $K$. d) Mean valley width for model runs with $K = 10^{-4}$ plotted against $\alpha$.

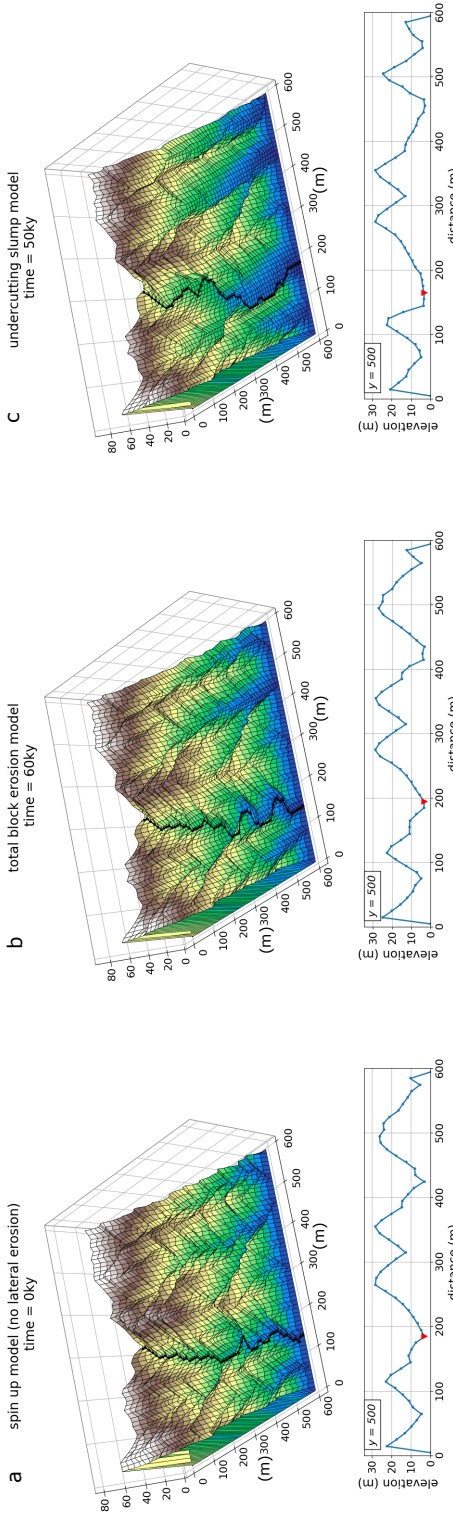

**Figure 5.** Model topography and cross sections at y=500 showing examples of valley widening. Black line indicates position of the main channel on the landscape. Red triangle shows position of the main channel in the cross section. a) Model with vertical incision only. b) Total block erosion model after 70 ky of lateral erosion. c)Undercutting-slump model after 50 ky of lateral erosion.

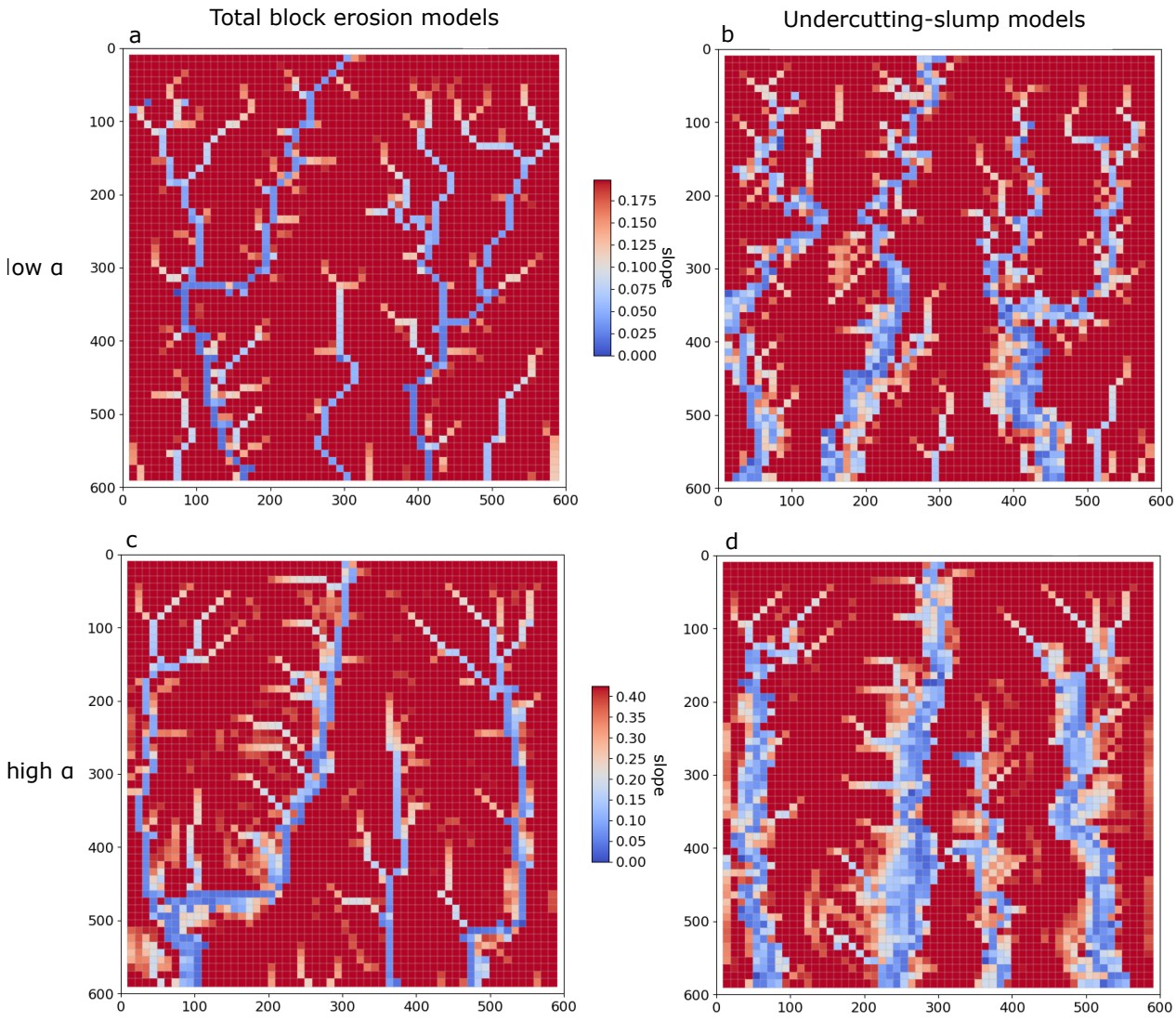

**Figure 6.** Slope maps showing fluvially carved valleys in total block erosion and undercutting-slump models with high and low values of $\alpha$. The white and blue areas in the maps that indicate slopes that are characteristic of fluvial channels, i.e. lower than the reference slope value (Equation 17). a. Total block erosion model, low $\alpha$ (detachment-limited) b. Undercutting-slump model, low $\alpha$ (detachment-limited) c. Total block erosion model, high $\alpha$ (transport-limited) d. Undercutting-slump model, high $\alpha$ (transport-limited)

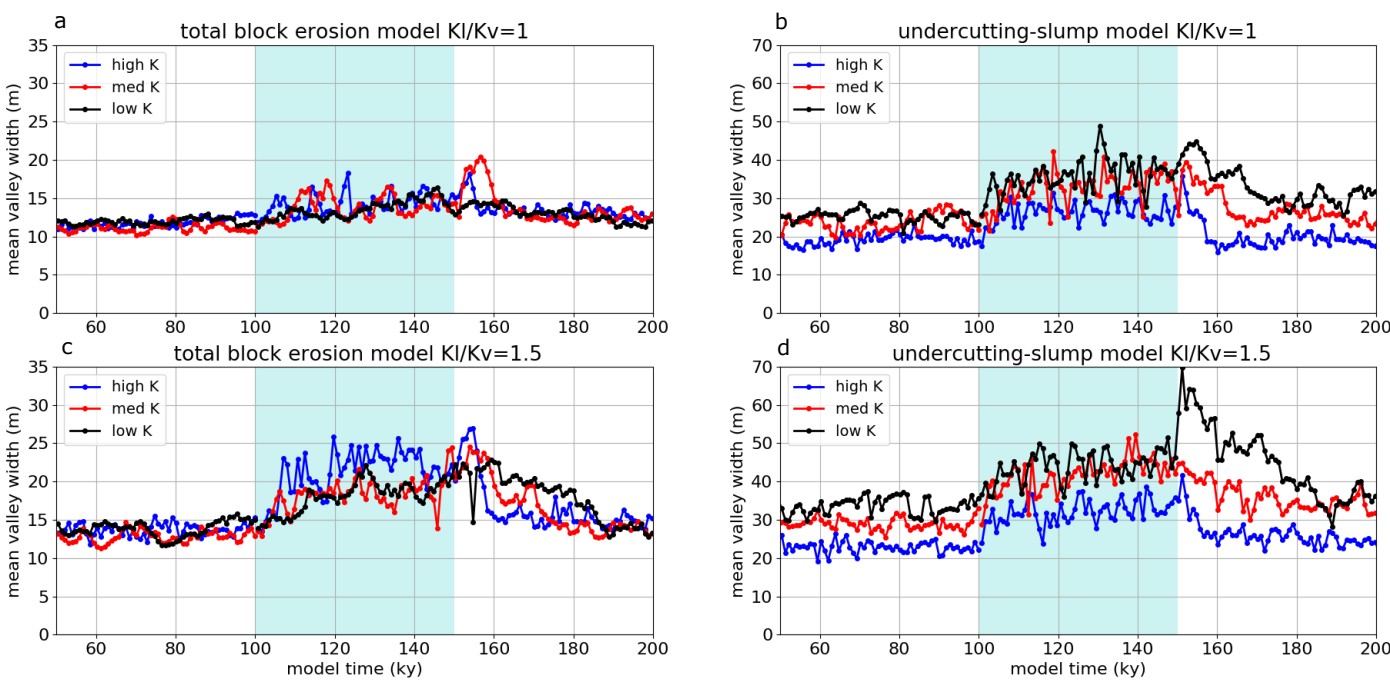

**Figure 7.** Valley width averaged over the model domain vs. model time for total block erosion and undercutting-slump models with $K_l/K_v =$ to 1 and 1.5. Increased water flux occurs from 100 ky to 150 ky, indicated by light blue shading.

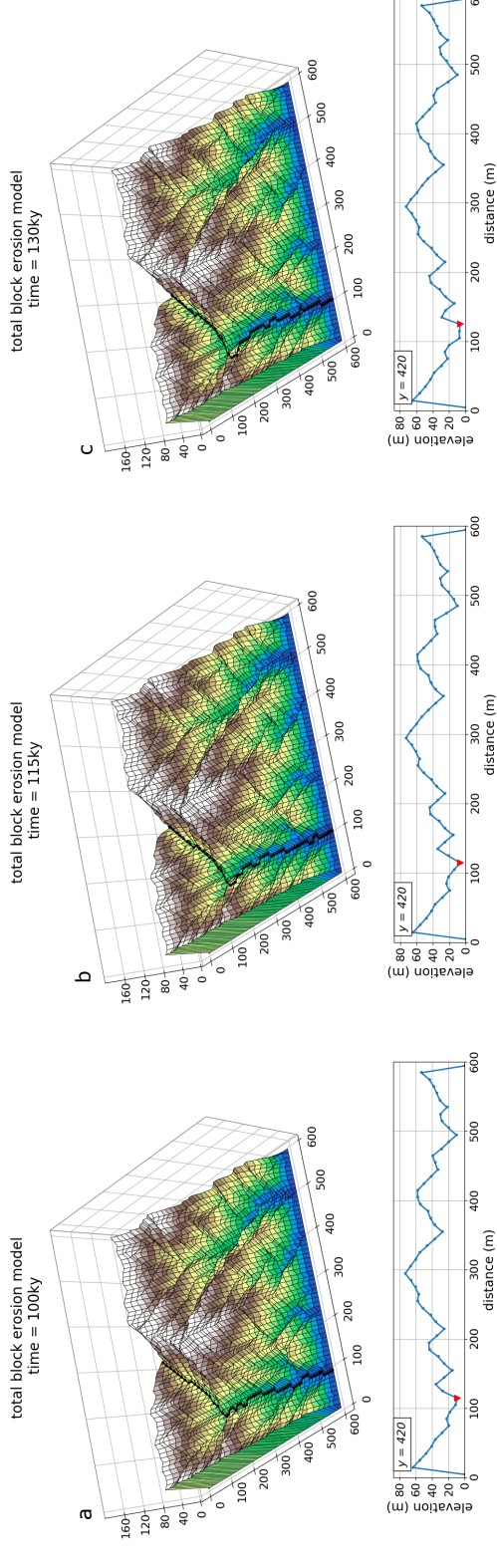

**Figure 8.** Surface topography and cross section at y=420 during period of increased water flux for the total block erosion models. Red triangle on cross sections indicates the channel position. a) Total block erosion model with low $K$ and $K_l/K_v = 1.0$ at 100 ky, before the increase in water flux. Note that this model looks similar to the spin up model runs with no lateral erosion. b) After 15 ky of increased water flux, the cross section shows vertical incision in the channel and increased relief between the channel and the hillslopes. c) At 30 ky after water flux increased, equilibrium is reached. Lateral erosion can begin and the valley is widens to 20 m at y=420.

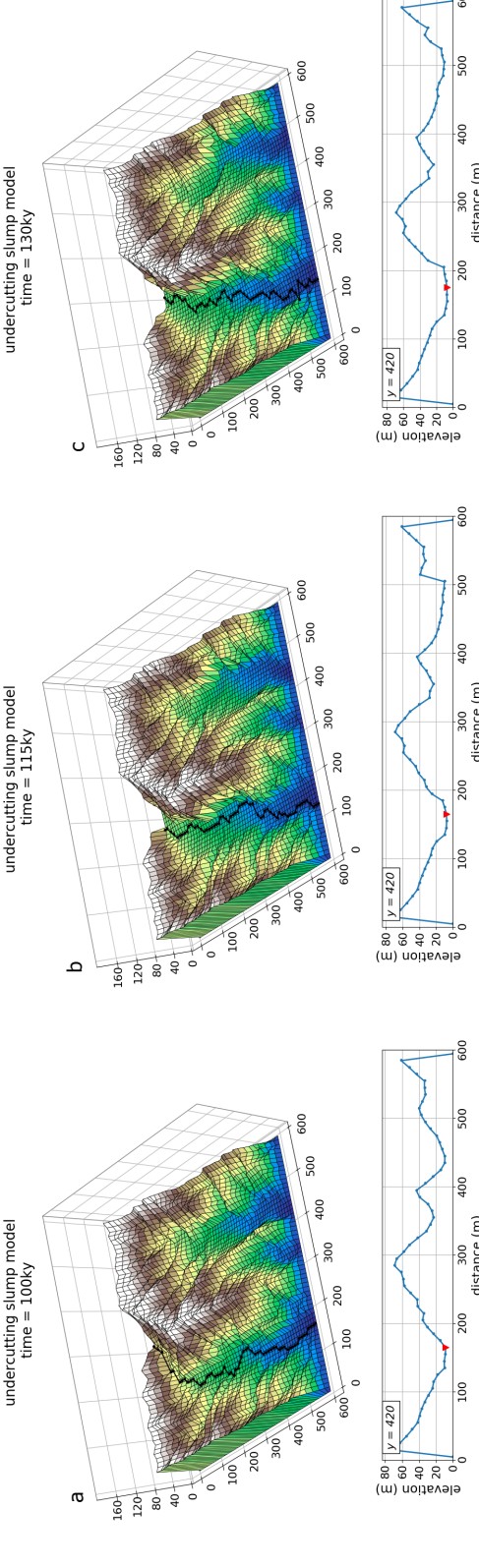

**Figure 9.** Surface topography and cross section at y=420 during period of increased water flux for the undercutting-slump models. Red triangle on cross sections indicates the channel position. a) Undercutting-slump model with low $K$ and $K_l/K_v = 1.5$ at 100 ky, before the increase in water flux. Valley is 30 m wide. b) After 15 ky of increased water flux, the channel has both vertically incised and laterally widened the valley to a width of 40 m. c) After 30 ky of increased water flux, the valley has a width of 60 m at y=420.

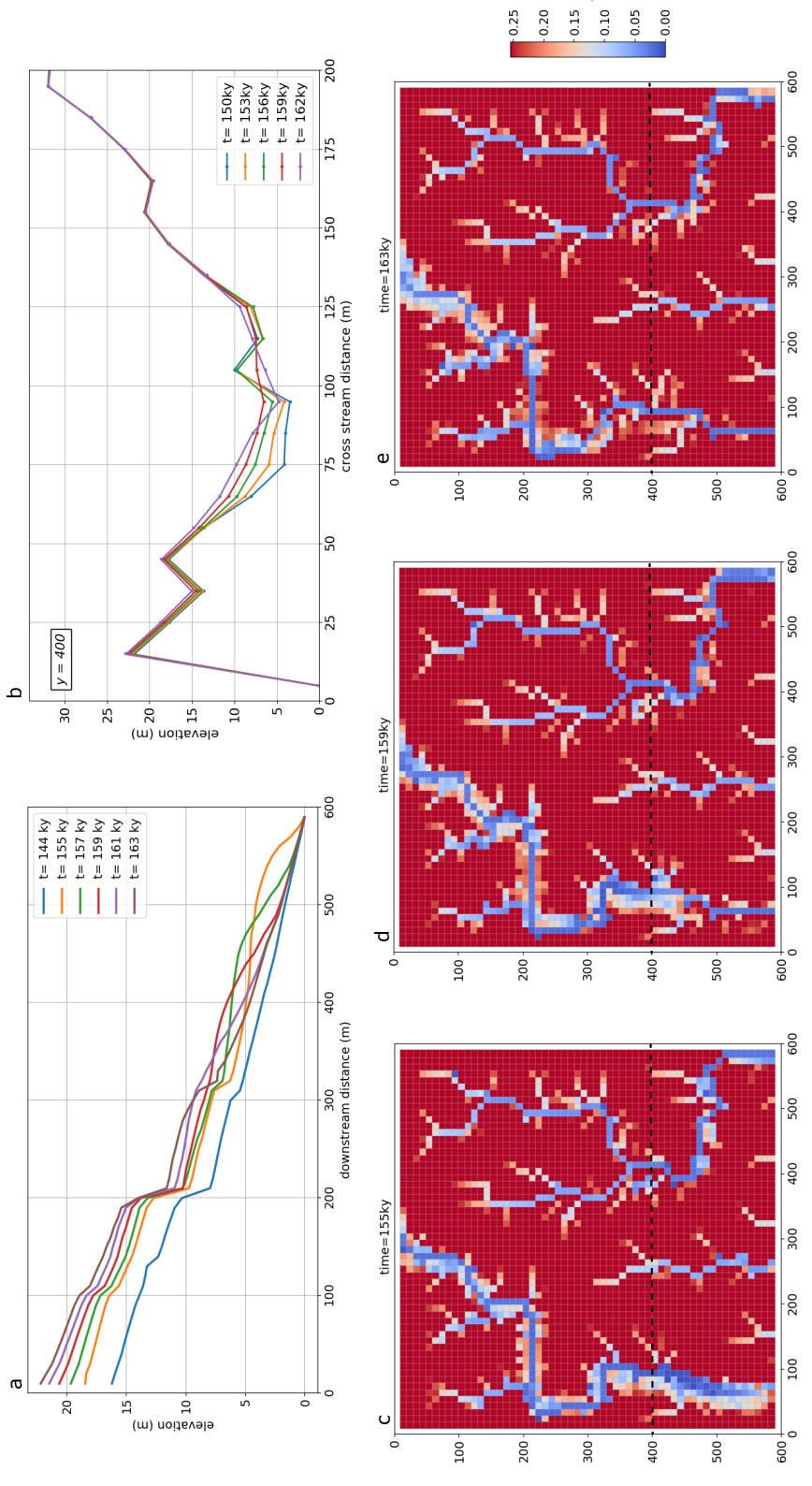

**Figure 10.** Longitudinal profile, cross sections, and slope maps from model run TB1.5, medium K after cession of increased water flux. a)Longitudinal channel profiles show uplift and aggradation, which produces a convexity that propagates upstream. b) Cross sections across the model domain at y=400 show channel aggradation and new lateral erosion of valley walls. c,d,e) Slope maps show valley narrowing following the passage of the knickpoint where y=400 (dashed line) at 155 ky, 159 ky, and 163 ky.

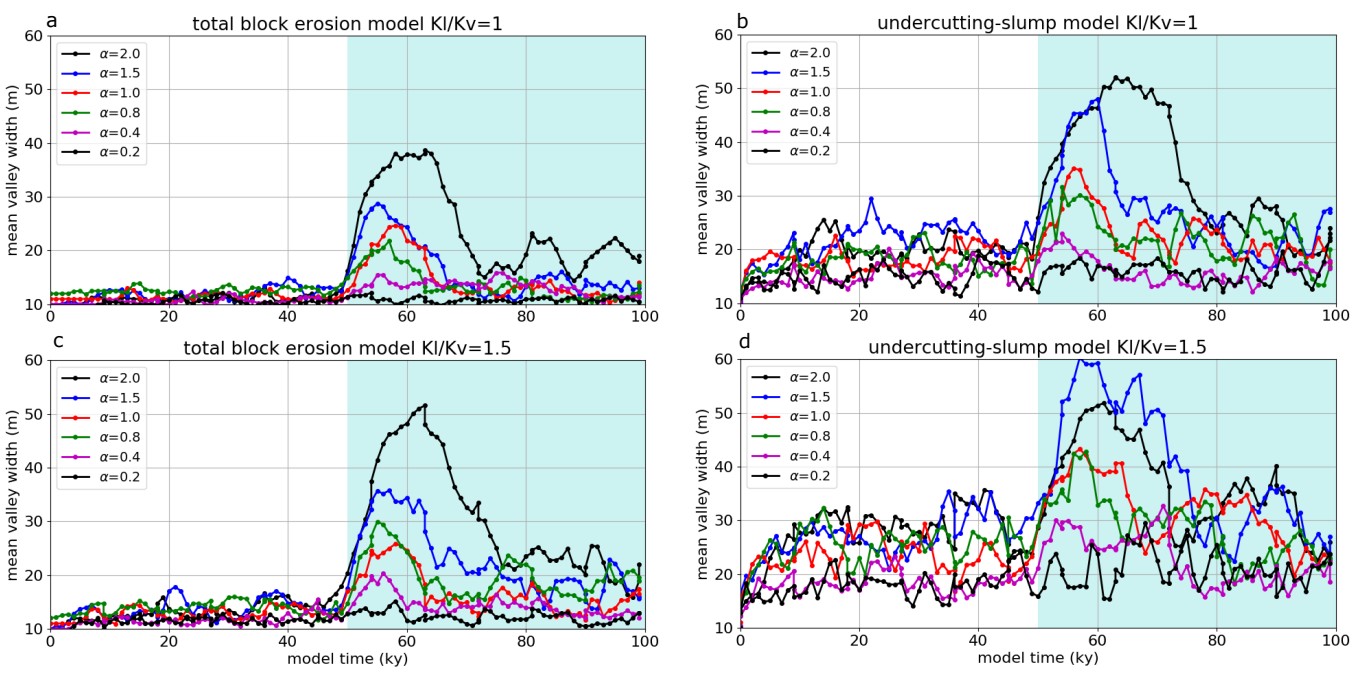

**Figure 11.** Mean valley width for the upper half of model domain over duration of additional sediment flux model run for total block erosion and undercutting-slump models with $K_l/K_v$ ratio of 1 and 1.5. Light blue shading indicates duration of increased sediment flux.

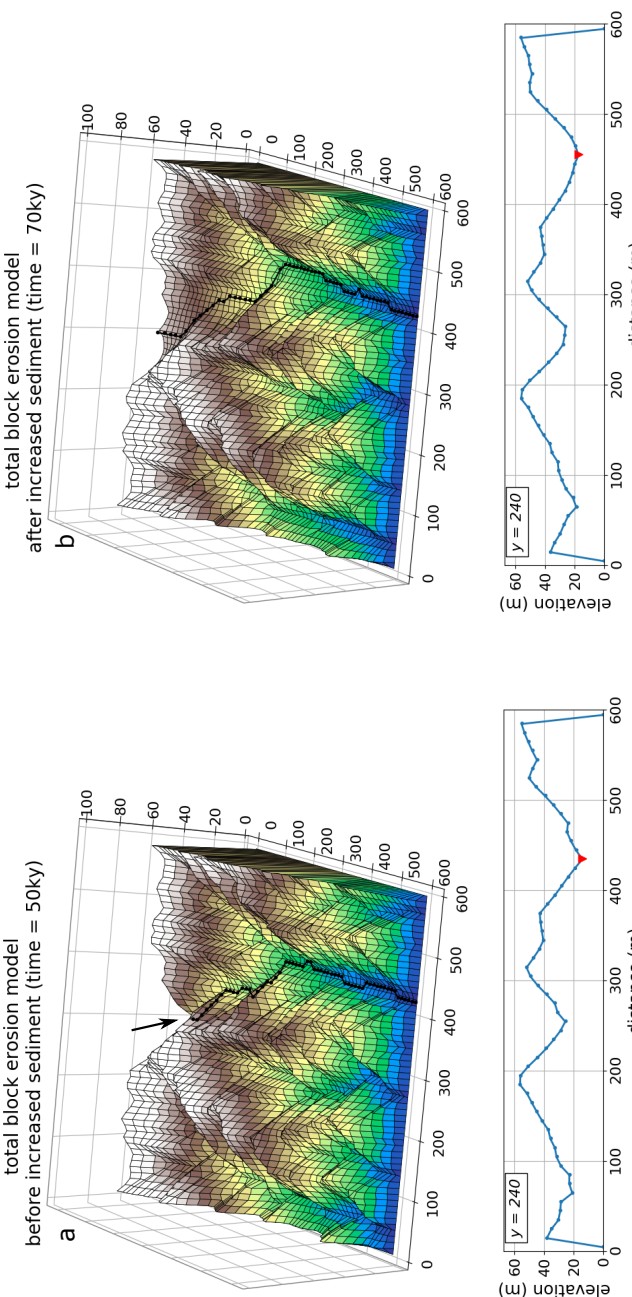

**Figure 12.** Model topography and cross sections at y=420 during period of increased sediment flux for the total block erosion model with $\alpha$=1.5 and $K_l/K_v$=1.5. Black line indicates position of the main channel on the landscape. Red triangle shows position of the main channel in the cross section. a) Before increased sediment flux is introduced at input point, indicated with the arrow. b) After 20 ky of increased sediment flux, the channel has aggraded by 4 m and has eroded the valley wall by 30 m.

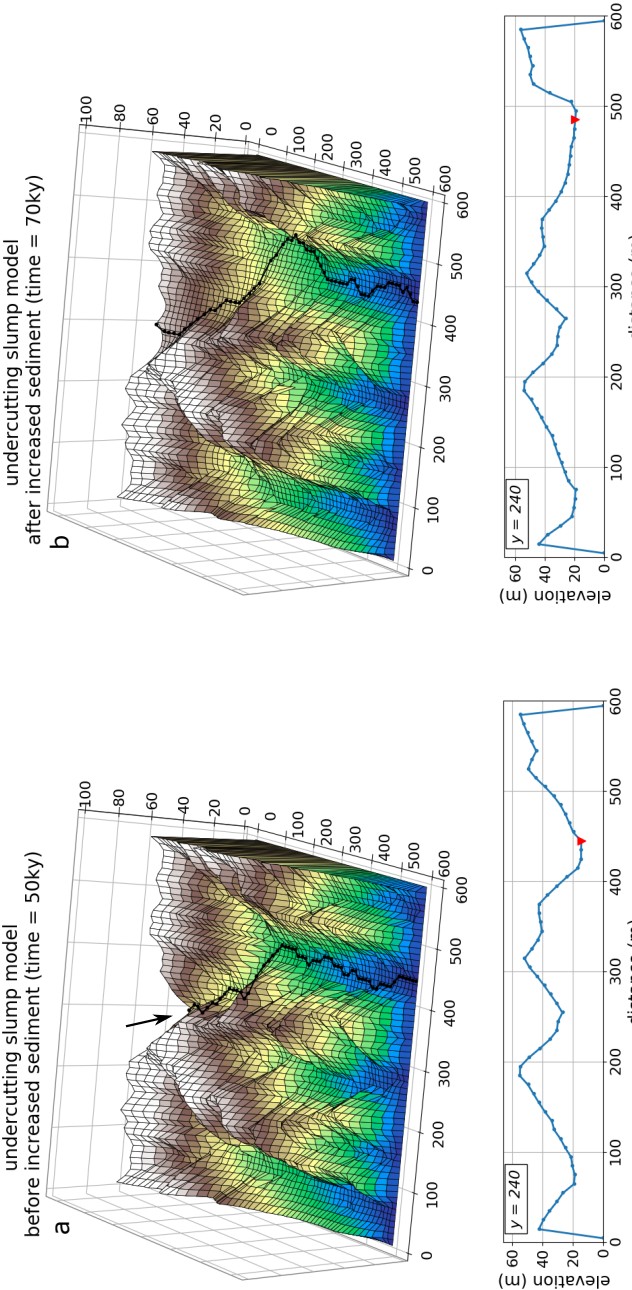

**Figure 13.** Model topography and cross sections at y=420 during period of increased sediment flux for the undercutting-slump model with $\alpha$=1.5 and $K_l/K_v$=1.5. Black line indicates position of the main channel on the landscape. Red triangle shows position of the main channel in the cross section. a) Before increased sediment flux is introduced at input point, indicated with the arrow. b) After 20 ky of increased sediment flux, the channel has aggraded by 5 m and has eroded the valley wall by 50 m.

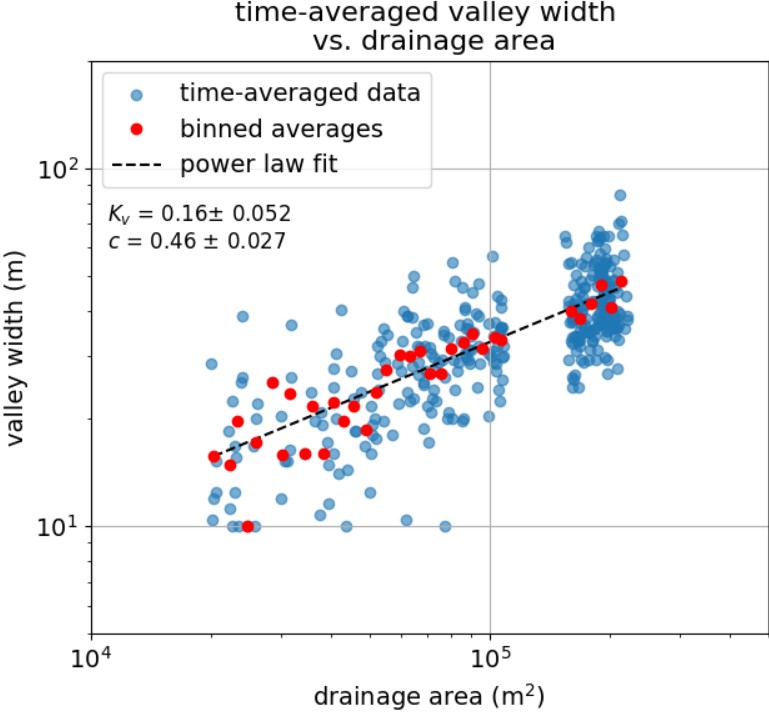

**Figure 14.** Valley width vs. drainage area for six time slices in the undercutting-slump, increased water flux model with dx=10, K=$10^{-4}$, and $\alpha$=0.8. All six time slices are from the model at steady state, with three time slices taken from the period of normal water flux and three time slices taken from the period of increased water flux. Each time slice represents data averaged over 2,500 years of model time, or 1.6% of the total length of the model run. Red dots show log-binned averages of valley width. The black line shows a least squares power law fit for the binned data. The $K_v$ coefficient has a value of 0.16 with a standard error of 0.052 and the $c$ exponent has a value of 0.46 with a standard error of 0.027.