# Peer review of "Developing and exploring a theory for the lateral erosion of bedrock channels for use in landscape evolution models"

_Earth Surface Dynamics, 2017_

## Referee Comment (RC1) · Anonymous Referee #1 · 23 Jun 2017

This paper describes a new approach to incorporating lateral channel erosion into landscape evolution models. This is clearly a worthwhile goal and I was excited to read this. That said, there are aspects of the model setup and motivation, as described below, that I think can be improved upon and which I think will lead to a paper with more impact.

1. Specifically, this paper uses a curvature based wall erosion law. While the authors don't expressly say they are modeling meandering, this is the implication of the choice of model. This makes sense as meanders are ubiquitous in bedrock channels and

the process is clearly important in many settings. The first numerical model of river meandering that I am aware of is Howard and Knutson (1984). Their first iteration of the model is one in which erosion scales inversely with the radius of curvature, which is basically the same as the model posed in equation 10. Howard and Knutson point out that such a model results in a channel that breaks down into 3 point bends with alternating positive and negative curvature. When applied to an existing meander bend, the bend can't be maintained. The ultimate conclusion of Howard and Knutson (1984) is that lateral channel motion can't be driven only by local curvature because such a model fails to produce realistic meander kinematics (down stream translation, cutoffs) as well as realistic meander forms. This is what leads to their downstream convolution approach, which in a simply way simulates the advection of the effects of upstream curvature downstream. Given that the setup of the model in the submitted MS is based on a centrifugal acceleration argument, and given that the morphologies of the channels produced in the model are reminiscent of the 3 point bends described by Howard and Knutson, it's not clear to me how this model represents a significant advance in understanding and modeling lateral erosion. Moreover, it's not clear how the river even changes from moving in one lateral direction to the other. Without passing information downstream, I would expect the bends to grow unstably.

What is novel, from my perspective, are the two different formulations of the wall erosion law. Why not, then, simply use the Howard and Knutson meandering model and then explore how the two different wall erosion formulations influence the emergent valley form? Given that field evidence that can discriminate between the two proposed lateral erosion processes should be straightforward to collect, I could see such an exercise leading to numerous field testable hypotheses.

2. While I like the exploratory aspect of this paper, I think it could benefit from either a sharply formulated research hypothesis or a field example or two that are targeted. As is, it's not clear how me can evaluate the performance of the model other than by simply noting that the river causes the valley walls to move. But I think we could do
better.
* * *
**ESurfD**

Interactive
comment

---

## Short Comment (SC1) · 23 Jun 2017

Dear Dr. Langston, dear Dr. Tucker,

I am excited to see this work come out and I am very much looking forward to seeing the paper published. Below, I am noting a few questions and comments that arose during my readthrough and I hope that some of these might be helpful in the revision process.

In summary, there are three major points that I thought could be clarified (and that I

address in more detail in the line comments below).

First, the treatment of sediment in the model is only mentioned relatively briefly at the very end of the discussion section. In order to better appreciate the modeling result, I suggest that it may be helpful to lay out the role of sediment in the model (and in particular how it "becomes" bedrock when deposited) early in the section about the numerical implementation.

Second, as marked below, I suggest to introduce the concept of channel mobility earlier and maybe comment on the difference between mobility within loose sediment (which, in my limited knowledge, much of the literature that uses the term 'channel mobility; is concerned with), and the mobility linked to movement within a pure bedrock landscape as it is defined in this paper. This distinction may also clarify the discussion about the importance of channel mobility on p 17. I comment more extensively on this below.

Third, I would suggest a more extensive discussion of the implications of using the stream power model without a treatment of sediment tools and cover. The lack of the cover effect is mentioned in a sentence on P 19 L11 but in the list of limitation that follows and in the following discussion I did not find a clear mention about the possible limitations that the absence of tools may introduce. An appreciation of the implications of the stream-power model are particularly important before model results are compared with other studies such as those by Hartshorn et al., 2001 and Fuller et al., 2016. The effects of tools and cover seem an integral part of the interpretations of the observations that were made by these cited studies (and in other studies such as Hancock and Anderson 2002 etc.). Therefore, without a discussion about the role of tools and cover, the link between the present stream power model and other studies (that is made early in the manuscript) appears problematic. Moreover, tools and cover may affect lateral and vertical erosion in different ways. For example, an increase of lateral erosion rates because of a change in the amount of sediment that is deflected toward a channel wall (Fuller et al., 2016), may or may not be accompanied by changes in vertical erosion rates. Therefore, the response of the system to a change in water

or sediment flux may be more complex than predicted by the model. In short, I can imagine that a more expansive discussion of this limit may be useful. In particular, these complications should probably be mentioned before the model is compared to results from other studies.

Line comments

P.2 L.7 – Nitpicky comment: it may or may not be worth noting that cessation of incision and cutting of straths has also been observed in harder lithologies such as quartzites or granites – These straths are narrow and they don't contradict the statement that sizes of strath terraces and rates of strath cutting seem strongly linked to bedrock strengths, but the way the sentence is phrased now, it could be understood that straths never form in stronger lithologies:

See for example:

- Pratt-Sitaula, B., D. W. Burbank, A. Heimsath, and T. Ojha (2004), Landscape disequilibrium on 1000–10,000 year scales Marsyandi River, Nepal, central Himalaya, Geomorphology, 58(1–4), 223-241

Or for an example of rapid widening in unweathered granite

- Anton, L., A. E. Mather, M. Stokes, A. Munoz-Martin, and G. De Vicente (2015), Exceptional river gorge formation from unexceptional floods, Nature Communications, 6.

Somewhere in the setup and introduction (for example somewhere in the paragraph starting P.2 L.11), it might be worth mentioning published models that consider the control of valley wall-height on widening rates e.g.:

- Malatesta, L. C., J. P. Prancevic, and J.-P. Avouac (2016), Autogenic entrenchment patterns and terraces due to coupling with lateral erosion in incising alluvial channels, J. Geophys. Res

P.5 L.7. Setting W=kwQ1/2 may be common knowledge but I wonder if it is worth citing the original works that this scaling is based on.

P.5 L.16. I like the idea of looking at the centripetal force. I was wondering at this stage what happens to straight segments of rivers. The way straight segments are treated is layed out later: They are treated as having a range of radii of curvature. However, there is also evidence for erosion in perfectly straight channels (e.g. Fuller et al., 2016). Maybe a quick note of the limits and possible alternatives of this model formulation could be made here or in the discussion? At the latest, this should probably be mentioned when the model results are compared to results from Fuller et al., 2016.

P.6 L. 12 It was a little unclear to me what the word "which" refers to here – either equation 12 or the variable Kl.

P.6 L. 16ff As far as I understand, the result that Kl/Kv scales with Q1/2 or R1/2 is derived within the framework of stream power models. I feel that the link to the field studies is a little misleading in this context. The changes in the ratio of lateral to vertical erosion rates between high and low flows measured in the Liwu river (Hartshorn et al., 2001) have been interpreted to be due to changes in the distribution of sediment in the flow (interpretation by Hartshorn et al., 2001) or to variable shielding of the bed (Turowski et al., 2008, ESPL). Because the importance of tools and cover for lateral and vertical erosion are not considered in the stream power model presented here, whereas the end result may be similar between model and field study (high discharge = high lateral erosion), the processes are likely different. Therefore, a comparison between field study and model without a more extensive discussion might be misunderstood. In turn, the increased sinuosity with storminess found by Stark et al., 2010 was interpreted by the authors as an expression of the importance for hillslope mass wasting in controlling lateral erosion. This interpretation may or may not be true, but it again, complicates the link of the model to this field examples. Later (P18 L10), there is a similar issue with the comparison of the model to the study by Fuller et al., 2016. – see comment further down.

Generally, I think it is valuable to discuss whether the model behavior is observed in nature but I think it necessitates a more detailed discussion of the limits of the stream power model before the comparison can be made.

P.7 L.15 I wonder, if, before detailing the way lateral and vertical erosion is calculated, it would be worth detailing one entire timestep and the order in which equations are solved. In particular, at this point of the manuscript, I was unsure how streams migrate. As far as I understand, at the beginning of a timestep, flow is routed across a topography via a D8 algorithm, then the lateral and vertical erosion is calculated, the topography updated and the flow rerouted through the landscape. Could this maybe be briefly laid out step by step? Or as a flow chart figure?

Even more importantly, at this point in the manuscript (and up to the very last paragraph of the discussion) it was unclear to me how sediment was treated. This is important to appreciate many of the features of the model (channel migration and channel mobility in particular) Questions that would be good to clarify are: Is deposited material added to the topography of a cell? What happens to a cell that is partly sediment and partly bedrock? Is the difference in erodibility considered or does deposited sediment "become" bedrock? Detailing the treatment of sediment could probably be intertwined with the walkthrough of one model timestep.

Section 3.1: I was a little confused by the (as I understood it) differentiation between resistant lithologies for which the slumped material has to be eroded (therefore bank height is important) and weak lithologies for which all material is swept away after a slump happens (therefore bank erosion is not important). The way it is described in the text is that the material is "transported away". This formulation seems ambiguous to me. Is the material added to the sediment flux Qs or does the material "vanish" in the model. I believe the later is meant. If the material "vanishes", I was wondering where such a model would be applicable in nature. I had thought that even for loose, non-resistant sediment, there should be a bank height control and that transport capacity is important. As in aside, the importance of wall height, even in loose sediment seems to

be implied by the later mentioned study by Bufe et al., 2016. Here, we demonstrated that in loose sediment, the width of valleys across an uplift is a function of the uplift rate (controlling the growth of valley walls) and the channel mobility (controlling the frequency at which a river revisits a given point in the valley). The area of valley that is cut across a fold reaches some equilibrium value that can be maintained and that is flanked by steep, high walls. One interpretation of this finding is that the equilibrium valley area that is actively "maintained" by the river is limited by the bank heights that the stream has to rework as it moves across the valley. For example, when a river moves from point x to point y and back to x, the bank height that has grown at point x during the time the river traveled across the valley depends on the channel mobility. The slower the migration rate of the river, the higher the walls that it encounters at x once the river returns. The observed equilibrium valley area therefore seems to imply that the wall-height and the capacity of the river to transport the material of the walls is important even in loose sediment.

Section 4.1.1: It may be clearer to introduce the concept of channel mobility and the way it is defined in this study either earlier in the paper or at the beginning of this section. At the moment, the channel mobility is defined at the beginning of the second paragraph of the section. As far as I know, the term channel mobility has mostly been used in the framework of alluvial rivers. I am guessing that the processes that limit the mobility of channels in loose sediment and in cohesive bedrock are partly different. Therefore, it would be helpful to clearly make a link here to the treatment of deposition of sediment in the model and to emphasize that in this model, any lateral movement involves bedrock erosion.

P.9 L.15 Because the treatment of deposition was not clear to me, at this point, I found it hard to wrap my head around how $\alpha$ affects channel mobility. Is it purely the effect of sediment deposition creating topography and therefore causing channel to switch more frequently? Or does sediment deposition also create an alluvial surface across which channel can migrate rapidly? After reading the end of the manuscript it became

clearer that sediment, when deposited, "becomes" bedrock. Therefore, I am guessing the reason that increased sediment flux creates more mobile channels is only because sediment deposition creates "topography" that moves channels? I could imagine, that such questions could be avoided if the treatment of sediment is explained earlier.

P.10 L. 6. I had to read the sentence a few times until I understood. Possibly rephrase to write it as "That is, the maximum possible extent of x positions occupied by the channel is equal to lambda, but the actual [. . .]".

P.10 L. 26: Here, the increase of channel mobility with alpha is discussed. I was just wondering if the decrease of channel mobility at alpha=2.0 is relevant to mention? At least that it may constitute an outlier?

P.11 L.18. The expression "The [. . .] models take [. . .] 10 ky to respond to lateral erosion" was a little unclear to me. What constitute a "response to lateral erosion"? Is it the time lag between the onset of the lateral erosion after the spin-up and the corresponding appearance of a signature in the topography? In which case, is there some characteristic that was used to define when the topography was thought to show a response? I am sorry if I misunderstood this. . .

P. 12 L. 20-21 Typos in this section "runs are easily", "processes due to their low relief", "has recently been shaped"

P. 12 L. 20-21: Again, I am sorry if I am misunderstanding but I would be interested in some expansion of the thoughts behind why the widest valleys occur in models with low channel mobility. I am unsure what is meant by "hillslope processes" in this context. I don't think any hillslope processes have been introduced in the model or in the introduction and theory sections of the paper. This word makes me think of landslides, hillslope creep or gullying – none of these processes are in the model I believe and I am not sure I understand what is meant here.

P. 13 L. 20-22: The sequence of incision followed by lateral erosion in the TB models

versus simultaneous incision and lateral erosion in the UC models would be nice to see in a figure. It is not clear from Fig. 6. In Figure 8, one panel is missing for the TB and the UC models respectively (the panel for 120 ky, just before lateral erosion starts in the TB model) to appreciate that sequence. Maybe it is possible to add one more panel and to refer to Fig. 8 at this point already? The same added panels would be nice to have at the end of this section (P. 14 L.12-19).

P14 L24-26: I did not understand the explanation for why there is no lag time. Is the argument: 1) Lateral erosion rate is increased more than incision rate and 2) the bank height is not important in the UC models -> Therefore any increase in lateral erosion rate translates directly into a widening rate?

P14 L24-26 typo: "two times" or "two time steps"?

P.16 L. 12. I might be missing something obvious, in which case ignore the comment, but I am unsure of how to distinguish valley width formed via valley infilling or via lateral erosion from the curves of Fig. 11c. . .

P17 L19-26 I am glad to see a discussion about channel mobility and I was thinking about whether this discussion could benefit from a few clarifications and some restructuring. Therefore, I briefly come back to the definition of channel mobility that I mentioned above. The cited studies (Wickert et al., 2013 and Bufe et al., 2016) define channel mobility in the context of the fluvial reworking of an aggrading (or steady) alluvial surface. In my mind, this "alluvial channel mobility" is not exactly equivalent to the rate at which actively uplifting valley walls are eroded – and therefore to the definition of channel mobility in this study. I totally agree that one can define a channel mobility in the context of the cumulative migration metric that was used in this paper. Such a metric can be calculated and defined independently of any regard for whether rivers are migrating across a valley, across an alluvial fan, or whether rivers are eroding valley walls. However, I am unsure that there is an a-priori reason to directly, and without further explanation, use the same terminology for migration rate of alluvial channels within

an alluvial valley and the lateral erosion rate of valley walls, and therefore valley width. There are of course reasons to make the link between channel mobility and valley wall erosion. Hancock and Anderson 2002 hypothesized the importance of the frequency of the contacts between the river and valley walls. Malatesta et al, 2016, and this study, demonstrate a potential importance of valley wall height. As mentioned above, if wall height is important, then channel migration rates across the active valley and the uplift rate should control valley width. This was demonstrated by Bufe et al., 2016 at least for loosely consolidated valley walls. In short, there seem to be links between the "classic, alluvial" channel mobility and the lateral erosion of valley walls but I think the link merits expanding upon before the term "channel mobility" is interchangeably in both contexts.

Section 5.2: Maybe here, the comparison with Hartshorn et al., 2001 and Stark et al., 2010 can be made. However, the limits of not considering tools and cover in the models might have to be discussed in more detail before that.

P.18 L.11. This paragraph discusses the model setup that links lateral erosion to channel curvature. I think it is worth noting in this context that Fuller et al., 2016 documented lateral erosion in a straight channel. As noted by the authors, the deflection of sediment (tools) toward the walls seemed to control lateral erosion in these experiments, thereby documenting the importance of tools. Because the model (and this paragraph in the paper) discusses the importance of channel curvature and because the significance of the absence of tools in the model has not been discussed, maybe the comparison with the Fuller et al., 2016 study can be moved and/or expanded upon?

P18 L 14 Typo: "has come into equilibrium"?

P18 L23. Maybe worth discussing - Anton, L., A. E. Mather, M. Stokes, A. Munoz-Martin, and G. De Vicente (2015), Exceptional river gorge formation from unexceptional floods, Nature Communications, 6. This study documents knickpoint retreat and subsequent widening (maybe comparable to the TB models?) in hard bedrock.

P18 L 33 Typo: "stream power to carry"?

P19 L1-2 As mentioned before, it would be worth to discuss the treatment of sediment in more detail, and earlier in my opinion.

Figure comments

Figs. 2-3: It might help to spell out the abbreviations "UC", "TB", and "spin" in the figure legend. There should be enough space. If not, it would be useful to have the definitions in the caption. It could also be helpful to add the other variable (K or alpha) to the boxes on top of the figures. For example "high K, moderate alpha", "high alpha, moderate K" etc.

Figure 4: The term "spinup model" could be used in panel 'a' to more easily relate these models to the previous figure. Also, I would tend to try not having text and grid overlap. Finally, the axis labels for x and y axes may be useful

Fig. 5; the c-axis (slope legend) needs a label and maybe x and y axes could use labels, even though it is fairly obvious what they are

Fig. 6. The last sentence in the caption reads as if there was only waterflux from 100-150 ky – I am guessing "increased drainage area" or "increased waterflux" is meant?

Fig. 9: Maybe you can add the type of model to the title of panels a and b – as well as give the actual number of K instead (or in addition to) "low" and "medium" K.

Fig. 11: Should the y axis label not be "difference in valley width"?

I hope these comments are clear enough and may be of some use. I am looking forward to seeing the study published!

Sincerely,

Aaron Bufe

Please also note the supplement to this comment:
http://www.earth-surf-dynam-discuss.net/esurf-2017-28/esurf-2017-28-SC1-
supplement.pdf

**ESurfD**

---

## Referee Comment (RC2) · D. Lague (Referee) · 6 Jul 2017

This article aims at developing and implementing a model of lateral mobility of rivers in long-term landscape evolution model of mountain ranges. This is timely needed as the lateral mobility of river is now known to play a significant contribution in landscape reshaping, and as most current numerical models of landscape evolution predict valley bottom that are simply 1 pixel wide and fixed in time.

The article introduces two aspects:

-A theoretical formulation in which lateral channel mobility is assumed to be proportional to the centripetal energy expenditure of water.

-A numerical implementation on a fixed regular grid with a description of the solutions to overcome the limits in describing the migration of a vertical front (valley side) on a horizontal grid.

Then the model is used to explore some basic simulations (steady configuration, transient dynamics) to "see" how it looks. The paper does not try to address a specific scientific question, but more a technical/methodological issue, which is fine with me. The challenge is important, given that if one could have a realistic model for channel mobility in large scale/long-term landscape evolution models, one could properly addresses issues such as drainage capture, valley bottom formation, drainage network advection, fold bevelling etc...Overall the MS is well written and clear to follow.

The problem is that I find that the numerical implementation have several flaws which prevent me from trusting the model outcome at this stage. Numerical modellers all know that it is very easy to create landscapes that look ok if you have some large degree of freedom in choosing your model parameters (erodibility, runoff, channel width coefficient etc...). Here, the modelling results look ok, as the model is tuned to looks right, but that does not mean that the dynamics and timing are relevant to natural systems, which is what we ultimately expect from a landscape evolution model. And because there's no real attempt to validate model predictions against quantified observables, it is very difficult, given some of the flaw in the implementation, to infer reliable results pertaining to the dynamics of natural mountain valleys.

I've made a lot of comments in order to help the authors improve their model, and I really hope that they will sort out the issues I raise or demonstrate that they are not that important, as it is indeed important to tackle the issue of channel mobility in landscape evolution models.

GENERAL ISSUES ON THE MODEL:

On the model description there is a fundamental inconsistency that needs to be sorted out: it is the difference between the local channel width and the pixel size. The problem can be treated in 2 ways on a fixed grid:

- Hydrodynamic models (either operating on reach or landscape scale, e.g., CAESAR (Coulthard et al., 2013), EROS (Davy and Lague, 2009; Croissant et al., in press)): the pixel size is significantly smaller than the channel width (on which you actually resolve "true bank erosion") and for which channel width is a self-emerging property.

-Non-hydrodynamic models: the pixel size is ALWAYS larger than the channel width and channel width is imposed by an external equation. In this case your channel may actually sit anywhere in the pixel, and may not, for instance be in contact with the neighbouring pixel (in which case, I don't see why it would erode laterally).

The model presented here is a non-hydrodynamic model aiming at including a "channel mobility" component. This is a great idea, and indeed barely addressed by landscape evolution models. But it is not strictly speaking a "bank erosion model" as it does not resolve 2D flow hydrodynamics. Yet there are many instances in the paper, where the model has some kind of schizophrenic behavior between the two types of models:

- First it uses a relatively small pixel size (10 m), which assumes practically that the channel width must never be larger than 10 m. Unfortunately, this condition is not verified all the time (unless I've missed something in the calculations): the basic model uses a drainage area of 20000 $m^2$, which coupled with a runoff of 36 mm/hr, and kw=10, gives Wmin = 4.5 m. However, multiplying the drainage area to 160000$m^2$ (section 4.2.1) violates this assumption from the inlet of the model Wmin = 12.65 m. At this point flow should be partitioned over 2 pixels to correctly resolve the equations. I don't know how this bias affect the model predictions, and how such a model could be uspcaled to larger catchments where channel width would be several pixel wide (here we're dealing with small catchments of $\sim$km$^2$ size).

- There is no real notion of "bank" in the model given that the channel is defined at subgrid, but rather some kind of "valley side". This makes it difficult to directly relate lateral erosion "end members" (fig.1 section 3.1) to actual physical processes. These are more numerical tricks to resolve vertical feature horizontal migration on fixed horizontal grid, but whose relevance to natural processes is quite debatable. They introduce artificial thresholds in model dynamics whose consequences are not explored thoroughly.

- The model implementation assumes that the channel is always in contact with the neighbour node (there is systematically lateral erosion), which contradicts the underlying assumption that channel width is smaller than the pixel size.

- The model does not account for lateral deposition which is an important driver of channel migration (but that's not the most critical point)

On top of this, there is an important limitation in the "undercutting- slump" model in assuming that flow depth only depend on discharge (eq. 30) while it must depend on slope (and width, but given that it is fixed by discharge in the model, there's no way to do better).

Hence I see at least two components missing in the model: 1 : A proper way to deal with cases in which the channel width becomes larger than the pixel size (as predicted by kwQˆ0.5): either you increase the pixel size (but this also increases the "numerical" threshold for channel migration), or you introduce some kind of flow partitioning/simplified 2D hydrodynamics (but then we're very close to existing models like CAESAR or EROS). I know width is lumped in the model through kw, but either you assume your channel width is never larger than 10 m (that's quite a limiting factor), or you have to partition the flow over several pixels. 2 : Adding a way to either explicitly or implicitly account for the sub-pixel position of the channel. For instance a kind of likelihood of bank erosion (which is a function of the ratio of channel width to pixel size) with an asymmetric probability related to alongstream curvature.

I also note that, even if it is not common practice in the litterature of landscape evolution models (it should), it is important for any numerical model implementation, to

demonstrate that the model results do not systematically depend on grid size (within limits) and time-step, or to acknowledge this dependency and demonstrate how it impact results. Also, I would also like to see the model evolve from an initial condition with the lateral erosion "on", and not activated only when the landscape and drainage is already organized: if a model works, it works all the time, and actually exploring drainage development on a plateau could tell us whether you generate realistic patterns or not.

Other comments Title: it is currently slightly misleading as there is no real evaluation nor comparison of the model prediction with actual results, and the link with the mechanics of bedrock channel bank erosion are extremely tenuous or not really clear. Something like: "Implementing lateral mobility of channels in landscape evolution" models would more represent the actual content of the paper.

Missing literature:

The CAESAR numerical model, although dedicated to reach scale (but there are also a few catchment scale simulations) should absolutely be cited and studied as it is relevant for the bank erosion law and the use of curvature. For instance, Coulthard et al., 2013, ESPL, Integrating the LISFLOOD-FP 2D hydrodynamic model with the CAESAR model: implications for modelling landscape evolution.

Relevant literature that you may or may not want to include (very recent papers): Eros numerical model (new version) : Croissant et al., in press, Nature Geosciences : illustrating the critical role of dynamic channel width in exporting sediment in bedrock valleys.

Detailed comments: P2 L23 : I tend to disagree with this statement: some models of channel width adjustment have been proposed, but none can actually fully explain the variety of responses found in nature (see Lague, 2014 ESPL, for a synthesis). As for incision thresholds, which can only been adequately accounted for if discharge variability is explicitly modelled, only two models that I know of properly account for it (CHILD, EROS and LANDLAB ?).

P2 L24: rarely: could you specify which models actually includes it ?

Section 2.1 : in this section, the author should emphasize more systematically that the "theory" presented is an assumption of the model. Too often, it is presented almost as a fact or acknowledged theory: P4L23: "vertical erosion rate is derived from", p5L2 "the rate of vertical erosion scales as".

P5L7 : given the emphasis in the introduction of the role of dynamic width, I'm surprised that you introduce a fixed width scaling with discharge without more justification. The width scaling should appear as an independent equation number so that it can be discussed much more extensively in the paper.

P6L16 : I fail to follow the logic in relating a higher $K_l/K_v$ to the work of Harsthorn et al. 2002 (who studied only one reach with variable discharge, and highlighted the role of bed cover not runoff per se) and to the increase in climate storminess described by Stark et al., which is not accounted for in your description of R (knowing that an increase in climate storminess can very likely affect $k_w$ too).

P6L20 : $k_w$ : we need more info on the range of possible values. Is this value extracted from alluvial channels (as would suggest the Leopold & Maddock, reference) which is inconsistent with your approach of "bedrock channels" as stated in the title, or from bedrock channels (which your model description seems to imply) ? You should also state at some point that $k_w$ is assumed fixed, which is a very strong assumption given that width variation with incision rate are very often observed or predicted in models explicitly modelling bed and bank erosion via an hydrodynamic model (e.g., Lague, 2014; Croissant et al., in press).

Questions on numerical implementation

CRITICAL : Is there an internal "safety check" that verifies that the actual channel width in the primary node ($k_w \times Q_w^{0.5}$) is systematically smaller than the pixel size ? otherwise you violate some of your assumptions.

Figure 1: the legend is quite hard to follow. Similarly there are several black arrows so it's hard to clearly understand which one you're referring to in the legend. Please revise this significantly for better clarity. There is also a typo ("after after" L 6). I suggest for instance to give a different color to the area being eroded in the lateral node to make it clearer.

Figure 1b: it is not clear why you choose to have the neighbouring node set to the downstream elevation node ($Z_d$), not the primary node ($Z_n$). It seems to me that this probably drives artificial mobility in the model without a real justification.

Lateral erosion : If I understand well, lateral erosion only occurs on a D4 grid, never for diagonal pixels ? Would this not generate asymmetric behaviour between orthogonal and diagonal directions favouring one orientation but not the other ?

P7L25: I note that if you add a subpixel description of the actual channel position, you would have a much more continuous description of the curvature (albeit with the issue of scale remaining).

P7L25: I fail to really understand this part ? how can you get a curvature with a straight channel ? Again this seems like assuming that you have a sub-pixel variability in the channel position, yet, you do not explicitly account for it and you do not have a model for it.

P7L30 : H only dependent on Q : incorrect assumption to have H independent of slope which can vary alongstream and through time. Why can't you use your local width, slope and friction to backcalculate the actual local flow depth ?

P7L32 : does all the sediment behaves according to eqs (1) to (6) or is there specific treatment for the collapsed material as mentioned in Fig 1d: 'collapse material" behaves as washload , which would potentially imply that it nevers redeposit in the channel ? More generally, I find that the behaviour of the sediment is not always clear. (note having reread the MS several time, I now understand, but it's really not clear on

the first or second read)

End-member formulation

P8L10: I think it would be way more justifiable to present the end-member as exploring lateral erosion laws scaling with bank height (as in Coulthard et al., 2013) or flow depth (as in many hydrodynamic models, Delft3D etc...), and using this terminology all along the paper, and trying to relate these to actual natural processes in the discussion section, rather than the other way around. Because, the link with actual processes is quite tenuous, and there is some kind of untold story that the actual erosion model is dependent on the rock resistance chosen in the model. It would be great to beef up the literature here, discussing for comparison how bank erosion is calculated in CAESAR or EROS.

Model experiments:

P8L22: Why cannot you use the model with lateral mobility from the beginning ? what kind of hillslope erosion law is used ?

how were the parameters chosen ? e.g., erodibility, alpha as well as the Kl/Kv ratio and a runoff rate of 14 mm/hr or 36 mm/hr ? I note that 36 mm/hr amounts at 315 m/yr of runoff... Given, that nowhere on earth you have this kind of mean annual runoff, I suspect that this is some kind of effective runoff, but it is really not clear. Given that you do not chose the runoff, ending up with such large values should be better discussed. Seems that to get results that look good, you have to end up using boundary conditions that are unrealistic More generally, it is not clear if your choice of parameter is such that the landscape & mobility looks "ok", or if at least, some can be independently chosen ? Maybe you should present a reference catchment on which model results could be compared.

Given that your parameter choice seems quite ad hoc, I find it quite misleading/dangerous to present "real ages" in the numerical simulations and in the results.
P11L14: maybe you could cite Davy and Lague (2009) in which there's the first derivation of the slope-area relationship in the general case of erosion-deposition with a transport distance.

P11L15: If you had an independent calibration of your elementary laws, which, when implemented in the numerical model, generates realistic geometries, then you would demonstrate that your new lateral erosion theory and its implementation successfully produce bedrock valleys significantly larger than the channel that created them. But right now, the model is calibrated and constructed to generate these wide valleys, so obviously. . .you get them. . . We are really bordering circular reasoning here.

P12L21: which hillslope processes, you did not describe them and in the discussion you seem to imply that there are no hillslope processes operating.

P13L13 : careful with the notion of threshold: this is not a true threshold in terms of physical processes (there are no thresholds in the constitutive equations of the problem), but solely an artificial threshold introduced by the numerical implementation and which depends on grid size.

Section 4.2.1 : this section needs to be revised in the light that the predicted channel width is very likely larger than the actual pixel width which violates a fundamental assumption of the model (see general comments)

P13L27 : this is an interesting feedback.

P14L25 : the increase in lateral erosion rate could be quite dependent on the incorrect assumption that H only varies with discharge (while it varies also with slope), and the flow partitioning errors as at this stage the "channel" theoretically occupies at least 2 pixels which means that discharge should not be as high than predicted given that it is focused in a single pixel.

P15 & P16 : in this section, assuming that channels only accommodate the increased sediment flux by varying their slope without varying their width (in that case kw), is a

pretty strong simplification. Croissant et al., in press at Nature Geosciences have recently demonstrated how important are dynamic width variations (i.e., kw variations) in boosting the transport capacity of mountain rivers, slope variations having secondary effects. This effect, important in driving channel reincision of deposits, terrace generation and channel mobility cannot be captured in your modelling framework if you assume kw is fixed.

P16L25 : here, assuming that water depth does not depend on slope overpredicts lateral erosion with respect to vertical erosion as water depth should decrease with slope for given discharge.

Discussion: P17L31: the valley width emerging from any of the lateral erosion model completely depends on the model parametrization which is not properly justified at present. You could obtain narrower valleys with the undercutting-slump model algorithm if the lateral erodibility is much smaller.

P19L20: this is debatable: alpha depends on runoff and settling velocity which can easily be estimated for natural systems. Only d* is more tricky. Setting runoff and settling velocity should set the value of alpha, not the other way around. At least you're sure to evolve in a range of parameters that is realistic.

---

## Editor Comment (EC1) · JM Turowski (Editor) · 10 Jul 2017

Dear authors,

we have received three reviews for your paper now. Two of them were solicited (one anonymous, one by Dimitri Lague), and one unsolicited (by Aaron Bufe). All three of these appreciate the aims of the paper, but also highlight serious problems. In my mind, the most important points are these:

- Reviewer #1 points out the connection of lateral erosion to meandering, the minimum

requirements for the modelling of which have been studied by Knutson and Howard (1984). The considerations in this paper should be discussed and incorporated in any revisions of the model.

- Reviewer #2 is concerned about grid resolution issues and the treatment of channel width. I think these points are very important and should be taken seriously. I'd like to point out here the paper of Stark and Stark (Am. J. Sci. 2001), who developed a sub-grid approach to treat channels, which may be instructive for dealing with this criticism. A related point here is the frequency of contact of the flowing water with the banks, as made by Bufe, the importance of which has been pointed out by Hancock and Anderson (GSAB 2002).

- Both reviewers #1 and #2 also pointed out some previous treatments of bank erosion/lateral mobility/meandering, for instance the above-mentioned paper by Knutson and Howard, but also the treatments within CAESAR (e.g., Coulthard and van de Wiel, ESPL 2006) and EROS (e.g., Carretier et al., ESurf 2016). A review of these treatments would be appropriate, highlighting of their different merits and why another (new) approach is necessary.

Many of these points culminate in the statement made explicitly by reviewer #1 at the end of the review. You construct some model and explore its dynamics to some extent, but the question remains as to why we should believe that it is a true or even useful description of reality. I agree here that the paper could benefit from a well-defined research hypothesis and a set of criteria that could be used to evaluate the model against field data or compare it against the performance of other available models.

All three reviews provide detailed comments and give suggestion in which direction the paper could be developed. Please treat all comments seriously and provide a detailed rebuttal. I will send out the paper for a full review again.

Best wishes and good luck with revisions,

Jens Turowski

---

## Author Comment (AC1) · 21 Aug 2017

**Authors' Response to Reviewer 1**

**Reviewer 1:** *This paper describes a new approach to incorporating lateral channel erosion into landscape evolution models. This is clearly a worthwhile goal and I was excited to read this. That said, there are aspects of the model setup and motivation, as described below, that I think can be improved upon and which I think will lead to a paper with more impact.*

**Authors:** Thank you, we appreciate your suggestions to help our manuscript have more impact and interest to a broad audience.

**Reviewer 1:** *1. Specifically, this paper uses a curvature based wall erosion law. While the authors don't expressly say they are modeling meandering, this is the implication of the choice of model. This makes sense as meanders are ubiquitous in bedrock channels and the process is clearly important in many settings.*

**Authors:** We view the class of "streams with fully developed meandering" as a relatively small subset of "streams able to widen valleys through lateral erosion." In our field experience, there are plenty of examples of streams that most geomorphologists would classify as single-thread, and yet which clearly show evidence of erosion and lateral migration at locations where an outer bend in the channel impinges on a valley wall or terrace. Conceptually, therefore, our approach is not meant to represent exclusively channels with fully developed meandering. To clarify this point for readers, we have added text to the manuscript in the section describing the lateral erosion component of the model.

**Reviewer 1:** *The first numerical model of river meandering that I am aware of is Howard and Knutson (1984). Their first iteration of the model is one in which erosion scales inversely with the radius of curvature, which is basically the same as the model posed in equation 10. Howard and Knutson point out that such a model results in a channel that breaks down into 3 point bends with alternating positive and negative curvature. When applied to an existing meander bend, the bend can't be maintained. The ultimate conclusion of Howard and Knutson (1984) is that lateral channel motion can't be driven only by local curvature because such a model fails to produce realistic meander kinematics (down stream translation, cutoffs) as well as realistic meander forms. This is what leads to their downstream convolution approach, which in a simply way simulates the advection of the effects of upstream curvature downstream. Given that the setup of the model in the submitted MS is based on a centrifugal acceleration argument, and given that the morphologies of the channels produced in the model are reminiscent of the 3 point bends described by Howard and Knutson, it's not clear to me how this model represents a significant advance in understanding and modeling lateral erosion. Moreover, it's not clear how the river even changes from moving in one lateral direction to the other.*

**Authors:** These comments arise reflect an understandable confusion about the key differences between a meander model and a landscape evolution model. We have now added a substantial amount of text to the new "Approach and Scope" section to articulate these differences. In brief, the former represents the trace of a single channel whereas the latter represents the topography in which channels are embedded. This is a very important distinction, which we hope is now clear in the revised manuscript.

Example excerpt from text added to new "Approach and Scope" section: "Considerable advances have been made in developing theory and models for the planform dynamics of single-thread meandering channels. As a result, the scientific community has a good understanding of how meander patterns form and evolve, and how meander wavelength and migration rate scale with properties such as water discharge, valley gradient, and sediment grain size (Hooke, 1975; Nanson

and Hickin, 1986; Schumm, 1967; Langbein and Leopold, 1966; Lancaster and Bras, 2002, e.g.). This body of work addresses the planform pattern of river channels, but does not deal with the broader drainage-basin topography in which those channels are embedded. [...] There is also a well-developed literature on process models of landscape evolution, and in particular the evolution of ridge-valley topography sculpted around drainage networks. We refer to these models as Landscape Evolution Models, or LEMs. LEMs differ from meander models in treating a self-forming, two-dimensional flow network rather than a single channel reach, and in explicitly modeling the evolution of topography."

**Reviewer 1:** *What is novel, from my perspective, are the two different formulations of the wall erosion law. Why not, then, simply use the Howard and Knutson meandering model and then explore how the two different wall erosion formulations influence the emergent valley form? Given that field evidence that can discriminate between the two proposed lateral erosion processes should be straightforward to collect, I could see such an exercise leading to numerous field testable hypotheses.*

**Authors:** Identifying field sites and collecting data to evaluate the model's performance is part of the future plan, but is beyond the scope of this manuscript, which is meant to introduce the model to the community. See also response below.

**Reviewer 1:** *2. While I like the exploratory aspect of this paper, I think it could benefit from either a sharply formulated research hypothesis or a field example or two that are targeted. As is, it's not clear how we can evaluate the performance of the model other than by simply noting that the river causes the valley walls to move. But I think we could do better.*

**Authors:** We have added a figure with examples of bedrock valleys and strath terraces that are much wider than their channels in several different environments, including wide valleys created by both meandering and non-sinuous rivers. This figure demonstrates qualitatively the differences between a typical narrow bedrock valley and a valley that has experienced a phase of significant lateral erosion. We have also added a significant amount of text at the end of the discussion section where we discuss different measurements and metrics needed from field or lab experiments in order to test this and future models. We also present a potential test of the model presented in the manuscript in a specific field site.

**References**

Hooke, R. L. B. (1975). Distribution of sediment transport and shear stress in a meander bend. *The Journal of geology*, 83(5):543–565.

Lancaster, S. T. and Bras, R. L. (2002). A simple model of river meandering and its comparison to natural channels. *Hydrological Processes*, 16(1):1–26.

Langbein, W. B. and Leopold, L. B. (1966). *River meanders–theory of minimum variance.* US Government Printing Office.

Nanson, G. C. and Hickin, E. J. (1986). A statistical analysis of bank erosion and channel migration in western canada. *Geological Society of America Bulletin*, 97(4):497–504.

Schumm, S. (1967). Meander wavelength of alluvial rivers. *Science*, 157(3796):1549–1550.

---

## Author Comment (AC2) · 21 Aug 2017

**Authors' Response to Reviewer 2**

**Reviewer 2:** *This article aims at developing and implementing a model of lateral mobility of rivers in long-term landscape evolution model of mountain ranges. This is timely needed as the lateral mobility of river is now known to play a significant contribution in landscape reshaping, and as most current numerical models of landscape evolution predict valley bottom that are simply 1 pixel wide and fixed in time.*

**Authors:** Thank you for recognizing the importance of the issue we address in the paper and taking the time to make many relevant and helpful comments to improve our manuscript.

**Reviewer 2:** *The problem is that I find that the numerical implementation have several flaws which prevent me from trusting the model outcome at this stage. Numerical modellers all know that it is very easy to create landscapes that look ok if you have some large degree of freedom in choosing your model parameters (erodibility, runoff, channel width coefficient etc. . .). Here, the modelling results look ok, as the model is tuned to looks right, but that does not mean that the dynamics and timing are relevant to natural systems, which is what we ultimately expect from a landscape evolution model. And because there's no real attempt to validate model predictions against quantified observables, it is very difficult, given some of the flaw in the implementation, to infer reliable results pertaining to the dynamics of natural mountain valleys.*

**Authors:** We readily admit that this is a model without a quantitative test... yet. Qualitative reproduction of commonly observed landforms may be a weak test, but it an essential one: if a model does not pass that bar, then it is clearly a failure (albeit possibly an instructive one). Moreover, the history of science is full of examples in which theory precedes empirical confirmation. Nonetheless, it is fair to expect us to provide some ideas on how this model could be tested, and we now do so in the discussion section.

**Reviewer 2:** *The model presented here is a non-hydrodynamic model aiming at including a "channel mobility" component. This is a great idea, and indeed barely addressed by landscape evolution models. But it is not strictly speaking a "bank erosion model" as it does not resolve 2D flow hydrodynamics. Yet there are many instances in the paper, where the model has some kind of schizophrenic behavior between the two types of models:*

*First it uses a relatively small pixel size (10 m), which assumes practically that the channel width must never be larger than 10 m. Unfortunately, this condition is not verified all the time (unless I've missed something in the calculations): the basic model uses a drainage area of 20000 m2 , which coupled with a runoff of 36 mm/hr, and kw=10, gives Wmin = 4.5 m. However, multiplying the drainage area to 160000m2 (section 4.2.1) violates this assumption from the inlet of the model Wmin = 12.65 m. At this point flow should be partitioned over 2 pixels to correctly resolve the equations. I don't know how this bias affects the model predictions, and how such a model could be uspcaled to larger catchments where channel width would be several pixel wide (here we're dealing with small catchments of km2 size).*

**Authors:** It is important to recognize that channel width is not explicitly represented in the model we describe. Rather, it is one element of the lumped parameters $K_v$ and $K_l$. The channel-width scaling parameter values we discuss, and which the reviewer quotes, are used only in the estimation of reasonable ranges for these parameters. So it is not really correct to say that the model channels are wider in some instances than their grid cells, because channels have no explicitly defined width (though the possibility exists that the "implied width" could potentially be wider than a cell: a problem common to all non-hydrodynamic LEMs). We have added text to section describing vertical and lateral erosion equations to make this point.

The reviewer remarks that grid cells are "relatively small". The word "relatively" is important here. Presumably he means small relative to what one would expect for channel width. If we consider that channel width tends to scale as the square root of drainage area, then all else equal it should also scale with the characteristic length of the drainage basin, or in the case of a model, the side length of the domain. Double this characteristic length scale, and you should also double the expected width of the largest channel. Given this scaling relation, it does not really make sense to speak in terms of the absolute size of model grid cell. Rather, it makes more sense to consider grid size in relation to the scale of the largest drainage basin. In that respect, our model resolution (considered as the ratio of cell size to domain size) is not notably different from that of most other non-hydrodynamic LEMs.

Nonetheless, the reviewer raises an important general critique of LEMs that use single-direction flow-routing schemes: it is possible in principle to have an "implied width" (implied, that is, by the width-discharge relation embedded in K) that is larger than a grid-cell size. This issue is not unique to our particular model; any non-hydrodynamic LEM with sufficient resolution would face the same inconsistency. We agree that it is an issue that should be resolved (interestingly, the same kind of issue arises in other fields, such as the representation of convection cells in atmospheric models or turbulent eddies in 3D flow models). However, our intent here lies not in re-writing the hydrology parameterization for LEMs, but rather with the more narrow goal of investigating how lateral erosion might be implemented within the context of an otherwise fairly generic and common model formulation, without excessive complexity. Therefore, while we acknowledge the "channel in cell" issue as a general problem for non-hydrodynamic LEMs (and indeed related to similar issues we don't think this paper is the right place to roll out a proposed solution. However, we have added text in the supplementary materials section that notes the existence of this issue and the need for an ultimate solution.

**Reviewer 2:** - *There is no real notion of "bank" in the model given that the channel is defined at sub grid, but rather some kind of "valley side". This makes it difficult to directly relate lateral erosion "end members" (fig.1 section 3.1) to actual physical processes. These are more numerical tricks to resolve vertical feature horizontal migration on fixed horizontal grid, but whose relevance to natural processes is quite debatable. They introduce artificial thresholds in model dynamics whose consequences are not explored thoroughly.*

**Authors:** Perhaps so, but note that the physical sciences are full of such "tricks". What, for example, is the "true" meaning of viscosity in a liquid? The linear viscosity law is just a parameterization ("trick") too, which happens to work well for certain materials under a certain range of conditions (and fails for others). Maybe our trick will ultimately prove to perform poorly when compared with data, yet by introducing it we draw attention to what we hypothesize is an important process in valley widening: the physical disaggregation of material due to erosional undermining and collapse.

The model end member section was revised (P9L16-30) to emphasize that in one formulation (total block erosion), lateral erosion scales with valley wall height and in the other (undercutting slump), lateral erosion is independent of valley wall height. All discussion of relevance to natural processes is moved to the discussion section.

**Reviewer 2:** *The model implementation assumes that the channel is always in contact with the neighbour node (there is systematically lateral erosion), which contradicts the underlying assumption that channel width is smaller than the pixel size.*

**Authors:** See response below under "two components missing in the model", item 2.

**Reviewer 2:** *The model does not account for lateral deposition which is an important driver of channel migration (but that's not the most critical point)*

**Authors:** Sustained deposition on surfaces dipping more than several degrees is so rare that we consider it a reasonable thing to neglect.

**Reviewer 2:** *On top of this, there is an important limitation in the "undercutting- slump" model in assuming that flow depth only depend on discharge (eq. 30) while it must depend on slope (and width, but given that it is fixed by discharge in the model, there's no way to do better).*

**Authors:** We neglect the influence of slope on water depth because its influence is much less than that of discharge. For example, the Manning equation states that depth scales like $Q^{3/5}$ but scales like $S^{3/10}$.

**Reviewer 2:** *Hence I see at least two components missing in the model: 1: A proper way to deal with cases in which the channel width becomes larger than the pixel size (as predicted by $kwQ^{0.5}$): either you increase the pixel size (but this also increases the "numerical" threshold for channel migration), or you introduce some kind of flow partitioning/simplified 2D hydrodynamics (but then we're very close to existing models like CAESAR or EROS). I know width is lumped in the model through kw, but either you assume your channel width is never larger than 10 m (that's quite a limiting factor), or you have to partition the flow over several pixels.*

**Authors:** See prior responses regarding the treatment of width in our model.

**Reviewer 2:** *2: Adding a way to either explicitly or implicitly account for the sub-pixel position of the channel. For instance a kind of likelihood of bank erosion (which is a function of the ratio of channel width to pixel size) with an asymmetric probability related to along stream curvature.*

**Authors:** We are aware of course of the excellent work by Hancock and Anderson (2002) that relates valley widening rate to the ratio of channel to valley width. We had originally avoided implementing such a rule because, as noted earlier, the model does not explicitly define channel width. However, even without tracking width explicitly, one could assume $W \sim A^{1/2}$ scaling and therefore allow a similar scaling in lateral erosion rate. One way to address this issue in the model is to multiply the erosion rate by the ratio of channel width/dx so that lateral erosion is decreased in narrow streams and enhanced in larger streams. We have created a version of the model that implements this rule, and run a series test models to evaluate the result. As expected, there is less lateral erosion in the smaller streams in the upper parts of the model, but little change in valley width and channel mobility in the lower parts of the channel. These figures and discussion of the modified model are included in the supplementary materials.

As to the notion of tracking sub-cell channel position: We are delighted that the manuscript is already provoking new ideas about how to address the problem that we've set out to highlight. Indeed, we spent a long time considering various approaches, including one in which channel position within a cell is explicitly tracked. We ultimately settled on the alternative method the paper describes out of considerations for simplicity. Complexity in theory and models comes at a cost. Our philosophy is that the goal of science is to understand things, and if a model becomes too complex to understand, well then all we've succeeded in doing is creating yet another thing we don't understand. In our view, the justification for adding something to a model should be a clear demonstration that the model doesn't "work" (i.e., account for an observed phenomenon) without that thing. So, we've leaned on the side of simplicity. If this paper stimulates others to come up with a demonstrably better approach, then we'll have succeeded in one of our main objectives.

**Reviewer 2:** *I also note that, even if it is not common practice in the literature of landscape*

*evolution models (it should), it is important for any numerical model implementation, to demonstrate that the model results do not systematically depend on grid size (within limits) and time-step, or to acknowledge this dependency and demonstrate how it impact results.*

**Authors:** A brief overview of model runs with the same domain size and grid size of 15 m and 20 m is included in the supplementary materials.

**Reviewer 2:** *Also, I would also like to see the model evolve from an initial condition with the lateral erosion "on", and not activated only when the landscape and drainage is already organized: if a model works, it works all the time, and actually exploring drainage development on a plateau could tell us whether you generate realistic patterns or not.*

**Authors:** The models can be run from an initial condition with lateral erosion. There is no observable difference in model topography. Figures and a brief discussion are included in the supplementary materials.

**Reviewer 2:** *Other comments Title: it is currently slightly misleading as there is no real evaluation nor comparison of the model prediction with actual results, and the link with the mechanics of bedrock channel bank erosion are extremely tenuous or not really clear. Something like: "Implementing lateral mobility of channels in landscape evolution" models would more represent the actual content of the paper.*

**Authors:** We have changed the title slightly to more accurately reflect the content of the paper. The new title of the paper is: Developing and exploring a theory for the lateral erosion of bedrock channels for use in landscape evolution models.

**Reviewer 2:** *Missing literature: The CAESAR numerical model...*

**Authors:** Missing literature added in background section and throughout the manuscript as noted below.

**Reviewer 2:** *P2 L23 : I tend to disagree with this statement: some models of channel width adjustment have been proposed, but none can actually fully explain the variety of responses found in nature (see Lague, 2014 ESPL, for a synthesis). As for incision thresholds, which can only been adequately accounted for if discharge variability is explicitly modelled, only two models that I know of properly account for it (CHILD, EROS and LANDLAB ?).*

**Authors:** Changed text to read: While theories that account for dynamic adjustment to bedrock channel width continue to be refined (Lague, 2014), landscape evolution models that include a relationship between sediment size and cover (e.g. Sklar and Dietrich, 2004), and incision thresholds in bedrock channels (Tucker et al, 2001; Crave and Davy, 2001; Tucker et al., 2013) are available and widely used (Tucker and Hancock, 2013).

**Reviewer 2:** *P2 L24: rarely: could you specify which models actually includes it?*

**Authors:** Changed text to read: "existing models do not address the lateral erosion of bedrock channel walls"

**Reviewer 2:** *Section 2.1 : in this section, the author should emphasize more systematically that the "theory" presented is an assumption of the model. Too often, it is presented almost as a fact or acknowledged theory:*

**Authors:** Changed text to clarify to readers that vertical incision in our model is represented by the stream power model and added text about we chose this model in the discussion section.

**Reviewer 2:** *P5L7 : given the emphasis in the introduction of the role of dynamic width, Im surprised that you introduce a fixed width scaling with discharge without more justification. The width scaling should appear as an independent equation number so that it can be discussed much more extensively in the paper.*

**Authors:** As explained above, the model does not explicitly calculate channel width. Rather, a discussion of width scaling is presented in the paper simply as a consideration of what parameter values might be considered reasonable. We have added text to the section following the lateral erosion equations to clarify this point.

**Reviewer 2:** *P6L16 : I fail to follow the logic in relating a higher Kl/Kv to the work of Harsthorn et al. 2002 (who studied only one reach with variable discharge, and highlighted the role of bed cover not runoff per se) and to the increase in climate storminess described by Stark et al., which is not accounted for in your description of R (knowing that an increase in climate storminess can very likely affect kw too).*

**Authors:** Moved these references to Hartshorn et al. and Stark et al. to the discussion section and expanded discussion of the effects of Kl/Kv ratio.

**Reviewer 2:** *P6L20 : kw : we need more info on the range of possible values. Is this value extracted from alluvial channels (as would suggest the Leopold Maddock, reference) which is inconsistent with your approach of "bedrock channels" as stated in the title, or from bedrock channels (which your model description seems to imply) ? You should also state at some point that kw is assumed fixed, which is a very strong assumption given that width variation with incision rate are very often observed or predicted in models explicitly modelling bed and bank erosion via an hydrodynamic model (e.g., Lague, 2014; Croissant et al., in press).*

**Authors:** Updated text to discuss use of fixed kw and range of possible values of kw (P8L1-5).

**Reviewer 2:** *CRITICAL : Is there an internal "safety check" that verifies that the actual channel width in the primary node (kw $Q^{0.5}$) is systematically smaller than the pixel size ? otherwise you violate some of your assumptions.*

**Authors:** As explained above, there is no explicitly defined width in our model. Text was added to the manuscript in the section describing vertical and lateral erosion equations to make this point.

**Reviewer 2:** *Figure 1: the legend is quite hard to follow. Similarly there are several black arrows so its hard to clearly understand which one you're referring to in the legend. Please revise this significantly for better clarity. There is also a typo ("after after" L 6). I suggest for instance to give a different color to the area being eroded in the lateral node to make it clearer.*

**Authors:** Typo fixed and figure revised for clarity.

**Reviewer 2:** *Figure 1b: it is not clear why you choose to have the neighbouring node set to the downstream elevation node (Zd), not the primary node (Zn). It seems to me that this probably drives artificial mobility in the model without a real justification.*

**Authors:** Setting the elevation of the lateral node equal to the elevation of the primary node would make the valley slightly wider, but the channel immobile. That is because water flow in that case would continue to prefer going from upstream through primary to downstream, because the "detour" through lateral would have a lower slope (same altitude difference, more distance covered). With flow continuing to prefer the shorter route, that is where the erosive action would be, and the just-eroded lateral node would be left at the original altitude of the primary node.

Setting the elevation of the lateral node equal to the downstream node gives the opportunity for flow to be rerouted through the lateral node, but does not require it.

**Reviewer 2:** *Lateral erosion : If I understand well, lateral erosion only occurs on a D4 grid, never for diagonal pixels? Would this not generate asymmetric behaviour between orthogonal and diagonal directions favouring one orientation but not the other?*

**Authors:** A new supplementary materials document has been written and includes a figure detailing how lateral nodes are chosen in the model. To briefly answer the question, a lateral neighbor node can only be the E,S,W,or N neighbor of the primary node, but the lateral node can be to the diagonal direction of a flow line in the case of 45 degree bends or two straight segments that flow across diagonals.

**Reviewer 2:** *P7L25: I note that if you add a subpixel description of the actual channel position, you would have a much more continuous description of the curvature (albeit with the issue of scale remaining).*

**Authors:** Yes, but as noted earlier, that would defeat the purpose of having a simple, low-dimensional model formulation.

**Reviewer 2:** *: P7L25 I fail to really understand this part ? how can you get a curvature with a straight channel ? Again this seems like assuming that you have a sub-pixel variability in the channel position, yet, you do not explicitly account for it and you do not have a model for it.*

**Authors:** A figure detailing how radius of curvature is calculated is included in the supplementary materials document. Yes, you are correct in that the way radius of curvature for straight channels is calculated is a simple way to account for sub-pixel variability in channel position.

**Reviewer 2:** *: P7L30 : H only dependent on Q : incorrect assumption to have H independent of slope which can vary alongstream and through time. Why can't you use your local width, slope and friction to back calculate the actual local flow depth?*

**Authors:** You are that it is possible to calculate flow depth in the model for each cell based on width, slope, and friction, but we choose to use a simpler hydraulic geometry relationship for flow depth as many other non-hydrodynamic landscape evolution models do (CHILD, etc). We emphasize that our goal in developing this model of lateral bedrock erosion is to start with the simplest reasonable erosion model so that we can focus on understanding the dynamics of lateral erosion. Additionally, as noted above, H scales more strongly with Q than with S.

**Reviewer 2:** *: P7L32 : does all the sediment behaves according to eqs (1) to (6) or is there specific treatment for the collapsed material as mentioned in Fig 1d: 'collapse material" behaves as washload , which would potentially imply that it nevers redeposit in the channel ? More generally, I find that the behaviour of the sediment is not always clear. (note having reread the MS several time, I now understand, but it's really not clear on the first or second read).*

**Authors:** Text has been added to clarify the treatment of sediment in the model at the end of the numerical implementation section (P9L9-14).

**Reviewer 2:** *P8L10: I think it would be way more justifiable to present the end-member as exploring lateral erosion laws scaling with bank height (as in Coulthard et al., 2013) or flow depth (as in many hydrodynamic models, Delft3D etc. . .), and using this terminology all along the paper, and trying to relate these to actual natural processes in the discussion section, rather than the other way around. Because, the link with actual processes is quite tenuous, and there is some*

*kind of untold story that the actual erosion model is dependent on the rock resistance chosen in the model. It would be great to beef up the literature here, discussing for comparison how bank erosion is calculated in CAESAR or EROS.*

**Authors:** Revised this section to emphasize the end member models as representing valley widening as a function of wall height and moved links to natural processes to the discussion section. Discussion of bank erosion in CAESAR and EROS has been added to the background section.

**Reviewer 2:** *P8L22: Why cannot you use the model with lateral mobility from the beginning ? what kind of hillslope erosion law is used ?*

**Authors:**It is possible to use the model with lateral erosion from the beginning. Figures comparing model runs with lateral erosion from the beginning and lateral erosion started after topography was initialized are shown in the supplementary materials section and show no difference in model topography.

**Reviewer 2:** *how were the parameters chosen ? e.g., erodibility, alpha as well as the Kl/Kv ratio and a runoff rate of 14 mm/hr or 36 mm/hr ? I note that 36 mm/hr amounts at 315 m/yr of runoff. . . Given, that nowhere on earth you have this kind of mean annual runoff, I suspect that this is some kind of effective runoff, but it is really not clear. Given that you do not chose the runoff, ending up with such large values should be better discussed. Seems that to get results that look good, you have to end up using boundary conditions that are unrealistic More generally, it is not clear if your choice of parameter is such that the landscape mobility looks "ok", or if at least, some can be independently chosen ? Maybe you should present a reference catchment on which model results could be compared.*

**Authors:** A range of values for K and alpha were chosen to explore model behavior, specifically channel mobility and valley width. References were added supporting the range of K and alpha values chosen in the model runs. In order to demonstrate the range of possible lateral erosion and valley widths in our new model actually works, we used a high, but justifiable value for runoff on event time scales. Text on how runoff values were chosen was added to the paper. High values of runoff, which are meant to represent peak values, not mean annual values, were chosen to get Kl/Kv ratios of 1 and 1.5 in order to demonstrate the lateral erosion that emerges from the model. The parameter values were chosen from a range of reasonable values found in nature. In some cases, significant channel mobility and lateral erosion occurred (these cases are highlighted in the manuscript), but in some cases, little observable lateral erosion occurred, see Figures 5a,6a; 9a,c;

**Reviewer 2:** *Given that your parameter choice seems quite ad hoc, I find it quite misleading/dangerous to present "real ages" in the numerical simulations and in the results.*

**Authors:** The choice, really, is whether to present figures like this in dimensional or non-dimensional form. The latter is of course more elegant, and has the advantage of demonstrating the role of multiple variables and their interactions. However, feedback from colleagues and students indicates that many find dimensional plots more intuitive, so we have stuck with them. As regards danger, all we can say is that no students or colleagues were harmed in the writing of this paper, and we don't think anyone will be harmed by reading it.

**Reviewer 2:** *P11L14: maybe you could cite Davy and Lague (2009) in which there's the first derivation of the slope-area relationship in the general case of erosion-deposition with a transport distance.*

**Authors:** Thank you, this paper is now cited here.

**Reviewer 2:** *P11L15: If you had an independent calibration of your elementary laws, which, when implemented in the numerical model, generates realistic geometries, then you would demonstrate that your new lateral erosion theory and its implementation successfully produce bedrock valleys significantly larger than the channel that created them. But right now, the model is calibrated and constructed to generate these wide valleys, so obviously. . .you get them. . . We are really bordering circular reasoning here.*

**Authors:** Not at all! The reviewer seems to assume that ANY set of rules or equations could reproduce any desired set of landforms as long as the right parameters are chosen. We disagree. When a model for a particular natural pattern is proposed, it may either succeed or fail at the basic test of qualitative reproduction of the pattern in question. We have shown that our model succeeds at the basic task of qualitative reproduction. This might be a weak test, but does not mean success is inevitable. Of course, it would be wonderful to have independent constraints on parameter values. Nonetheless, we adamantly disagree that demonstrating qualitative consistency between a model and observations, given certain parameter ranges, constitutes circular reasoning.

**Reviewer 2:** *P12L21: which hillslope processes, you did not describe them and in the discussion you seem to imply that there are no hillslope processes operating.*

**Authors:** Text changed to reflect that indeed there are no hillslope processes in this model.

**Reviewer 2:** *P13L13 : careful with the notion of threshold: this is not a true threshold in terms of physical processes (there are no thresholds in the constitutive equations of the problem), but solely an artificial threshold introduced by the numerical implementation and which depends on grid size.*

**Authors:** The word threshold is removed and text clarified.

**Reviewer 2:** *Section 4.2.1 : this section needs to be revised in the light that the predicted channel width is very likely larger than the actual pixel width which violates a fundamental assumption of the model (see general comments)*

**Authors:** We have run the models in section 4.2.1 with with grid size of 15 m and 20 m and compared them to the original models runs with dx=10 m. The new models with larger grid size shows some differences with the original models, but are largely similar in the amount of lateral erosion accomplished and width of bedrock valley created. Figures and discussion of these model runs is included the supplementary materials section.

**Reviewer 2:** *P14L25 : the increase in lateral erosion rate could be quite dependent on the incorrect assumption that H only varies with discharge (while it varies also with slope), and the flow partitioning errors as at this stage the "channel" theoretically occupies at least 2 pixels which means that discharge should not be as high than predicted given that it is focused in a single pixel.*

**Authors:** See responses above regarding scaling relationships and treatment of channel width used in this model.

**Reviewer 2:** *P15  P16 : in this section, assuming that channels only accommodate the increased sediment flux by varying their slope without varying their width (in that case kw), is a pretty strong simplification. Croissant et al., in press at Nature Geosciences have recently demonstrated how important are dynamic width variations (i.e., kw variations) in boosting the transport capacity of mountain rivers, slope variations having secondary effects. This effect, important in driving channel reincision of deposits, terrace generation and channel mobility cannot be captured in your modelling framework if you assume kw is fixed.*

**Authors:** Added text in this section to remind readers of the fixed width scaling in this model that prevents channels from changing width in response to sediment flux and added a paragraph in the discussion section detailing the implications for the model (P21L8-13).

**Reviewer 2:** *P17L31: the valley width emerging from any of the lateral erosion model completely depends on the model parametrization which is not properly justified at present. You could obtain narrower valleys with the undercutting-slump model algorithm if the lateral erodibility is much smaller.*

**Authors:** Model parameters have been more thoroughly explained in model experiments section. We acknowledge that in this initial version of the lateral erosion model, valley width is often strongly related to the imposed Kl/Kv ratio. But we note that the model produced narrow valleys in undercutting slump models with high values of K and low values of alpha for both values of Kl/Kv that were tested in this paper (Figure 3c,d).

**Reviewer 2:** *P19L20: this is debatable: alpha depends on runoff and settling velocity which can easily be estimated for natural systems. Only d\* is more tricky. Setting runoff and settling velocity should set the value of alpha, not the other way around. At least you're sure to evolve in a range of parameters that is realistic.*

**Authors:** Text here reworded to clarify our intention to note the limitations of the current erosion/deposition model in future work that may address spatial and temporal changes in runoff and changes in and/or multiple grain sizes.

---

## Author Comment (AC3) · 21 Aug 2017

**Authors' Response to Short Comment**

**Commenter 1:** *I am excited to see this work come out and I am very much looking forward to seeing the paper published. Below, I am noting a few questions and comments that arose during my read through and I hope that some of these might be helpful in the revision process.*

   **Authors:** Aaron, thanks so much for taking the time to read the manuscript so thoroughly and offer some very helpful comments.

   **Commenter 1:** *In summary, there are three major points that I thought could be clarified (and that I address in more detail in the line comments below).*

   *First, the treatment of sediment in the model is only mentioned relatively briefly at the very end of the discussion section. In order to better appreciate the modeling result, I suggest that it may be helpful to lay out the role of sediment in the model (and in particular how it "becomes" bedrock when deposited) early in the section about the numerical implementation.*

   **Authors:** A section of text that more clearly describes the treatment of sediment in the lateral erosion component was added at the end of the "numerical implementation" section (P9L9-14).

   **Commenter 1:** *Second, as marked below, I suggest to introduce the concept of channel mobility earlier and maybe comment on the difference between mobility within loose sediment (which, in my limited knowledge, much of the literature that uses the term channel mobility; is concerned with), and the mobility linked to movement within a pure bedrock landscape as it is defined in this paper. This distinction may also clarify the discussion about the importance of channel mobility on p 17. I comment more extensively on this below*

   **Authors:** We added text to clarify the distinction between channel mobility in alluvial literature and the use of the term in this paper (P10L31-P11L2) and included this introduction at the beginning of the section on channel mobility. See below for more details on changes made to the paper regarding channel mobility.

   **Commenter 1:** *Third, I would suggest a more extensive discussion of the implications of using the stream power model without a treatment of sediment tools and cover. The lack of the cover effect is mentioned in a sentence on P 19 L11 but in the list of limitation that follows and in the following discussion I did not find a clear mention about the possible limitations that the absence of tools may introduce. An appreciation of the implications of the stream-power model are particularly important before model results are compared with other studies such as those by Hartshorn et al., 2001 and Fuller et al., 2016. The effects of tools and cover seem an integral part of the interpretations of the observations that were made by these cited studies (and in other studies such as Hancock and Anderson 2002 etc.).*

   *Therefore, without a discussion about the role of tools and cover, the link between the present stream power model and other studies (that is made early in the manuscript) appears problematic. Moreover, tools and cover may affect lateral and vertical erosion in different ways. For example, an increase of lateral erosion rates because of a change in the amount of sediment that is deflected toward a channel wall (Fuller et al., 2016), may or may not be accompanied by changes in vertical erosion rates.*

   *Therefore, the response of the system to a change in water or sediment flux may be more complex than predicted by the model. In short, I can imagine that a more expansive discussion of this limit may be useful. In particular, these complications should probably be mentioned before the model is compared to results from other studies.*

   **Authors:** A section of text that acknowledges the limitations of the stream power model and

explains why we chose to use the stream power model is added to the end of the "vertical erosion theory" section of the manuscript (P6L15-22). A section of text that discusses the possible effects of using a tools and cover model and how a different incision model would affect model results is added to the discussion (P20L28-P21L4).

**Commenter 1:** *P.2 L.7 Nitpicky comment: it may or may not be worth noting that cessation of incision and cutting of straths has also been observed in harder lithologies such as quartzites or granites These straths are narrow and they dont contradict the statement that sizes of strath terraces and rates of strath cutting seem strongly linked to bedrock strengths, but the way the sentence is phrased now, it could be understood that straths never form in stronger lithologies:*
**Authors:** Added sentence to clarify straths also occur in hard lithologies.

**Commenter 1:** *Somewhere in the setup and introduction (for example somewhere in the paragraph starting P.2 L.11), it might be worth mentioning published models that consider the control of valley wall-height on widening rates e.g.: Malatesta 2016*
**Authors:** Added text in the introduction to include Malatesta et al. and brief discussion of bank height.

**Commenter 1:** *P.5 L.7. Setting W=kwQ1/2 may be common knowledge but I wonder if it is worth citing the original works that this scaling is based on.*
**Authors:** Leopold and Maddock (1953) cited here.

**Commenter 1:** *P.5 L.16. I like the idea of looking at the centripetal force. I was wondering at this stage what happens to straight segments of rivers. The way straight segments are treated is layed out later: They are treated as having a range of radii of curvature. However, there is also evidence for erosion in perfectly straight channels (e.g. Fuller et al., 2016). Maybe a quick note of the limits and possible alternatives of this model formulation could be made here or in the discussion? At the latest, this should probably be mentioned when the model results are compared to results from Fuller et al., 2016.*
**Authors:** Text added in discussion on lateral erosion in straight channels: P22L1-5

**Commenter 1:** *P.6 L. 12 It was a little unclear to me what the word "which" refers to here either equation 12 or the variable Kl.*
**Authors:** changed to read : "Kl is a dimensional erosion coefficient for lateral erosion composed of known or measurable quantities..."

**Commenter 1:** *P.6 L. 16 As far as I understand, the result that Kl/Kv scales with Q1/2 or R1/2 is derived within the framework of stream power models. I feel that the link to the field studies is a little misleading in this context. The changes in the ratio of lateral to vertical erosion rates between high and low flows measured in the Liwu river (Hartshorn et al., 2001) have been interpreted to be due to changes in the distribution of sediment in the flow (interpretation by Hartshorn et al., 2001) or to variable shielding of the bed (Turowski et al., 2008, ESPL). Because the importance of tools and cover for lateral and vertical erosion are not considered in the stream power model presented here, whereas the end result may be similar between model and field study (high discharge = high lateral erosion), the processes are likely different. Therefore, a comparison between field study and model without a more extensive discussion might be misunderstood. In turn, the increased sinuosity with storminess found by Stark et al., 2010 was interpreted by the authors as an expression of the importance for hillslope mass wasting in controlling lateral erosion. This interpretation may*

*or may not be true, but it again, complicates the link of the model to this field examples. Later (P18 L10), there is a similar issue with the comparison of the model to the study by Fuller et al., 2016. see comment further down.*

*Generally, I think it is valuable to discuss whether the model behavior is observed in nature but I think it necessitates a more detailed discussion of the limits of the stream power model before the comparison can be made.*

**Authors:** Removed references to field studies here, and added them to the discussion section. Added text on the motivation for using the stream power model for vertical incision and the potential impact of using a tools and cover model to discussion section.

**Commenter 1:** *P.7 L.15 I wonder, if, before detailing the way lateral and vertical erosion is calculated, it would be worth detailing one entire timestep and the order in which equations are solved. In particular, at this point of the manuscript, I was unsure how streams migrate. As far as I understand, at the beginning of a timestep, flow is routed across a topography via a D8 algorithm, then the lateral and vertical erosion is calculated, the topography updated and the flow rerouted through the landscape. Could this maybe be briefly laid out step by step? Or as a flow chart figure? Even more importantly, at this point in the manuscript (and up to the very last paragraph of the discussion) it was unclear to me how sediment was treated. This is important to appreciate many of the features of the model (channel migration and channel mobility in particular) Questions that would be good to clarify are: Is deposited material added to the topography of a cell? What happens to a cell that is partly sediment and partly bedrock? Is the difference in erodibility considered or does deposited sediment "become" bedrock? Detailing the treatment of sediment could probably be intertwined with the walkthrough of one model timestep.*

**Authors:** A section detailing the treatment of sediment was added to the "numerical implementation" section (P9L9-14). We have made a flow chart detailing the steps taken in one model time step and included this in the supplementary material section.

**Commenter 1:** *Section 3.1: I was a little confused by the (as I understood it) differentiation between resistant lithologies for which the slumped material has to be eroded (therefore bank height is important) and weak lithologies for which all material is swept away after a slump happens (therefore bank erosion is not important). The way it is described in the text is that the material is "transported away". This formulation seems ambiguous to me. Is the material added to the sediment flux Qs or does the material "vanish" in the model. I believe the latter is meant. If the material "vanishes", I was wondering where such a model would be applicable in nature. I had thought that even for loose, nonresistant sediment, there should be a bank height control and that transport capacity is important. As in aside, the importance of wall height, even in loose sediment seems to be implied by the later mentioned study by Bufe et al., 2016. Here, we demonstrated that in loose sediment, the width of valleys across an uplift is a function of the uplift rate (controlling the growth of valley walls) and the channel mobility (controlling the frequency at which a river revisits a given point in the valley). The area of valley that is cut across a fold reaches some equilibrium value that can be maintained and that is flanked by steep, high walls. One interpretation of this finding is that the equilibrium valley area that is actively "maintained" by the river is limited by the bank heights that the stream has to rework as it moves across the valley. For example, when a river moves from point x to point y and back to x, the bank height that has grown at point x during the time the river traveled across the valley depends on the channel mobility. The slower the migration rate of the river, the higher the walls that it encounters at x once the river returns. The observed equilibrium valley area therefore seems to imply that the wall-height and the capacity of the river to transport the material of the walls is important even in loose sediment.*

**Authors:** Changed wording in this section to clarify that in the undercutting-slump model, slumped material is transported away as wash load and not considered in Qs calculations, but we wouldnt say that it vanishes from the model, just becomes unimportant in this end member model formulation (P9L23-24). We changed text in this section to emphasize the end member models as representing valley widening as a function of wall height and moved links to natural processes to the discussion section. The background section has been expanded to include the work of Malatesta et al. 2016, who found that bank height affects the way alluvial streams erode vertically and laterally. Again, the focus here is to begin to probe how lateral erosion occurs in bedrock channels.

Following the enormous flooding on the Colorado Front Range in September 2013, we observed locations along the creek had stripped the vegetation and sediments from the base of the hillside/terrace, undercut the shale bedrock bank, and the bank slumped into the creek. Because the flood stage was so high and the shale erodes as small, flakey pieces, the slumped material was more or less immediately transported away. So the undercutting-slump model, a model formulation that describes end member behavior, is applicable in a location with an under-capacity stream and lithology that breaks down into a transportable size. We have included an example from Johnson and Finnegan (2015, GSAB) in the discussion section; they document a similar effect of shear stress-driven lateral erosion in weak mudstone.

**Commenter 1:** *Section 4.1.1: It may be clearer to introduce the concept of channel mobility and the way it is defined in this study either earlier in the paper or at the beginning of this section. At the moment, the channel mobility is defined at the beginning of the second paragraph of the section. As far as I know, the term channel mobility has mostly been used in the framework of alluvial rivers. I am guessing that the processes that limit the mobility of channels in loose sediment and in cohesive bedrock are partly different. Therefore, it would be helpful to clearly make a link here to the treatment of deposition of sediment in the model and to emphasize that in this model, any lateral movement involves bedrock erosion.*

**Authors:** Added text to clarify the distinction between channel mobility in alluvial literature and the use of the term in this paper. (P10L31-P11L2)

**Commenter 1:** *P.9 L.15 Because the treatment of deposition was not clear to me, at this point, I found it hard to wrap my head around how alpha affects channel mobility. Is it purely the effect of sediment deposition creating topography and therefore causing channel to switch more frequently? Or does sediment deposition also create an alluvial surface across which channel can migrate rapidly? After reading the end of the manuscript it became clearer that sediment, when deposited, "becomes" bedrock. Therefore, I am guessing the reason that increased sediment flux creates more mobile channels is only because sediment deposition creates "topography" that moves channels? I could imagine, that such questions could be avoided if the treatment of sediment is explained earlier.*

**Authors:** Yes, you are correct. A clearer explanation of how sediment is handled is now discussed earlier in the paper at the end of the "numerical implementation" section (P9L9-14).

**Commenter 1:** *P.11 L.18. The expression "The [...] models take [...] 10 ky to respond to lateral erosion" was a little unclear to me. What constitute a "response to lateral erosion"? Is it the time lag between the onset of the lateral erosion after the spin-up and the corresponding appearance of a signature in the topography? In which case, is there some characteristic that was used to define when the topography was thought to show a response? I am sorry if I misunderstood this...*

**Authors:** This section rewritten to make clearer: "it is not surprising that the total block models take longer to respond to the onset of lateral erosion and valleys are more narrow than

in the undercutting-slump models. The total block erosion models take on the order of 10 ky to produce an observable response to lateral erosion and ultimately produce bedrock valleys that are up to 25 meters wide, while the undercutting-slump models take about 5 ky show a response to lateral erosion and ultimately produce valleys that are up to 50 m wide."

**Commenter 1:** *P. 12 L. 20-21 Typos in this section "runs are easily", "processes due to their low relief", "has recently been shaped"*
**Authors:** Typos fixed

**Commenter 1:** *P. 12 L. 20-21: Again, I am sorry if I am misunderstanding but I would be interested in some expansion of the thoughts behind why the widest valleys occur in models with low channel mobility. I am unsure what is meant by "hillslope processes" in this context. I dont think any hillslope processes have been introduced in the model or in the introduction and theory sections of the paper. This word makes me think of landslides, hillslope creep or gullying none of these processes are in the model I believe and I am not sure I understand what is meant here.*
**Authors:** You are correct, "hillslope processes" is removed and meaning is clarified. Explanation of why the widest valleys occur in models with lower channel mobility is explained lower in the same paragraph.

**Commenter 1:** *P. 13 L. 20-22: The sequence of incision followed by lateral erosion in the TB models versus simultaneous incision and lateral erosion in the UC models would be nice to see in a figure. It is not clear from Fig. 6. In Figure 8, one panel is missing for the TB and the UC models respectively (the panel for 120 ky, just before lateral erosion starts in the TB model) to appreciate that sequence. Maybe it is possible to add one more panel and to refer to Fig. 8 at this point already? The same added panels would be nice to have at the end of this section (P. 14 L.12-19).*
**Authors:** Figures revised to show differences in sequence of lateral erosion in two model formulations.

**Commenter 1:** *P14 L24-26: I did not understand the explanation for why there is no lag time. Is the argument: 1) Lateral erosion rate is increased more than incision rate and 2) the bank height is not important in the UC models -¿ Therefore any increase in lateral erosion rate translates directly into a widening rate?*
**Authors:** The argument is the first suggestion. The text was rewritten to make this clear.

**Commenter 1:** *P14 L24-26 typo: "two times" or "two time steps"?*
**Authors:** Text changed to read "two times".

**Commenter 1:** *P.16 L. 12. I might be missing something obvious, in which case ignore the comment, but I am unsure of how to distinguish valley width formed via valley infilling or via lateral erosion from the curves of Fig. 11c: : :*
**Authors:** Interpretation of figure clarified.

**Commenter 1:** *P17 L19-26 I am glad to see a discussion about channel mobility and I was thinking about whether this discussion could benefit from a few clarifications and some restructuring. Therefore, I briefly come back to the definition of channel mobility that I mentioned above. The cited studies (Wickert et al., 2013 and Bufe et al., 2016) define channel mobility in the context of the fluvial reworking of an aggrading (or steady) alluvial surface. In my mind, this "alluvial*

*channel mobility" is not exactly equivalent to the rate at which actively uplifting valley walls are eroded  and therefore to the definition of channel mobility in this study.*

*I totally agree that one can define a channel mobility in the context of the cumulative migration metric that was used in this paper. Such a metric can be calculated and defined independently of any regard for whether rivers are migrating across a valley, across an alluvial fan, or whether rivers are eroding valley walls. However, I am unsure that there is an a-priori reason to directly, and without further explanation, use the same terminology for migration rate of alluvial channels within an alluvial valley and the lateral erosion rate of valley walls, and therefore valley width. There are of course reasons to make the link between channel mobility and valley wall erosion. Hancock and Anderson 2002 hypothesized the importance of the frequency of the contacts between the river and valley walls. Malatesta et al, 2016, and this study, demonstrate a potential importance of valley wall height. As mentioned above, if wall height is important, then channel migration rates across the active valley and the uplift rate should control valley width. This was demonstrated by Bufe et al., 2016 at least for loosely consolidated valley walls. In short, there seem to be links between the "classic, alluvial" channel mobility and the lateral erosion of valley walls but I think the link merits expanding upon before the term "channel mobility" is interchangeably in both contexts.*

**Authors:** We have clarified what we mean regarding channel mobility in our model. The commenter defines channel mobility in this model as "rate of erosion for actively uplifting valley walls", which is correct if you define valley walls as nodes immediately to next to the channel. We would say that channel mobility here is the rate of near-channel node erosion (which always occurs in bedrock here). The position of channel does not have to be next to what we define as the valley walls, the high slope nodes to either side of the channel/flat valley bottom. We think using the term channel mobility is justified, as it describes exactly what occurs in the model as well as in nature to describe bedrock valley widening. We do take your suggestion and note the differences between alluvial channel mobility and lateral channel mobility in bedrock valleys.

**Commenter 1:** *Section 5.2: Maybe here, the comparison with Hartshorn et al., 2001 and Stark et al., 2010 can be made. However, the limits of not considering tools and cover in the models might have to be discussed in more detail before that.*

**Authors:** Discussion of Hartshorn et al. 2002 and Stark et al. 2010 has been moved here (P20L23-27).

**Commenter 1:** *P.18 L.11. This paragraph discusses the model setup that links lateral erosion to channel curvature. I think it is worth noting in this context that Fuller et al., 2016 documented lateral erosion in a straight channel. As noted by the authors, the deflection of sediment (tools) toward the walls seemed to control lateral erosion in these experiments, thereby documenting the importance of tools. Because the model (and this paragraph in the paper) discusses the importance of channel curvature and because the significance of the absence of tools in the model has not been discussed, maybe the comparison with the Fuller et al., 2016 study can be moved and/or expanded upon?*

**Authors:** A significant amount of text was added here discussing Fuller et al. (2016) and the potential effects of implementing a tools and cover model for vertical incision (P20L28-P21L4).

**Commenter 1:** *P18 L 14 Typo: "has come into equilibrium"?*
**Authors:** Typo fixed.

**Commenter 1:** *P18 L23.  Maybe worth discussing - Anton, L., A. E. Mather, M. Stokes, A. Munoz- Martin, and G. De Vicente (2015), Exceptional river gorge formation from unexcep-*

*tional floods, Nature Communications, 6. This study documents knickpoint retreat and subsequent widening (maybe comparable to the TB models?) in hard bedrock.*

**Authors:** Thank you for the suggestion, this paper is briefly discussed.

**Commenter 1:** *P18 L 33 Typo: "stream power to carry"?*
**Authors:** Typo fixed

**Commenter 1:** *P19 L1-2 As mentioned before, it would be worth to discuss the treatment of sediment in more detail, and earlier in my opinion.*
**Authors:** Sediment discussed in more detail earlier in manuscript and expanded discussion of tools and cover model in the discussion.

**Commenter 1:** *Figs. 2-3: It might help to spell out the abbreviations UC, TB, and spin in the figure legend. There should be enough space. If not, it would be useful to have the definitions in the caption. It could also be helpful to add the other variable (K or alpha) to the boxes on top of the figures. For example high K, moderate alpha, high alpha, moderate K etc.*
**Authors:** Definitions for UC, TB, and spin models now included in figure caption. Text added to figure 2 for improved clarity.

**Commenter 1:** *Figure 4: The term "spinup model" could be used in panel a to more easily relate these models to the previous figure. Also, I would tend to try not having text and grid overlap. Finally, the axis labels for x and y axes may be useful*
**Authors:** Text on figure changed and labels for x and y axes added for improved clarity.

**Commenter 1:** *Fig. 5; the c-axis (slope legend) needs a label and maybe x and y axes could use labels, even though it is fairly obvious what they are*
**Authors:** Slope legend and labels for x and y axes added for improved clarity.

**Commenter 1:** *Fig. 6. The last sentence in the caption reads as if there was only waterflux from 100- 150 ky I am guessing "increased drainage area" or "increased waterflux" is meant?*
**Authors:** Text in caption changed to improve clarity.

**Commenter 1:** *Fig. 9: Maybe you can add the type of model to the title of panels a and b as well as give the actual number of K instead (or in addition to) low and medium K.*
**Authors:** Text on figure changed for improved clarity.

**Commenter 1:** *Fig. 11: Should the y axis label not be "difference in valley width"?*
**Authors:** Y axis label fixed.

---

## Author Comment (AC4) · 21 Aug 2017

**Authors' Response to AE**

**AE:** *Reviewer 1 points out the connection of lateral erosion to meandering, the minimum requirements for the modelling of which have been studied by Knutson and Howard (1984). The considerations in this paper should be discussed and incorporated in any revisions of the model.*

**Authors:** The work of Knutson and Howard (1984) and other advances in modeling meandering have been added and discussed more thoroughly. We have also added a substantial amount of text to the new "Approach and Scope" section to articulate the differences between a meander model and a landscape evolution model. In brief, the former represents the trace of a single channel whereas the latter represents the topography in which channels are embedded. This is a very important distinction, which we hope is now clear in the revised manuscript.

**AE:** *Reviewer 2 is concerned about grid resolution issues and the treatment of channel width. I think these points are very important and should be taken seriously. Id like to point out here the paper of Stark and Stark (Am. J. Sci. 2001), who developed a sub-grid approach to treat channels, which may be instructive for dealing with this criticism. A related point here is the frequency of contact of the flowing water with the banks, as made by Bufe, the importance of which has been pointed out by Hancock and Anderson (GSAB 2002).*
*- Both reviewers 1 and 2 also pointed out some previous treatments of bank erosion/ lateral mobility/meandering, for instance the above-mentioned paper by Knutson and Howard, but also the treatments within CAESAR (e.g., Coulthard and van de Wiel, ESPL 2006) and EROS (e.g., Carretier et al., ESurf 2016). A review of these treatments would be appropriate, highlighting of their different merits and why another (new) approach is necessary.*

**Authors:** We have added text regarding the issues reviewer 1 raises about channel width and grid size to the manuscript in several sections and have written a new supplementary materials section that presents the results of new model runs showing the effect of grid size on valley width and the magnitude of lateral erosion in the models. More background on previously published models with a treatment of lateral erosion was added to better place our model within the context of existing landscape evolution models and highlight the advances we have made with our model.

**AE:** *Many of these points culminate in the statement made explicitly by reviewer 1 at the end of the review. You construct some model and explore its dynamics to some extent, but the question remains as to why we should believe that it is a true or even useful description of reality. I agree here that the paper could benefit from a well-defined research hypothesis and a set of criteria that could be used to evaluate the model against field data or compare it against the performance of other available models.*

**Authors:** We have added a figure with examples of bedrock valleys and strath terraces that are much wider than their channels in several different environments, including wide valleys created by both meandering and non-sinuous rivers. This figure demonstrates qualitatively the differences between a typical narrow bedrock valley and a valley that has experienced a phase of significant lateral erosion. We have also added a significant amount of text at the end of the discussion section where we discuss different measurements and metrics needed from field or lab experiments in order to test this and future models. We also discuss a potential test of the model presented in the manuscript from a specific field site.

The approach presented here is intended to be a starting point, but not an ending point. Our main goal is to draw attention to the importance of lateral stream erosion within the context of drainage-basin evolution, and to offer some ideas for how this might be addressed in the framework of a conventional grid-based LEM.

---

## Referee Report (RR1)

IMPORTANT NOTE: all line numbers refer to the manuscript WITH tracked changes

Dear Authors, dear Editor

Thank you for giving me the chance to review the second version of this manuscript. I find that the additional sections on the scope of the model and on model limitations work well. Moreover, the supporting information are very helpful in further clarifying the model approach and in understanding the limits. Most of my comments have been clearly addressed, but, especially in the light of the comments by R1 and R2, I still see some problems that could be targeted. I hope these comments help to further improve the manuscript and look forward to seeing the next version.

With best wishes,

Aaron Bufe

**Model predictions, conclusions and limits**

Questions were raised on what can be predicted by the model and how sensitive the model is to its limitations. Now, in the supporting information, the authors have included sensitivity tests of the model to some of the most important of these limits. However, I found that these tests were not clearly referenced in the main manuscript (in particular in the section on model limits). Therefore, I would suggest to expand the limit section to include references to the supporting information and address (1) the grid size issue and the variability of results with different grid size, (2) the fact that the channel width is not explicit in the model (which is linked to the grid size issue), (3) the potential effects of the height scaling that is independent on slope, (4) the extent to which the model is scalable in space and time (see below).

Then, it might help to explicitly sum up the conclusions that can be safely made with the model and contrast these with observations and predictions of the model that should be treated with care because they are highly variable and could be dependent on the specific model limits (such as the grid size, treatment of sediment etc.). As far as I understand, (1) the model is able to show that a curvature-based erosion law in a landscape evolution model can produce wide valleys that vary as expected with bedrock erodibility; (2) the model can predict some relative differences in patterns and timing of lateral erosion that depend on the UC versus TB model – these could be summarized. Importantly, unless more sensitivity tests

are made, the effect of some of the limits remain unknown and, therefore, the model does not seem to be able to predict the absolute timing, lengths, and magnitude of the lateral erosion. This becomes important when discussing field tests of the model (see below).

**Grid size and implicit channel width**

R2 raised an issue with the grid size of the model, and the exploration in the supporting information in response to that comment is useful. However, it does not get around the complication that he stream is assumed to occupy the entire cell in all cases. This part was separately addressed by another point in the supporting information which is great. However, these two limits (varying grid size and no explicit width of the stream) are linked. As far as I understand, vertical erosion is always assumed to be uniform across the entire cell (because the stream is assumed to occupy the entire cell). Therefore, by increasing the pixel without changing anything else, it seems to me that there should be "more" vertical incision at a point with the same discharge and slope. In contrast, the lateral erosion should not be affected in the same way. Could that explain the part of the model response at higher grid sizes? In any case, the sensitivity test performed, the link between grid size and "water occupying one cell", and the corresponding uncertainties on how to interpret the sensitivity test could be addressed more clearly in the model limit section.

**Absolute time in the model**

As R2 noted, the chosen runoff values are very large and the authors explain that these are representative of peak values. In the supporting information the term "flood" was used so it seems like that there is a mechanism somewhere in the model that alternates these floods with times when the runoff is lower. However, I either missed that in the manuscript, or the mechanism by which to go from peak flow to kiloyears of model run time is not explained. How is the absolute time in the model obtained?

**Comparison with field examples**

I tend to agree with R2 that, given the variability of absolute lateral erosion rates with grid size (demonstrated in the supporting info), and the potential dependence of these parameters on other model limits, it seems like using absolute numbers of the timing, rate, and scale of lateral erosion events as an indicator for the correctness or incorrectness of the model is not possible. Therefore, it is not clear to me how it would help to have a field experiment in which the rates of lateral erosion and the exact channel geometries are known (P26 L24-27). Are there any qualitative observations or ratios of parameters that hold for any of the model

parameterizations and grid sizes? For example, are there responses of the TB or UC to changes in water flux that are consistent for all different grid sizes, or are there patterns of upstream versus downstream width that always evolve in a certain way etc.? In short, it isn't clear to me which part of the model is a key predictive part that could be compared with a field example. Could this be clarified?

**Contrast with meandering models and channel movement in the model**

R1 notes that it is not clear how streams switch directions. I believe that the key here is that the neighboring node is made to have the elevation of the downstream node. I might have missed that in the text but I couldn't find that information in the manuscript (apart from the figure). I would suggest to mention this point. Then, once the lateral node is eroded, there are, in some cases, two possible paths. I would guess the path is chosen at random but I am not sure whether that is mentioned anywhere. Mentioning these two points in the manuscript might help clarify how channels move in the model.

R1 further notes that the Howard and Knutson meandering model could not explain meandering with only a curvature-based law. Within the LEM, lateral erosion is obtained with just this simple law. Maybe this difference could be emphasized in the results.

**Deducing valley widening due to lateral erosion**

Figure 12 and P19L15-17: The argument here is that the valley widening in TB1 is only due to sediment infilling. Therefore, the difference between the other models and TB1 is the component of the widening due to lateral erosion. I think, this is not strictly true. For example, everything else being equal, the same amount of sediment deposited in a narrow valley and in a wider valley should lead to a larger "widening" in the narrow valley. Therefore, as valleys are widened by lateral erosion, the sediment aggradation component becomes "less important". Moreover, the observed widening will depend on the slopes of the valley walls. The difference between all models and TB1 can still serve as a proxy but, perhaps, this limit should be addressed.

**Treatment of deposition in the model**

In the supporting material (page 1, point 3, subpoint 5), it seems unclear whether eroded material is added to Qs or immediately deposited as bedrock downstream. The former makes the most sense to me. Can this be made clearer in the supporting info and in the main text?

This raises another point. It seems like the eroded material is added to Qs even in the UC models – only the "collapsed" material is removed as washload. Isn't that inconsistent with the premise of the UC models that all eroded material can be easily transported away?

**Detachment versus transport and the definition of "weak" and "strong" bedrock**

P22L1210-14: I wonder if it is worth to specify what is meant by "weak bedrock"? One could have loose sediment (for example loose sand or gravel) that is easy to detach but when you detach it, it has to be transported away as part of the bedload. On the contrary, one could have consolidated clays that need to be detached, but once they are, they will wash down as washload. I think, somewhere at the beginning of the manuscript (perhaps under "scope") there should be a definition of the term "weak bedrock" and "strong bedrock" as used in this paper and perhaps a brief clarification of the simplification of the detachment and the transport components.

**Structure of the introduction and scope sections**

Large parts of the Scope section seem to introduce previous literature and partly repeat points that have been made in the introduction. Perhaps sections 1-2 could be restructured into three different sections: Section 1: a short, focused introduction, Section 2: A detailed background of existing model approaches and limits (including the differences between LEMs and channel-scale physics models), and Section 3: The modeling approach and scope of the model.

**Minor line comments**

P1L23: "wide bedrock valleys *in* incising rivers" – seems odd

P2L1: can the term "virtual velocity of sediment" be used without explanation? It is not a word that I intuitively associate with but maybe that is my problem.

P3L1: Mention that Howard and Knutson's model was developed for an alluvial river

P3L3: "scales inversely with **the** radius of curvature"

P3L7: Suggest change to "at reach and small catchment scales and **at** time scales **of** up to […]".

P3L23: "Lateral migration […]" this sentence seems a bit out of place in the middle of the discussion of different strath terrace formation models. Maybe move this? Also, I think, this needs references.

P7:L18: Typo in equation

P7L20: $K_l$ is mentioned here but it wasn't defined yet.

P8L8: "We hypothesize […]" – you can mention the meandering models here again. Something like "Consistent with previous meandering models (references), we hypothesize xxx.

P10L10: I think "greater" should be "greatest"

P10L35: for two occurrences, need: "of streams that **are**"

P11L28: "bedrock channels **is** less clear"

P12L3-5: This sentence is odd. Suggesting: "Water flux was introduced **at** the top of the model by designating a node as an inlet with an area of 20,000 m$^2$**, and at this node,** sediment flux was introduced at carrying capacity**. This setup allowed each run to have** a primary channel […]".

P13L24: Comma missing "Most often**,** the"

P14L19: I would rephrase. "A wide valley implies that significant lateral erosion has occurred **relative to vertical incision**".

P14L28: Please specify the direction in which this slope is calculated. Down channel or perpendicular to the channel?

P16L20: "In order for this **model** to be"

P16L28: "to an event such as **a** stream capture"

P17L12: The reference to Figure 10 is out of sequence.

P17L21: I suggest to specify the timescale over which the valleys persist. When I read it, I thought that valleys are thought to persists for the remainder of the model but it is only a few 10s of ky.

P18L2-9: I believe that I mentioned that in my comments from last time. I think, in order to see the described sequence of event in the figures, there are some time periods missing from the figure. Moreover, I am unsure, why the UC and TB models compared here are with two different ratios of Kv/Kl. Now, we have two variables that change, the erosion model and the ratio of vertical-to-lateral. This makes it harder to compare the effect of the two erosion models.

P19 Line 2 (when using the line number 5 as reference), Line 4 when counting from the top of the page: In addition to the model only responding by changes in width, I would also remind the reader that the sediment deposition is producing bedrock.

P19L30: I would suggest to say "**differential** bedrock valley width" because the figure doesn't show the absolute valley width

P19L34: Same as above, "variability in **differential** valley width"

P20L11: Just after discussing the differential valley widths shown in Fig 12, I wasn't sure whether the term "mean valley width" now referred to absolute values or the differential values. A reference to Figure 11 could resolve this issue.

P20L16: to better qualify "flat" give the range of slope values. Moreover, is the downstream slope meant or is the slope of the valley perpendicular to the stream referred to?

P21L13: The way the sentence is phrased, it seems to assume an equivalence between landscapes of weak bedrock and landscapes of low relief. This equivalence has not been established before. Could this be expanded?

P21L23 valley**s**

P22L1 (counted from top of page): I am not sure about the term "blocky material" in this context. What does that mean? What about blocky but very easily eroded material? Maybe consider removing this term or expanding on it?

P22L6-7 (counted from top of page): the switch to "terraces" here seems fairly specific – especially in this mode that does not distinguish between bedrock and sediment. I am unsure I understand what the sentence is meant to convey

P22L8 (counted from line number 5): "lateral erosion **of** a bank that has been laterally undercut and **where** the remaining material"

P22L10: It wasn't clear to me until I read the supplement that all sediment produced by lateral erosion is not just added to Qs but immediately deposited downstream. Therefore, I wasn't sure about the statement "not redeposited in the model" – maybe that deposition of material can be clarified earlier in the manuscript.

P22L26: The comparison between bedrock valleys that are either "several times" (in the model) or "many times" (in some natural examples) the width of the channel seems vague. Can you specify how many times? Perhaps 3-5-times (in the model) or up to 100 times (in the real world)?

P22L28-29: The sentence "The model also did not show […]" directly contradicts the conclusion statement on P 27L21: "Increased channel mobility […]". In general, I think it can be confusing to speak of bedrock shielded by sediment in this model that does not treat sediment and bedrock separately.

P23L7 (counted from line number 5): "**an** important next step"

P23L19: I suggest changing to "**One** main impact". It is not clear that the changes in vertical incision are the main impact. In a study that should be accepted shortly, we demonstrate that some combination of autogenic dynamics and changes in water and sediment fluxes can cause order of magnitude changes in the rate of lateral erosion with only small changes in vertical incision.

P23L24: Suggest "needs to be **constructed**".

P23L25: For a proposed mechanism for how sediment cover can change lateral erosion rates See Turowski's paper in Earth Surface Dynamics Discussion "Alluvial cover controlling the width, slope and sinuosity of bedrock channels"

P23L29: I believe this should be "erosion **minus** deposition"

P23L32: I do not understand how a fixed kw is appropriate for landscapes in "quasi-equilibrium". First, I am not sure what is meant by a "quasi-equilibrium", second, I am not sure if the model that is proposed here looks at landscapes in equilibrium (or quasi-equilibrium), especially when water and sediment fluxes are changed.

P24L28: I suggest to weaken the statement by saying "it **appears to be** an important one" because there is no clear proof put forward here (or did I misunderstand something?)

P25L14: "aggradation **in** the high alpha"

P25L18-19: The sentence "does not lead to increased sediment cover on the bed […] but […] results in […] channel aggradation" is a bit odd when the process that is commonly referred to as "channel aggradation" will lead to "increased sediment cover on the bed". Maybe rephrase.

P25L21: It would be good to specify whether a "relative" (as in relative to vertical incision) or an "absolute" increase in lateral erosion rates is referred to.

P26L11: Shouldn't this be the "channel-scale" rather than the landscape scale?

P26L24: "A challenge remains **in** how"

P26L24: I believe, the sentence starting with "The robust data set […] are" has to be rephrased.

P26L31: valley**s**

P27L1-2 (counted from top of the page): The sentence "nor have we identified an appropriate natural experiment" seems to contradict the following paragraph where a natural experiment that is apparently deemed appropriate is described.

P27L17: "channel equilibrium" is not unambiguous in my mind. Maybe say something like "until the slopeof the channel is adjusted to the new sediment and water input conditions"

P27L21-23: Again, I find it odd to talk about implications of sediment cover of the bed in a model that does not treat sediment differently from bedrock. The observation that can be made in the model is that "when the bed is aggraded, we do not incise"

**Figure comments**

Figure 1: I suggest to:

- Enlarge the labels (especially the axes labels on the cross section)

- Indicate north-south, or northeast southwest etc. on the cross section

- Change the color of the title in panel b – the black Is hard to see

- Make the cross section on panel b thicker and the labels larger (could be done on all panels but on panel b it is especially hard to see

- Make the north arrow larger and underlain with a white box

- Indicate the wetted area on the cross sections (for example, make the topography black and then the wetted area in blue – with thick lines)

- Change the color of the outlined strath terraces in panel c (for example to white) – these are extremely hard to see

- Possibly change the cross section labels according to the panel letter (A-A', B-B' etc.)

Figure 1 Caption

- Suggest to change first sentence to: "Field examples of wide bedrock valleys cut by lateral erosion"

- It is not clear where the cross sections and images are from. I presume from Google Earth – in that case, I suggest indicating that the images are from that source and that the cross sections are based on the 90m-srtm DEM from Google Earth.

- I suggest to make labels on the cross section with arrows to the parts of the cross section that are referred to in the figure caption (e.g. the table mountain and the 10-m terrace in panel c).

Figure 2 I suggest to

- Annotate the black arrow on the figure "height that has to be eroded"

- Note in panel d that the slumped material is transported as washload

Figure 2 caption

- Line 3: "H, is shown **by** the dashed".

- Line 5: Could say "black double arrow" to make clear which arrow is meant

Figure 3:

- Panel letters are missing in the figure

- Y axis is missing in the upper panels

- Legend could be made bigger (UC model, TB model)

Figure 4

- I would suggest to reorganize the panels so that a and b are next to each other as in all other figures in the paper. That will also make the y and x axes the same in each line and column

- One could consider to spell out the parameter λ to help readers that are quickly glancing over the figures and don't carefully read the entire manuscript.

Figure 5

- The z axis could be labeled
- I know this is nitpicky, but I still think the figure titles could be made so that they do not overlap with the grid. That would make the figure look cleaner.

Figure 7

- There is enough space to label the zone of increased waterflux on the figure panel
- This zone could also be shaded in light grey or light green to make clear that it is a zone
- The same could be done in Figures 11 and 12 in which the shading would go until the end of the experiment.

Figure 8

- On panel b, the legend could be moved down to not overlap with data

**Supporting information comments**

Page 1 point 5 – subpoint 1: I think it should read "The volume of sediment […] so that it's elevation is equal to the **downstream**, node".

Page 7 Paragraph 4 (last paragraph), L6: "valley width **is** generally increased".

Page 7 paragraph 4 L13: delete "carved"

Page 9 L4: "actual valley width **that** emerges"

Figure 2

- In panel a, I would suggest to shade the two possible lateral nodes red and add a label on the figure that indicates that these two nodes are chosen at random

Figure 5

- I got confused why the model with the spinup (panel a) shows lateral erosion right at t=0 whereas the other model (panel b) shows lateral erosion only after c.a. t=200. Maybe expand upon that
- The time is missing units (I presume y or ky?)

Figure 7

- I note that the direction of the y axis (decreasing values downstream) is inconsistent with the direction of equivalent figures in the main manuscript. That led to a short confusion

Figure 8

- Same comment with the shading and labeling the zone of increased water flux as in the main manuscript

Figure 9

- The axes of panel c and f could probably be changed so that all data is included on the figure.
- Same comment with the shading and labeling the zone of increased water flux as in the main manuscript

Figure 10

- Why is t = 0 and t = 25ky here whereas all other models (Figs 11-12) the time steps shown are t=50 and t=75?

---

## Author Response (AR2)

Associate Editor Decision: Publish subject to minor revisions (review by Editor) (16 Oct 2017) by Jens Turowski

Comments to the Author:

Dear authors,

thank you for the revised version of your paper. I was lucky to be able to solicit two of the reviewers from the first round (Bufe and Lague) to look at the paper again. Both reviewers agree that the paper has much improved and the addition that you provided are seen as useful. Nevertheless, there are a number of open questions. First, some of the material that you added to the supplement is not or only insufficiently picked up in the main manuscript. I would like to see a more open treatment of the shortcomings of the model that have been highlighted by the reviewers (e.g., grid resolution issues, hydraulics) within the discussion. I agree with Lague that there is a discrepancy with the attention to details in the physical modelling of the erosion process, while hydraulics and flow routing are treated in a rather rough way. Bufe also gives a lot of details on a number of points that the authors need to deal with in a more systematic manner than is currently done.

One aspect that seems important to me is the testability of predictions. The relevant section in the manuscript leaves me wanting. The observation that lateral erosion is in some way dependent on curvature (as has been observed in the cited study of Cook et al.) is not a good model test, since erosion rates driven by other mechanisms such as impact erosion would also depend on curvature. Can you come up with clear model predictions that go beyond the assumptions put into model construction? A number of the results presented in the paper could be used to construct such tests. This could be for example the location on size or the sinuosity of the valleys produced by the erosion model.

Although there might be a substantial amount of work, neither of the two reviewers has asked for extensive new analyses and most of what needs to be done is in writing and presentation. I have therefore chosen 'minor revisions'. Please submit a revised version of the paper and a rebuttal dealing in detail with all the reviewers' comments. If these are convincing, I will not send out the paper for review again.

Good luck with revisions. I am looking forward to seeing the new version of the paper.

Best, Jens

***Authors' Response:*** *In response to comments from both reviewers and the AE, we added text that draw the reader's attention to the model limitations we highlight in the supplementary materials section and note some of the significant findings. We added two paragraphs at the beginning of section 7.2 (formerly*

*section 6.2) that (1) highlight how the simplified hydrological assumptions may affect model outcomes when implied channel width approaches or is greater than the cell size and (2) discuss the effects of grid resolution on lateral erosion and valley width.*

*We agree that it is important to explicitly state where this model succeeds in reproducing wide valleys found in natural systems and where it fails. Therefore, it is useful to discuss generalized model results that can be compared to field metrics and ultimately determine how well this model describes lateral bedrock erosion and valley widening in nature. We have added text to the reorganized and edited field section that points to a scaling relationship between valley width and drainage area predicted by our model and also observed in natural systems.*

*We have also addressed the detailed comments made by reviewer Bufe and made many of the changes he suggested. These comments have improved the readability and impact of the figures and is very much appreciated.*

*Authors' Response to Reviewer 1*

IMPORTANT NOTE: all line numbers refer to the manuscript WITH tracked changes

Dear Authors, dear Editor

Thank you for giving me the chance to review the second version of this manuscript. I find that the additional sections on the scope of the model and on model limitations work well. Moreover, the supporting information are very helpful in further clarifying the model approach and in understanding the limits. Most of my comments have been clearly addressed, but, especially in the light of the comments by R1 and R2, I still see some problems that could be targeted. I hope these comments help to further improve the manuscript and look forward to seeing the next version.

With best wishes,

Aaron Bufe

*Authors' Response: We thank you for your thoughtful and thorough review. While making the changes has been a substantial amount of work, your suggestions and comments serve to make the manuscript much improved.*

▪▪▪▪▪▪▪▪▪▪▪▪▪▪▪▪▪▪▪▪▪▪▪▪▪▪▪▪▪▪▪▪▪▪▪▪▪▪▪▪▪▪▪▪▪▪▪▪▪▪▪▪▪▪▪▪▪▪▪▪▪▪▪▪

**Model predictions, conclusions and limits**

Questions were raised on what can be predicted by the model and how sensitive the model is to its limitations. Now, in the supporting information, the authors have included sensitivity tests of the model to some of the most important of these limits. However, I found that these tests were not clearly referenced in the main manuscript (in particular in the section on model limits). Therefore, I would suggest expanding the limit section to include references to the supporting information and address (1) the grid size issue and the variability of results with different grid size, (2) the fact that the channel width is not explicit in the model (which is linked to the grid size issue), (3) the potential effects of the height scaling that is independent on slope, (4) the extent to which the model is scalable in space and time (see below).

*Authors' Response: Another reviewer also requested that the information presented in the supplementary materials section be included in the main manuscript. We have added new text that draw the reader's attention to the model limitations we highlight in the supplementary materials section and note some of the significant findings. We added two paragraphs at the beginning of section 7.2 (formerly section 6.2) that (1) highlight how the simplified hydrological assumptions may affect model outcomes when implied channel width approaches or is greater than the cell size and (2) discuss the effects of grid resolution on lateral erosion and valley width.*

Then, it might help to explicitly sum up the conclusions that can be safely made with the model and contrast these with observations and predictions of the model that should be treated with care because they are highly variable and could be dependent on the specific model limits (such as the grid size, treatment of sediment etc.). As far as I understand, (1) the model is able to show that a curvature-based erosion law in a landscape evolution model can produce wide valleys that vary as expected with bedrock erodibility; (2) the model can predict some relative differences in patterns and timing of lateral erosion that depend on the UC versus TB model – these could be summarized. Importantly, unless more sensitivity tests are made, the effect of some of the limits remain unknown and, therefore, the model does not seem to be able to predict the absolute timing, lengths, and magnitude of the lateral erosion.

*Authors' Response: We took your suggestion to more clearly highlight the most important findings and outcomes from this study in the conclusions section and rewrote a significant portion of the conclusions.*

**Grid size and implicit channel width**

R2 raised an issue with the grid size of the model, and the exploration in the supporting information in response to that comment is useful. However, it does not get around the complication that the stream is assumed to occupy the entire cell in all cases. This part was separately addressed by another point in the supporting information which is great.

However, these two limits (varying grid size and no explicit width of the stream) are linked. As far as I understand, vertical erosion is always assumed to be uniform across the entire cell (because the stream is assumed to occupy the entire cell). Therefore, by increasing the pixel without changing anything else, it seems to me that there should be "more" vertical incision at a point with the same discharge and slope. In contrast, the lateral erosion should not be affected in the same way. Could that explain the part of the model response at higher grid sizes?

In any case, the sensitivity test performed, the link between grid size and "water occupying one cell", and the corresponding uncertainties on how to interpret the sensitivity test could be addressed more clearly in the model limit section.

*Authors' Response: We have now added text that draw the reader's attention to the model limitations we highlight in the supplementary materials section and note some of the significant findings. We added two paragraphs at the beginning of section 7.2 (formerly section 6.2) that (1) highlight how the simplified hydrological assumptions may affect model outcomes when implied channel width approaches or is greater than the cell size and (2) discuss the effects of grid resolution on lateral erosion and valley width.*

**Absolute time in the model**

As R2 noted, the chosen runoff values are very large and the authors explain that these are representative of peak values. In the supporting information the term "flood" was used so it seems like that there is a mechanism somewhere in the model that alternates these floods with times when the runoff is lower. However, I either missed that in the manuscript, or the mechanism by which to go from

peak flow to kiloyears of model run time is not explained. How is the absolute time in the model obtained?

*Authors' Response: There is no mechanism in the model for changing the value of runoff. The storm duration (not flood) mentioned in the supplementary materials is the time step over which erosion and deposition are calculated. This was clarified in the supplementary materials. In all of the model runs, the storm duration was 10 years. Absolute time in the model is storm duration\*number of model time steps and the model runs range from 100 ky – 200 ky. We are aware that these model times in the model presented here should not be conflated with absolute timing of bedrock valley development.*

**Comparison with field examples**

I tend to agree with R2 that, given the variability of absolute lateral erosion rates with grid size (demonstrated in the supporting info), and the potential dependence of these parameters on other model limits, it seems like using absolute numbers of the timing, rate, and scale of lateral erosion events as an indicator for the correctness or incorrectness of the model is not possible.

Therefore, it is not clear to me how it would help to have a field experiment in which the rates of lateral erosion and the exact channel geometries are known (P26 L24-27). Are there any qualitative observations or ratios of parameters that hold for any of the model parameterizations and grid sizes? For example, are there responses of the TB or UC to changes in water flux that are consistent for all different grid sizes, or are there patterns of upstream versus downstream width that always evolve in a certain way etc.? In short, it isn't clear to me which part of the model is a key predictive part that could be compared with a field example. Could this be clarified?

*Authors' Response: Thank you for bringing up this important point. To address it, we have added a new figure of a valley width vs. drainage area that shows a scaling relationship that can be used as a model prediction that can be compared to wide valleys found in natural systems. We have also added three new paragraphs that discuss this model prediction and mention what model results should be interpreted with caution.*

**Contrast with meandering models and channel movement in the model**

R1 notes that it is not clear how streams switch directions. I believe that the key here is that the neighboring node is made to have the elevation of the downstream node. I might have missed that in the text but I couldn't find that information in the manuscript (apart from the figure). I would suggest to mention this point. Then, once the lateral node is eroded, there are, in some cases, two possible paths. I would guess the path is chosen at random but I am not sure whether that is mentioned anywhere. Mentioning these two points in the manuscript might help clarify how channels move in the model.

R1 further notes that the Howard and Knutson meandering model could not explain meandering with only a curvature-based law. Within the LEM, lateral erosion is obtained with just this simple law. Maybe this difference could be emphasized in the results.

***Authors' Response:*** *Added text in the Numerical implementation section to clarify how the stream switches directions. After a node is eroded laterally, the water path is not chosen randomly, but is rerouted by a D8 flow director.*

**Deducing valley widening due to lateral erosion**

Figure 12 and P19L15-17: The argument here is that the valley widening in TB1 is only due to sediment infilling. Therefore, the difference between the other models and TB1 is the component of the widening due to lateral erosion. I think, this is not strictly true. For example, everything else being equal, the same amount of sediment deposited in a narrow valley and in a wider valley should lead to a larger "widening" in the narrow valley. Therefore, as valleys are widened by lateral erosion, the sediment aggradation component becomes "less important". Moreover, the observed widening will depend on the slopes of the valley walls. The difference between all models and TB1 can still serve as a proxy but, perhaps, this limit should be addressed.

***Authors' Response:*** *We chose to remove figure 12 as we felt it did not add much substance to the paper.*

**Treatment of deposition in the model**

In the supporting material (page 1, point 3, subpoint 5), it seems unclear whether eroded material is added to Qs or immediately deposited as bedrock downstream. The former makes the most sense to me. Can this be made clearer in the supporting info and in the main text?

This raises another point. It seems like the eroded material is added to Qs even in the UC models – only the "collapsed" material is removed as washload. Isn't that inconsistent with the premise of the UC models that all eroded material can be easily transported away?

***Authors' Response:*** *Explanation of the treatment of the laterally eroded material is included in the manuscript in the "Numerical implementation" section, at the end of the fourth paragraph. Treatment of laterally eroded sediment has been clarified in the supplementary materials. As for the second point regarding the treatment of sediment in the UC models, yes, you could say that sending sediment downstream with Qs in the UC model is inconsistent with the premise that all eroded sediment is easily transported as wash load in a strictly physical sense. But given the insightful comments from the reviewers, we hesitate to assign a process-based mechanism and justification to what is essentially an end-member algorithm for changing the elevation of lateral node cells.*

**Detachment versus transport and the definition of "weak" and "strong" bedrock**

P22L1210-14: I wonder if it is worth to specify what is meant by "weak bedrock"? One could have loose sediment (for example loose sand or gravel) that is easy to detach but when you detach it, it has to be transported away as part of the bedload. On the contrary, one could have consolidated clays that need to be detached, but once they are, they will wash down as washload. I think, somewhere at the beginning of the manuscript (perhaps under "scope") there should be a definition of the term "weak bedrock" and "strong bedrock" as used in this paper and perhaps a brief clarification of the simplification of the detachment and the transport components.

***Authors' Response:*** *Clarified sentence on P22L10-14 to specify what is meant by weak bedrock, but opted not to add a definition earlier in the manuscript based on recommendations from other reviewers suggesting that interpretation of physical processes be reserved for the discussion section.*

**Structure of the introduction and scope sections**

Large parts of the Scope section seem to introduce previous literature and partly repeat points that have been made in the introduction. Perhaps sections 1-2 could be restructured into three different sections: Section 1: a short, focused introduction, Section 2: A detailed background of existing model approaches and limits (including the differences between LEMs and channel-scale physics models), and Section 3: The modeling approach and scope of the model.

***Authors' Response:*** *We took your suggestion and restructured the Introduction and Approach and Scope sections into three sections and removed some redundant text.*

**Minor line comments**

P1L23: "wide bedrock valleys in incising rivers" – seems odd

***Authors' Response:*** *Changed text to read "wide bedrock valleys created by incising rivers" to distinguish from bedrock valleys created by aggrading rivers.*

P2L1: can the term "virtual velocity of sediment" be used without explanation? It is not a word that I intuitively associate with but maybe that is my problem.

***Authors' Response:*** *Removed "virtual velocity" and replaced with "average transport velocity of sediment grains"*

P3L1: Mention that Howard and Knutson's model was developed for an alluvial river

***Authors' Response:*** *Changed text to clarify Howard and Knutson's model was developed for an alluvial river.*

P3L3: "scales inversely with the radius of curvature"

***Authors' Response:*** *added text*

P3L7: Suggest change to "at reach and small catchment scales and at time scales of up to […]".

***Authors' Response:*** *added text*

P3L23: "Lateral migration […]" this sentence seems a bit out of place in the middle of the discussion of different strath terrace formation models. Maybe move this? Also, I think, this needs references.

***Authors' Response:*** *Deleted this sentence.*

P7:L18: Typo in equation

***Authors' Response:*** *Fixed typo in equation*

P7L20: Kl is mentioned here but it wasn't defined yet.

***Authors' Response:*** *added text referring readers to text below for explanation of Kv and Kl.*

P8L8: "We hypothesize […]" – you can mention the meandering models here again. Something like "Consistent with previous meandering models (references), we hypothesize xxx.

***Authors' Response:*** *added suggested text.*

P10L10: I think "greater" should be "greatest"

***Authors' Response:*** *Changed text*

P10L35: for two occurrences, need: "of streams that are"

***Authors' Response:*** *added "are" in two places*

P11L28: "bedrock channels is less clear"

***Authors' Response:*** *added text*

P12L3-5: This sentence is odd. Suggesting: "Water flux was introduced at the top of the model by designating a node as an inlet with an area of 20,000 m2, and at this node, sediment flux was introduced at carrying capacity. This setup allowed each run to have a primary channel […]".

***Authors' Response:*** *Changed text as suggested.*

P13L24: Comma missing "Most often, the"

***Authors' Response:*** *added comma.*

P14L19: I would rephrase. "A wide valley implies that significant lateral erosion has occurred relative to vertical incision".

***Authors' Response:*** *Text revised as suggested.*

P14L28: Please specify the direction in which this slope is calculated. Down channel or perpendicular to the channel?

***Authors' Response:*** *Changed text to clarify this is slope perpendicular to channel.*

P16L20: "In order for this model to be"

***Authors' Response:*** *Changed text as suggested.*

P16L28: "to an event such as a stream capture"

*Authors' Response: Did not change text here.*

P17L12: The reference to Figure 10 is out of sequence.

*Authors' Response: Removed reference to figure 10 here.*

P17L21: I suggest to specify the timescale over which the valleys persist. When I read it, I thought that valleys are thought to persists for the remainder of the model but it is only a few 10s of ky.

*Authors' Response: Added text to clarify.*

P18L2-9: I believe that I mentioned that in my comments from last time. I think, in order to see the described sequence of event in the figures, there are some time periods missing from the figure. Moreover, I am unsure, why the UC and TB models compared here are with two different ratios of Kv/Kl. Now, we have two variables that change, the erosion model and the ratio of vertical-to-lateral. This makes it harder to compare the effect of the two erosion models.

*Authors' Response: We added a panel showing an intermediate time step for both the total block erosion model and undercutting slump model and split this figure into two figures. We did not do this last time as we felt figure 10 showed the same pattern. But we have also removed figure 10 from the manuscript as it did not add much.*

P19 Line 2 (when using the line number 5 as reference), Line 4 when counting from the top of the page: In addition to the model only responding by changes in width, I would also remind the reader that the sediment deposition is producing bedrock.

*Authors' Response: Added text: "In this model, no distinction is made between the erodibility of deposited material and bedrock; any deposited material in the model has the properties of bedrock rather than sediment."*

P19L30: I would suggest to say "differential bedrock valley width" because the figure doesn't show the absolute valley width

P19L34: Same as above, "variability in differential valley width"

P20L11: Just after discussing the differential valley widths shown in Fig 12, I wasn't sure whether the term "mean valley width" now referred to absolute values or the differential values. A reference to Figure 11 could resolve this issue.

*Authors' Response: We chose to remove figure 12.*

P20L16: to better qualify "flat" give the range of slope values. Moreover, is the downstream slope meant or is the slope of the valley perpendicular to the stream referred to?

*Authors' Response: Changed text to clarify.*

P21L13: The way the sentence is phrased, it seems to assume an equivalence between landscapes of weak bedrock and landscapes of low relief. This equivalence has not been established before. Could this be expanded?

*Authors' Response: Added reference to Whipple and Tucker, 1999 to establish link between low relief landscapes in weaker lithologies.*

P21L23 valleys

*Authors' Response: Changed text.*

P22L1 (counted from top of page): I am not sure about the term "blocky material" in this context. What does that mean? What about blocky but very easily eroded material? Maybe consider removing this term or expanding on it?

*Authors' Response: Added text to clarify meaning of phrase.*

P22L6-7 (counted from top of page): the switch to "terraces" here seems fairly specific – especially in this mode that does not distinguish between bedrock and sediment. I am unsure I understand what the sentence is meant to convey

*Authors' Response: Text changed to clarify meaning.*

P22L8 (counted from line number 5): "lateral erosion of a bank that has been laterally undercut and where the remaining material"

*Authors' Response: No change made.*

P22L10: It wasn't clear to me until I read the supplement that all sediment produced by lateral erosion is not just added to Qs but immediately deposited downstream. Therefore, I wasn't sure about the statement "not redeposited in the model" – maybe that deposition of material can be clarified earlier in the manuscript.

*Authors' Response: We have changed text in the supplementary materials section to clarify this point. Material eroded from the lateral nodes is sent to the Qs term and is not automatically deposited downstream. The line referenced here indicates that after the lateral node has been fully undercut, the overlying material is removed from the model as wash load, i.e. not redeposited or added to the Qs term.*

P22L26: The comparison between bedrock valleys that are either "several times" (in the model) or "many times" (in some natural examples) the width of the channel seems vague. Can you specify how many times? Perhaps 3-5-times (in the model) or up to 100 times (in the real world)?

*Authors' Response: Suggested change made.*

P22L28-29: The sentence "The model also did not show […]" directly contradicts the conclusion statement on P 27L21: "Increased channel mobility […]". In general, I think it can be confusing to speak of bedrock shielded by sediment in this model that does not treat sediment and bedrock separately.

*Authors' Response:* *Changed text to clarify that we did not see the expected change in lateral erosion with a step increase in sediment flux.*

P23L7 (counted from line number 5): "an important next step"

*Authors' Response:* *Changed text*

P23L19: I suggest changing to "One main impact". It is not clear that the changes in vertical incision are the main impact. In a study that should be accepted shortly, we demonstrate that some combination of autogenic dynamics and changes in water and sediment fluxes can cause order of magnitude changes in the rate of lateral erosion with only small changes in vertical incision.

*Authors' Response:* *No change made here. We wrote this section with how our model will change when a sediment flux-dependent incision rule is applied instead of the stream power erosion rule. We look forward to reading the new study you mention, but don't feel we can rely on information that essentially word of mouth at this point.*

P23L24: Suggest "needs to be constructed".

*Authors' Response:* *Change made.*

P23L25: For a proposed mechanism for how sediment cover can change lateral erosion rates See Turowski's paper in Earth Surface Dynamics Discussion "Alluvial cover controlling the width, slope and sinuosity of bedrock channels"

*Authors' Response:* *Thank you for alerting us to this new paper. We will look to this paper for inspiration for future iterations of our lateral erosion work.*

P23L29: I believe this should be "erosion minus deposition"

*Authors' Response:* *Change made.*

P23L32: I do not understand how a fixed kw is appropriate for landscapes in "quasi-equilibrium". First, I am not sure what is meant by a "quasi-equilibrium", second, I am not sure if the model that is proposed here looks at landscapes in equilibrium (or quasi-equilibrium), especially when water and sediment fluxes are changed.

*Authors' Response:* *We use the term quasi equilibrium to mean that spatially averaged erosion in the model is equal to uplift. Erosion may not be equal to uplift at each model cell at every time in the model. While the models are responding to changes in sediment or water fluxes, they are not in equilibrium, but this study is not aimed at exploring changing channel width coefficients in transient states. The line of text you refer to is there to explain why our model behaves as it does and does not change width although channels do in natural systems.*

P24L28: I suggest to weaken the statement by saying "it appears to be an important one" because there is no clear proof put forward here (or did I misunderstand something?)

*Authors' Response: Change made.*

P25L14: "aggradation in the high alpha"

*Authors' Response: Change made.*

P25L18-19: The sentence "does not lead to increased sediment cover on the bed […] but […] results in […] channel aggradation" is a bit odd when the process that is commonly referred to as "channel aggradation" will lead to "increased sediment cover on the bed". Maybe rephrase.

*Authors' Response: Sentence rephrased to clarify.*

P25L21: It would be good to specify whether a "relative" (as in relative to vertical incision) or an "absolute" increase in lateral erosion rates is referred to.

*Authors' Response: Change made.*

P26L11: Shouldn't this be the "channel-scale" rather than the landscape scale?

*Changed text to read "reach-scale drivers".*

P26L24: "A challenge remains in how"

*Authors' Response: Change made.*

P26L24: I believe, the sentence starting with "The robust data set […] are" has to be rephrased.

*Authors' Response: Text in this section was rewritten.*

P26L31: valleys

*Authors' Response: Change made.*

P27L1-2 (counted from top of the page): The sentence "nor have we identified an appropriate natural experiment" seems to contradict the following paragraph where a natural experiment that is apparently deemed appropriate is described.

*Text changed in this section and apparent contradiction resolved.*

P27L17: "channel equilibrium" is not unambiguous in my mind. Maybe say something like "until the slopeof the channel is adjusted to the new sediment and water input conditions"

*Authors' Response: Change made.*

P27L21-23: Again, I find it odd to talk about implications of sediment cover of the bed in a model that does not treat sediment differently from bedrock. The observation that can be made in the model is that "when the bed is aggraded, we do not incise"

***Authors' Response:*** *We take your point about overinterpreting the model. Text changed here.*

**Figure comments**

Figure 1: I suggest to:

• Enlarge the labels (especially the axes labels on the cross section)

• Indicate north-south, or northeast southwest etc. on the cross section

• Change the color of the title in panel b – the black Is hard to see

• Make the cross section on panel b thicker and the labels larger (could be done on all panels but on panel b it is especially hard to see

• Make the north arrow larger and underlain with a white box

• Indicate the wetted area on the cross sections (for example, make the topography black and then the wetted area in blue – with thick lines)

• Change the color of the outlined strath terraces in panel c (for example to white) – these are extremely hard to see

• Possibly change the cross section labels according to the panel letter (A-A', B-B' etc.)

Figure 1 Caption

• Suggest to change first sentence to: "Field examples of wide bedrock valleys cut by lateral erosion"

• It is not clear where the cross sections and images are from. I presume from Google Earth – in that case, I suggest indicating that the images are from that source and that the cross sections are based on the 90m-srtm DEM from Google Earth.

• I suggest to make labels on the cross section with arrows to the parts of the cross section that are referred to in the figure caption (e.g. the table mountain and the 10-m terrace in panel c).

***Authors' Response:*** *Suggested changes made in figure 1.*

Figure 2 I suggest to

• Annotate the black arrow on the figure "height that has to be eroded"

• Note in panel d that the slumped material is transported as washload

***Authors' Response:*** *Did not make these changes, as another reviewer found the figure too crowded and confusing. Changes made in the figure caption.*

Figure 2 caption

• Line 3: "H, is shown by the dashed".

• Line 5: Could say "black double arrow" to make clear which arrow is meant

Figure 3:

• Panel letters are missing in the figure

• Y axis is missing in the upper panels

• Legend could be made bigger (UC model, TB model)

***Authors' Response:*** *Changes made.*

Figure 4

• I would suggest to reorganize the panels so that a and b are next to each other as in all other figures in the paper. That will also make the y and x axes the same in each line and column

***Authors' Response:*** *Change made*

• One could consider to spell out the parameter λ to help readers that are quickly glancing over the figures and don't carefully read the entire manuscript.

***Authors' Response:*** *Change not made as the figure title and caption make the parameter lambda clear.*

Figure 5

• The z axis could be labeled

• I know this is nitpicky, but I still think the figure titles could be made so that they do not overlap with the grid. That would make the figure look cleaner.

***Authors' Response:*** *Thanks for the attention to detail, we appreciate the improvements to the paper. Changes made.*

Figure 7

• There is enough space to label the zone of increased waterflux on the figure panel

• This zone could also be shaded in light grey or light green to make clear that it is a zone

• The same could be done in Figures 11 and 12 in which the shading would go until the end of the experiment.

*Authors' Response: Change made*

Figure 8

• On panel b, the legend could be moved down to not overlap with data

*Authors' Response: Change made*

**Supporting information comments**

Page 1 point 5 – subpoint 1: I think it should read "The volume of sediment […] so that it's elevation is equal to the downstream, node".

Page 7 Paragraph 4 (last paragraph), L6: "valley width is generally increased".

Page 7 paragraph 4 L13: delete "carved"

Page 9 L4: "actual valley width that emerges"

*Authors' Response: Suggested changes made to supplementary materials.*

Figure 2

• In panel a, I would suggest to shade the two possible lateral nodes red and add a label on the figure that indicates that these two nodes are chosen at random

*Authors' Response: changes made to figure 2.*

Figure 5

• I got confused why the model with the spinup (panel a) shows lateral erosion right at t=0 whereas the other model (panel b) shows lateral erosion only after c.a. t=200. Maybe expand upon that

• The time is missing units (I presume y or ky?)

*Authors' Response: Added text to clarify the above and made changes to figure.*

Figure 7

• I note that the direction of the y axis (decreasing values downstream) is inconsistent with the direction of equivalent figures in the main manuscript. That led to a short confusion

*Authors' Response: Yes, changes to the landlab code between running the models that are discussed in the main manuscript and models shown in the supplementary materials caused the y axis to be reversed.*

Figure 8

• Same comment with the shading and labeling the zone of increased water flux as in the main manuscript

*Authors' Response:* Changes made to figure.

Figure 9

• The axes of panel c and f could probably be changed so that all data is included on the figure.

• Same comment with the shading and labeling the zone of increased water flux as in the main manuscript

*Authors' Response:* Changes made to figure.

Figure 10

• Why is t = 0 and t = 25ky here whereas all other models (Figs 11-12) the time steps shown are t=50 and t=75?

*Authors' Response:* Changes made to figure labels.

*Authors' Response to Reviewer 2:*

Dimitri Lague, October 2016:

After going through the response to reviewers, the modified version of the paper and the supplementary material, I find the MS has significantly improved. The authors have provided a detailed response to the reviewer's comments, acknowledging some of the limitations of their model and the new MS answers most of the comments.

I have however few remaining comments regarding this revised version that prevent me to recommend its publication as it is:

• The channel width/pixel size issue: new modelling results are shown in the supplementary material for which the lateral erosion is scaled by W/dx. This is interesting and welcome, but does not solve the fundamental issue that flow has to be split over two pixels (or more) when W>dx. Without actually modelling this, you can't highlight the deficiencies of the current model. I'm fine with that, and I see the value in testing new ideas on an old LEM formalism.

However, I find the authors have a tendency to go very far into details of so called "physical processes" (centripetal forces, details of bank erosion etc…), but when the basics of the water balance and routing are not met, they tend to not really care [AL: matter] too much. I was particularly disappointed to not see this point discussed in the new section 6.2. A few lines are needed here to reflect on whether single D8 models on a fixed grid are really suitable for this kind of "high resolution approach" where the pixel size is of similar size than the channel width.

• The supplementary material shows that there can be a strong dependency of model results on grid size (e.g., fig 8 in suppl material, compare dx= 10 and dx=20), something not wanted in numerical modelling. It is honest from the authors to show it. But again, I was disappointed to not see this point mentioned in section 6.2 "model limitation". This should have been at the start, as a …"model limitation" of the numerical solution. This would give a slightly different perspective to the reader on whether a regular grid using a D8 flow algorithm is really suitable to explore channel mobility.

***Authors' Response:*** *In response to the reviewer's above concerns, we added text that draw the reader's attention to the model limitations we highlight in the supplementary materials section and note some of the significant findings. We added two paragraphs at the beginning of section 7.2 (formerly section 6.2) that (1) highlight how the simplified hydrological assumptions may affect model outcomes when implied channel width approaches or is greater than the cell size and (2) discuss the effects of grid resolution on lateral erosion and valley width.*

• A point that I had not raised in the previous version, is that many of the wide terrasses and present day wide bedrock valleys are actually generated by braiding systems that cannot be modelled by single pixel flow D8 algorithm…another model limitation.

***Authors' Response:*** *We address how this model is not limited to modeling meandering rivers in the "numerical implementation" section: "Lateral erosion rate presented here (Equation 13b 
[revised manuscript text omitted]

---

## Author Response (AR3)

November 20, 2017

Dear Niels and Jens,

Thank you for the comments. We have made the technical corrections suggested and are happy to present the final manuscript for publication. It's been a pleasure working with Jens and the reviewers who improved this paper with many insightful comments through the two rounds of review. We also appreciate the fast response times we had while working with ESurf.

Best,

Abigail Langston and Greg Tucker

Editor Decision: Publish subject to technical corrections (14 Nov 2017) by Niels Hovius

Comments to the Author:        Dear Abigail and Greg,

Thank you for working with AE Jens Turowski and his referees to optimize your manuscript. Jens has gone through your revised submission with a fine toothed comb. Please can you consider his suggestions in preparing a final version of the manuscript, which I shall be happy to approve for publication once these technical corrections have been made.

Niels Hovius

Associate Editor Decision: Publish subject to technical corrections (08 Nov 2017) by Jens Turowski

Comments to the Author:

Dear authors,

thanks for the revised version of the paper. I think the work is nearly ready for publication. Please edit the minor points below. Since these are mostly small language points, I do not see the point to hold up the publication process further and have decided on technical corrections. The paper will be accepted as soon as you submit a revised version.

All the best,

Jens Turowski

[revised manuscript text omitted]